# Raw signal segmentation for estimating RNA modification from Nanopore direct RNA sequencing data

**Guangzhao Cheng[1], Aki Vehtari[1], Lu Cheng[1,2]\***

[1]Department of Computer Science, Aalto University, Espoo, Finland; [2]Institute of Biomedicine, University of Eastern Finland, Kuopio, Finland

## eLife Assessment

This study presents SegPore, a **valuable** new method for processing direct RNA nanopore sequencing data, which improves the segmentation of raw signals into individual bases and boosts the accuracy of modified base detection. The evidence presented to benchmark SegPore is **solid**, and the authors provide a fully documented implementation of the method. SegPore will be of particular interest to researchers studying RNA modifications.

**\*For correspondence:**
lu.cheng.ac@gmail.com

**Competing interest:** The authors declare that no competing interests exist.

**Abstract** Estimating RNA modifications from Nanopore direct RNA sequencing data is a critical task for the RNA research community. However, current computational methods often fail to deliver satisfactory results due to inaccurate segmentation of the raw signal. We have developed a new method, SegPore, which leverages a molecular jiggling translocation hypothesis to improve raw signal segmentation. SegPore is a pure white-box model with enhanced interpretability, significantly reducing structured noise in the raw signal. We demonstrate that SegPore outperforms state-of-the-art methods, such as Nanopolish and Tombo, in raw signal segmentation across three large benchmark datasets. Moreover, the improved signal segmentation achieved by SegPore enables SegPore+m6Anet to deliver state-of-the-art performance in site-level m6A identification. Additionally, SegPore surpasses baseline methods like CHEUI in single-molecule level m6A identification.

## Introduction

RNA modifications play important roles in different diseases, such as Acute Myeloid Leukemia (*Yankova et al., 2021*) and Fragile X Syndrome (*Prieto et al., 2020*), as well as fundamental biological processes like cell differentiation (*Bellodi et al., 2013*; *Lee et al., 2019*) and immune response (*Quin et al., 2021*). To date, researchers have identified over 150 different types of RNA modifications (*Boccaletto et al., 2022*; *Zimna et al., 2023*; *Ohira and Suzuki, 2024*; *Chen et al., 2023*), highlighting the complexity and diversity of RNA regulation. These modifications are essential for proper RNA function, including maintaining secondary structures and facilitating accurate protein synthesis, such as the role of tRNA modifications in decoding mRNA codons (*Agris et al., 2017*).

The practical applications of RNA modifications are far-reaching. For instance, N1-methylpseudouridine (m1 $\Psi$) has been used to enhance the efficacy of COVID-19 mRNA vaccines, highlighting their therapeutic potential (*Nance and Meier, 2021*). However, identifying and characterizing RNA modifications presents significant challenges due to the limitations of existing experimental techniques.

Traditional methods for RNA modification detection, such as MeRIP-Seq (*Meyer et al., 2012*), miCLIP (*Linder et al., 2015*), and m6ACE-Seq (*Koh et al., 2019*), rely on immunoprecipitation techniques that use modification-specific antibodies. While effective, these methods have several

drawbacks. First, each method requires a separate antibody for each RNA modification, making it difficult to study multiple modifications simultaneously. Additionally, these techniques can only infer modification locations from next-generation sequencing (NGS) data, rather than measuring the modifications directly. As a result, they struggle to provide single-molecule resolution, limiting their accuracy and scope.

Recent advancements in direct RNA sequencing (DRS) by Oxford Nanopore Technologies (ONT) offer a promising alternative. DRS allows for the direct measurement of electrical currents as RNA molecules translocate through a nanopore, providing a potential avenue for detecting RNA modifications at the single-molecule level. Two versions of the direct RNA sequencing (DRS) kits are available: RNA002 and RNA004. Unless otherwise specified, this study focuses on RNA002 data. In this technique, five nucleotides (5-mers) reside in the nanopore at a time, and each 5-mer generates a characteristic current signal based on its unique sequence and chemical properties (*Loman et al., 2015*). By analyzing these signals, it is possible to infer the original RNA sequence and detect modifications like N6-methyladenosine (m6A).

The general workflow of Nanopore direct RNA sequencing (DRS) data analysis is as follows. First, the raw electrical signal from a read is basecalled using tools such as Guppy or Dorado (*blawrence-ont and malton-ont, 2026*), which produce the nucleotide sequence of the RNA molecule. However, these base-called sequences do not include the precise start and end positions of each ribonucleotide (or k-mer) in the signal. Because basecalling errors are common, the sequences are typically mapped to a reference genome or transcriptome using minimap2 (*Li, 2018*) to recover the correct reference sequence. Next, tools such as Nanopolish (*Loman et al., 2015*; *Simpson et al., 2017*) and Tombo (*Stoiber et al., 2017*) align the raw signal to the reference sequence to determine which portion of the signal corresponds to each k-mer. We define this process as the segmentation and alignment task (abbreviated as the segmentation task), which is referred to as 'eventalign' in Nanopolish. Based on this alignment, Nanopolish extracts various features—such as the start and end positions, mean, and standard deviation of the signal segment corresponding to a k-mer. This signal segment or its derived features is referred to as an 'event' in Nanopolish. The resulting events serve as input for downstream RNA modification detection tools such as m6Anet (*Hendra et al., 2022*) and CHEUI (*Acera Mateos et al., 2024*).

However, significant computational challenges remain. Segmenting the raw current signal into distinct 5-mers and distinguishing between normal nucleotides and their modified counterparts is a complex task. Current methods, such as Nanopolish, employ change-point detection methods to segment the signal and use dynamic programming methods and Hidden Markov Models (HMM) to align the derived segments to the reference sequence, but they are prone to noise and inaccuracies, which degrade performance in downstream tasks like RNA modification prediction. The root of this issue lies in the fact that these methods do not accurately model the physical process of Nanopore sequencing, particularly the dynamics of the motor protein that drives RNA through the pore. As a result, the segmentation process is not well-aligned with the actual translocation mechanics, leading to signal noise that hinders precise modification detection.

SegPore is a novel tool for direct RNA sequencing (DRS) signal segmentation and alignment, designed to overcome key limitations of existing approaches. By explicitly modeling motor protein dynamics during RNA translocation with a Hierarchical Hidden Markov Model (HHMM), SegPore segments the raw signal into small, biologically meaningful fragments, each corresponding to a k-mer sub-state, which substantially reduces noise and improves segmentation accuracy. After segmentation, these fragments are aligned to the reference sequence and concatenated into larger events, analogous to Nanopolish's 'eventalign' output, which serve as the foundation for downstream analyses. Moreover, the 'eventalign' results produced by SegPore enhance interpretability in RNA modification estimation. While deep learning–based tools such as m6Anet classify RNA modifications using complex, non-transparent features (see *Figure 4—figure supplement 1*), SegPore employs a simple Gaussian Mixture Model (GMM) to distinguish modified from unmodified nucleotides based on baseline current levels. This transparent modeling approach improves confidence in the predictions and makes SegPore particularly well-suited for biological applications where interpretability is essential.

By introducing SegPore, we bridge the gaps left by traditional tools. SegPore utilizes a hierarchical hidden Markov model (HHMM) for more precise segmentation and combines it with signal alignment and a GMM for RNA modification prediction, offering both greater accuracy and interpretability. This

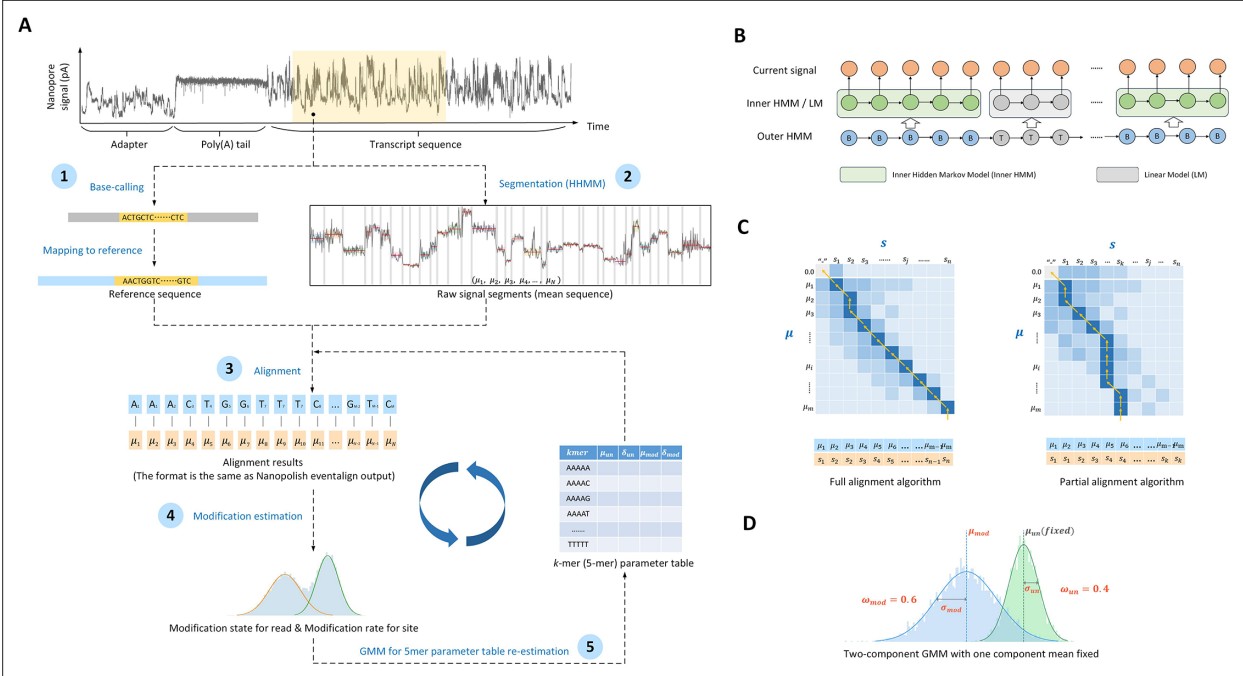

**Figure 1.** SegPore workflow. (**A**) General workflow. The workflow consists of five steps: (1) First, raw current signals are basecalled and mapped using Guppy and Minimap2. The raw current signal fragments are paired with the corresponding reference RNA sequence fragments using Nanopolish. (2) Next, the raw current signal of each read is segmented using a hierarchical hidden Markov model (HHMM), which provides an estimated mean ($\mu_i$) for each segment. (3) These segments are then aligned with the 5-mer list of the reference sequence fragment using a full/partial alignment algorithm, based on a 5-mer parameter table. For example, $A_j$ denotes the base '$A$' at the $j$-th position on the reference. In this example, $A_1$ and $A_2$ refer to the first and second occurrences of '$A$' in the reference sequence, respectively. Accordingly, $\mu_1$ and $\mu_2$ are aligned to $A_1$, while $\mu_3$ is aligned to $A_2$ (4). All signals aligned to the same 5-mer across different genomic locations are pooled together, and a two-component Gaussian Mixture Model (GMM) is used to predict the modification at the site-level or single-molecule level. One component of the GMM represents the unmodified state, while the other represents the modified state. (5) GMM is used to re-estimate the 5-mer parameter table. (**B**) Hierarchical hidden Markov model (HHMM). The outer HMM segments the current signal into alternating base and transition blocks. The inner HMM approximates the emission probability of a base block by considering neighboring 5-mers. A linear model is used to approximate the emission probability of a transition block. (**C**) Full/partial alignment algorithms. Rows represent the estimated means of base blocks from the HHMM, and columns represent the 5-mers of the reference sequence. Each 5-mer can be aligned with multiple estimated means from the current signal. (**D**) Gaussian mixture model (GMM) for estimating modification states. The GMM consists of two components: the green component models the unmodified state of a 5-mer, and the blue component models the modified state. Each component is described by three parameters: mean ($\mu$), standard deviation ($\sigma$), and weight ($\omega$).

integrated approach enables robust m6A modification detection at the single-molecule level, facilitating reliable, transparent predictions for both site-specific and molecule-wide m6A modifications.

# Materials and methods
## SegPore workflow
### Workflow overview
An overview of SegPore is illustrated in *Figure 1A*, which outlines its five-step process. The output of Step 3 is the 'events', which is analogous to the output generated by the Nanopolish (v0.14.0) 'eventalign' command and can be used as input for downstream models such as m6Anet. An 'event' refers to a segment of the raw signal that is aligned to a specific k-mer on a read, along with its associated features such as start and end positions, mean current, standard deviation, and other relevant statistics. Step 4 allows for direct modification prediction at both the site and single-molecule levels. Notably, a key feature of SegPore is the k-mer parameter table, which defines the mean and standard deviation for each k-mer in either an unmodified or modified state. During the training phase, Steps 3~5 are iterated multiple times to stabilize the parameters in the k-mer table, which are subsequently fixed and applied for modification prediction on the test data. A detailed description of the model

is provided in Appendix 1. Unless otherwise noted, the following analysis focuses on RNA002 data, using 5-mers rather than k-mers.

## Preprocessing

We begin by performing basecalling on the input fast5 file using Guppy (v6.0.1), which converts the raw signal data into ribonucleotide sequences. Next, we align the basecalled sequences to the reference genome using Minimap2 (v2.28-r1209) (*Li, 2018*), generating a mapping between the reads and the reference sequences. Nanopolish provides two independent commands: 'polya' and 'eventalign'. The 'polya' command identifies the adapter, poly(A) tail, and transcript region in the raw signal, which we refer to as the poly(A) detection results. The raw signal segment corresponding to the poly(A) tail is used to standardize the raw signal for each read. The 'eventalign' command aligns the raw signal to a reference sequence, assigning a signal segment to individual k-mers in the reference. It also computes summary statistics (e.g. mean, standard deviation) from the signal segment for each k-mer. Each k-mer together with its corresponding signal features is termed an event. These event features are then passed into downstream tools such as m6Anet and CHEUI for RNA modification detection. For full transcriptome analysis (Figure 3), we extract the aligned raw signal segment and reference sequence segment from Nanopolish's events for each read by using the first and last events as start and end points. For in vitro transcription (IVT) data with a known reference sequence (Figure 4), we extract the raw signal segment corresponding to the transcript region for each input read based on Nanopolish's poly(A) detection results.

Due to inherent variability between nanopores in the sequencing device, the baseline levels and standard deviations of k-mer signals can differ across reads, even for the same transcript. To standardize the signal for downstream analyses, we extract the raw current signal segments corresponding to the poly(A) tail of each read. Since the poly(A) tail provides a stable reference, we standardize the raw current signals for each read, ensuring that the mean and standard deviation are consistent across the poly(A) tail region. This step is crucial for reducing variability between different reads and ensuring more accurate signal segmentation and modification prediction in subsequent steps. See Section 3 of Appendix 1 for more details.

## Signal segmentation via hierarchical Hidden Markov model

The RNA translocation hypothesis (see details in the first section of *Results*) naturally leads to the use of a hierarchical Hidden Markov Model (HHMM) to segment the raw current signal. As shown in *Figure 1B*, the HHMM consists of two layers. The outer HMM divides the raw signal into alternating base and transition blocks, represented by hidden states 'B' (base) and 'T' (transition). Within each base block, the inner HMM models the current signal at a more granular level.

The inner HMM includes four hidden states: 'prev', 'next', 'curr', and 'noise'. These correspond to the previous, next, and current 5-mer in the pore, while 'noise' refers to random fluctuations. Each raw current measurement is emitted from one of these hidden states, providing a detailed model of the signal within the base blocks. A linear model with a large absolute slope is used to represent sharp changes in the transition blocks.

To segment the signal, we first model the likelihood of the HHMM. Given the raw current signal $\boldsymbol{y}$ of a read, the hidden states of the outer hidden HMM are denoted by $\boldsymbol{g}$. $\boldsymbol{y}$ and $\boldsymbol{g}$ are divided into $2K+1$ blocks $c$, where $\boldsymbol{y}^{(k)}$, $\boldsymbol{g}^{(k)}$ correspond to $k$th block and $\boldsymbol{c} = (c_1, c_2, \ldots, c_k, \ldots, c_{2K+1})$, $c_k \in \{B, T\}$. Blocks with odd indices ($k = 1, 3, 5, \ldots, 2K+1$) are base blocks, while those with even indices ($k = 2, 4, \ldots, 2K$) are transition blocks. The likelihood of the HHMM is given by

$$p(\boldsymbol{y}, \boldsymbol{g}) = p(\boldsymbol{y}|\boldsymbol{g}) \, p(\boldsymbol{g}) \tag{1}$$

$$= p(\boldsymbol{y}|\boldsymbol{g}) \left\{ \pi_{g_1}^{outer} \prod_{i=2}^{N} T_{g_{i-1}g_i}^{outer} \right\} \tag{2}$$

$$= \left\{ \prod_{i=1}^{N} p(y_i|g_i) \right\} \left\{ \pi_{g_1}^{outer} \prod_{i=2}^{N} T_{g_{i-1}g_i}^{outer} \right\} \tag{3}$$

$$= \left\{ \prod_{k=0}^{K} p\left( \mathbf{y}^{(2k+1)} | c_{2k+1} = \mathbf{B} \right) \right\} \left\{ \prod_{k=1}^{K} p\left( \mathbf{y}^{(2k)} | c_{2k} = \mathbf{T} \right) \right\} \left\{ \pi_{g_1}^{outer} \prod_{i=2}^{N} T_{g_{i-1}g_i}^{outer} \right\} \qquad (4)$$

where $T_{g_{i-1}g_i}^{outer}$ is the transition matrix of the outer HMM and $\pi_{g_1}^{outer}$ is the probability for the first hidden state. The first part of *Equation 2* is emission probabilities, and the second part is the transition probabilities. It is not possible to directly compute the emission probabilities of the outer HMM (*Equation 3*) since there exist dependencies for the current signal measurements within a base or transition block. Therefore, we use the inner HMM and linear model (*Equation 4*) to handle the dependencies and approximate emission probabilities.

The inner HMM models transition between the 'prev', 'next', 'curr', and 'noise' states. For the 'prev', 'next', and 'curr' states, a Gaussian distribution is used to model the emission probabilities, while a uniform distribution models the 'noise' state. The forward-backward algorithm is used to compute the marginal likelihood of the inner HMM, which approximates the emission probabilities for base blocks. For transition blocks, a standard linear model computes the emission probabilities.

For any given $g$, we can calculate the joint likelihood (*Equation 1*). We enumerate different configurations and select the one with the highest likelihood. This process segments the raw current signal into alternating base and transition blocks, where one or more base blocks may correspond to a single 5-mer. Each base block is characterized by its mean and standard deviation, which are used as input for downstream alignment tasks.

Due to the large size of fast5 files (which can reach terabytes), parameter inference for this model is computationally intensive. To address this, we developed a GPU-based inference algorithm that significantly accelerates the process. More details on this algorithm can be found in Appendix 2.

In summary, the HHMM allows us to accurately segment the raw current signal, providing estimates of the mean and variance for each base block, which are crucial for downstream analyses such as RNA modification prediction.

## 5-mer and 9-mer parameter table

We downloaded the k-mer models 'r9.4_180 mv_70bps_5-mer_RNA' from the ONT GitHub repository (here; *Brennen, 2023*) for RNA002 data. The columns labeled 'level_mean' and 'level_stdv' in these models were used as the mean and standard deviation values for the unmodified 5-mers. These values serve as the initial parameters in the 5-mer parameter table for SegPore, which we refer to as the *ONT 5-mer parameter table*. In the RNA004 data analysis, we obtained the 9-mer parameter table from the source code of f5c (version 1.5). Specifically, we used the array named 'rna004_130bps_u_to_t_rna_9mer_template_model_builtin_data' from the file here (accessed on 17 October 2025).

The initialization of the k-mer parameter table is a critical step in SegPore's workflow. By leveraging ONT's established k-mer models, we ensure that the initial estimates for unmodified k-mers are grounded in empirical data. These initial estimates are refined through SegPore's iterative parameter estimation process, enabling the model to accurately differentiate between modified and unmodified k-mers during segmentation and modification prediction tasks. The refined k-mer parameters also provide a foundation for downstream analysis, such as alignment and m6A identification.

## Alignment of raw signal segment with reference sequence

After segmenting the raw current signal of a read into base and transition blocks using HHMM, we align the means of base blocks with the 5-mer list of the reference sequence.

The alignment process involves three main cases:

1. Base block matching with a 5-mer: In this case, a base block aligns with a 5-mer, which is modeled by a Gaussian distribution. The 5-mer may exist in either an unmodified or modified state, and the corresponding Gaussian parameters are retrieved from the 5-mer parameter table.
2. Base block matching with an insertion: Here, the base block aligns with an indel ('-'), indicating an inserted nucleotide in the read.
3. Deletion in the read: In this case, an indel (0.0) matches with a 5-mer, representing a deleted nucleotide in the read.

The alignment score function models these matching cases as follows. For the first case (a base block matching a 5-mer), we calculate the probability that the base block mean is sampled from either

the unmodified or modified 5-mer's Gaussian distribution. The larger of the two probabilities is used as the match score. For the second and third cases, the alignment is treated as noise, and a fixed uniform distribution is used to calculate the match score.

An important distinction from classical global alignment algorithms (Needleman–Wunsch algorithm) is that one or multiple base blocks may align with a single 5-mer. Given the base block means $\boldsymbol{\mu} = (\mu_1, \mu_2, \ldots, \mu_i, \ldots, \mu_m)$ and 5-mer list $\boldsymbol{s} = (s_1, s_2, \ldots, s_j, \ldots, s_n)$, we define $(m+1) \times (n+1)$ the score matrix as $\boldsymbol{M}$ (*Figure 1C*). The first row and column of the matrix are reserved for indels ("0.0" and "-"), representing insertions or deletions in the base blocks or 5-mers, respectively.

We denote the score function by $f$. The recursion formula of the dynamic programming alignment algorithm is given by

$$\boldsymbol{M}(i,j) = max \begin{cases} \boldsymbol{M}(i-1, j-1) + f(\mu_i, s_j) \\ \boldsymbol{M}(i, j-1) + f(0.0, s_j) \\ \boldsymbol{M}(i-1, j) + f(\mu_i, \text{-}) \\ \boldsymbol{M}(i-1, j) + f(\mu_i, s_j) \end{cases}, \tag{5}$$

where $f$ is the score function. It can be seen that we can still align $\mu_i$ with $s_j$ after we have aligned $\mu_{i-1}$ with $s_j$, which fulfills the special consideration that one or multiple base blocks might be aligned with one 5-mer.

We implement two types of alignment algorithms (*Figure 1C*) based on the score matrix $\boldsymbol{M}$:

1. Full Alignment Algorithm: This algorithm aligns the full list of base block means with the full list of 5-mers, similar to classical global alignment. It traces back from the $(m+1, n+1)$ position in the score matrix.
2. Partial Alignment Algorithm: This aligns the full list of base blocks with the initial part of the 5-mer list, with no indels allowed in the base block means or the 5-mer list. The trace back starts from the maximum value in the last row of the score matrix.

A detailed description of both alignment algorithms is provided in Appendix 1. The output of the alignment algorithm is an alignment that pairs the base blocks with the 5-mers from the reference sequence for each read (*Figure 1C*). Base blocks aligned to the same 5-mer are concatenated into a single raw signal segment (referred to as an 'event'), from which various features—such as start and end positions, mean current, and standard deviation—are extracted. A detailed derivation of the mean and standard deviation is provided in Section 5.3 in Appendix 1. In the remainder of this paper, we refer to these resulting events as the output of eventalign analysis, which also represents the final output of the segmentation and alignment task.

## Modification prediction

After obtaining the eventalign results, we estimate the modification state of each motif using the 5-mer parameter table. Specifically, for each 5-mer, we compare the probability that its mean is sampled from either the modified or unmodified 5-mer's Gaussian distribution. If the probability under the modified 5-mer distribution is higher, the 5-mer is predicted to be in the modification state, and vice versa for the unmodified state.

To estimate the overall modification rate at a specific genomic location, we pool all reads that map to that location on the reference sequence. The modification rate is calculated as the proportion of reads that are predicted to be in the modification state at that location. A detailed description of the modification state prediction process can be found in Appendix 1.

## GMM for 5-mer parameter table re-estimation

To improve alignment accuracy and enhance modification predictions, we use a Gaussian Mixture Model (GMM) to iteratively fine-tune the 5-mer parameter table. As illustrated in *Figure 1A*, the rows of the 5-mer parameter table represent the 5-mers, while the columns provide the mean and standard deviation for both the unmodified and modified states. We denote a 5-mer by $s$, with its relevant parameters as $\mu_{s,un}$, $\delta_{s,un}$, $\mu_{s,mod}$, $\delta_{s,mod}$.

Using the alignment results from all reads, we collect the base block means aligned to the same 5-mer (denoted by $s$) across different reads and genomic locations, particularly those with high modification rates. A two-component GMM is then fit to these base blocks. In this process, the mean of the first component is fixed to $\mu_{s,un}$. From the GMM, we update the parameters $\delta_{s,un}$, $\omega_{s,un}$, $\mu_{s,mod}$, $\delta_{s,mod}$, and $\omega_{s,mod}$, where $\omega_{s,un}$, $\omega_{s,mod}$ represent the weights for the unmodified and modified components, respectively. Afterward, we manually adjust the 5-mer parameter table using heuristics to ensure that the modified 5-mer distribution is significantly distinct from the unmodified distribution (see details in Section 7 of Appendix 1).

This re-estimation process is only performed on the training data. The initial 5-mer parameter table is based on the table provided by ONT. After each iteration of updating the table, we rerun the SegPore workflow from the alignment step onward. Typically, the process is repeated three to five times until the 5-mer parameter table stabilizes (when the average change of mean values of all 5-mers is less than 5e-3). Once a stabilized 5-mer parameter table is estimated from the training data, it is used for RNA modification estimation in the test data without further updates. A more detailed description of the GMM re-estimation process is provided in Section 6 of Appendix 1.

## m6A site level benchmark

The HEK293T wild-type (WT) samples were downloaded from the ENA database (accession number PRJEB40872), while the HCT116 samples were obtained from ENA under accession PRJEB44348. The reference sequence (Homo_sapiens.GRCh38.cdna.ncrna_wtChrIs_modified.fa) was downloaded from https://doi.org/10.5281/zenodo.4587661. Ground truth data were sourced from Supplementary Data 1 of *Pratanwanich et al., 2021*. Fast5 files for the test dataset (mESC WT samples, mESCs_Mettl3_WT_fast5.tar.gz; *Jenjaroenpun et al., 2021*) were retrieved from the NCBI Sequence Read Archive (SRA) under accession SRP166020.

During training, we initialized the 5-mer parameter table using ONT's data. The standard SegPore workflow was executed on the training data (HEK293T WT samples), using the full alignment algorithm. The 5-mer parameter table estimation was iterated five times. For mapping, reads were first aligned to the cDNA and subsequently converted to genomic locations using Ensembl GTF file (GRCh38, v9). The same 5-mer across different genomic locations was pooled together. A 5-mer was considered significantly modified if its read coverage exceeded 1500 and the distance between the means of the two Gaussian components in the GMM was greater than 5 picoamperes (pA). As a result, modification parameters were specified for ten significant 5-mers, as illustrated in *Figure 3—figure supplement 1A*.

With the estimated 5-mer parameter table from the training data, we then ran the SegPore workflow on the test data. The transcript sequences from GENCODE release version M18 were used as the reference sequence for mapping, with the corresponding GTF file (gencode.vM18.chr_patch_hapl_scaff.annotation.gtf) downloaded from GENCODE to convert transcript locations into genomic coordinates. It is important to note that the 5-mer parameter table was not re-estimated for the test data. Instead, modification states for each read were directly estimated using the fixed 5-mer parameter table. Due to the differences between human (*Figure 3—figure supplement 1A*) and mouse (*Figure 3—figure supplement 1B*), only six 5-mers were found to have m6A annotations in the test data's ground truth (*Figure 3—figure supplement 1C*). For a genomic location to be identified as a true m6A modification site, it had to correspond to one of these six common 5-mers and have a read coverage of greater than 20. SegPore derived the ROC and PR curves for benchmarking based on the modification rate at each genomic location.

In the SegPore+m6Anet analysis, we fine-tuned the m6Anet model using SegPore's eventalign results to demonstrate improved m6A identification. We started with the pre-trained m6Anet model (available at here; *Hendra, 2025*, model version: HCT116_RNA002) and fine-tuned it using the eventalign results from HCT116 samples. SegPore's eventalign output provided the pairing between each genomic location and its corresponding raw signal segment, which allowed us to extract normalized features such as the normalized mean $\mu_i$, standard deviation $\sigma_i$, dwell time $l_i$ (number of data points in the event). For genomic location $i$, m6Anet extracts a feature vector $x_i = \{\mu_{i-1}, \sigma_{i-1}, l_{i-1}, \mu_i, \sigma_i, l_i, \mu_{i+1}, \sigma_{i+1}, l_{i+1}\}$, which was used as input for m6Anet. Feature vectors from a randomly selected 80% of the genomic locations were used for training, while the remaining 20% were set aside for validation. We ran 100 epochs during fine-tuning, selecting the model that performed best on the validation set.

Ground truth data and the performance of other methods (Tombo v1.5.1, Nanom6A v2.0, m6Anet v1.0, and Epinano v1.2.0) on the mESC dataset were provided through personal communications with Prof. Luo Guanzheng, the corresponding author of the benchmark study referenced (*Zhong et al., 2023*).

## m6A single molecule level benchmark

The benchmark IVT data for single molecule m6A identification was downloaded from NCBI-SRA with accession number SRP166020. CHEUI was used for the benchmark. For the ground truth, every A in every read of the IVT_m6A sample is treated as a m6A modification and every A in every read of the IVT_normalA is treated as a normal A. Detailed methods for calculating modification probability at single molecule level were provided in Appendix 1, 'Modification state estimation on single molecule level' and 'Modification probability estimation on single molecule level'.

# Results

## RNA translocation hypothesis

Accurate segmentation of raw current signals in direct RNA sequencing remains a major challenge, largely because the precise dynamics of RNA translocation through the pore are not fully understood. To address this, we propose a hypothesis that better reflects the physical movement of the RNA molecule. In traditional basecalling algorithms such as Guppy and Albacore, we implicitly assume that the RNA molecule is translocated through the pore by the motor protein in a monotonic fashion,

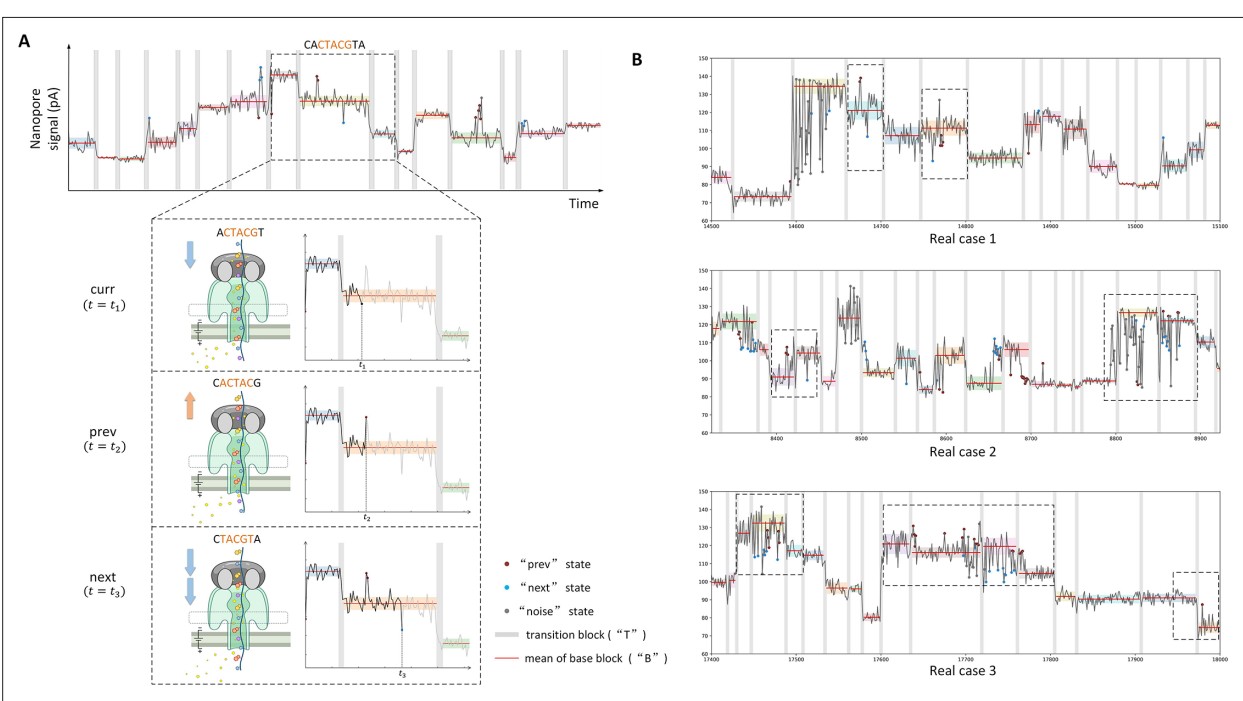

**Figure 2.** RNA translocation hypothesis. (**A**) Jiggling RNA translocation hypothesis. The top panel shows the raw current signal of Nanopore direct RNA sequencing, with gray areas representing SegPore-estimated transition blocks. We focus on three neighboring 5-mers, considering the central 5-mer (CTACG) as the current 5-mer. The RNA molecule may briefly move forward or backward during the translocation of the current 5-mer. If the RNA molecule is pulled backward, the previous 5-mer is placed in the pore, and the current signal ("prev" state, red dots) resembles the previous 5-mer's baseline (mean and standard deviation highlighted by red lines and shades). If the RNA is pushed forward, the current signal ("next" state, blue dots) is similar to the next 5-mer's baseline. (**B**) Example raw current signals supporting the jiggling hypothesis. The dashed rectangles highlight base blocks, with red and blue points representing measurements corresponding to the previous and next 5-mer, respectively. Red points align closely with the previous 5-mer's baseline, and blue points match the next 5-mer's baseline, reinforcing the hypothesis that the RNA molecule jiggles between neighboring 5-mers. The raw current signals were extracted from mESC WT samples of the training data in the m6A benchmark experiment.

The online version of this article includes the following figure supplement(s) for figure 2:

**Figure supplement 1.** RNA translocation hypothesis illustrated on RNA004 data.

that is the RNA is pulled through the pore unidirectionally. In the DNN training process of Guppy and Albacore, we try to align the current signal with the reference RNA sequence. The alignment is unidirectional, which is the source of the implicit monotonic translocating assumption.

Contrary to the conventional assumption of monotonic translocation, the raw current data suggests that the motor protein drives RNA both forward and backward during sequencing. *Figure 2B* illustrates this with several example fragments of DRS raw current signal (*Pratanwanich et al., 2021*), where each fragment roughly corresponds to three neighboring 5-mers. The raw current signals, as shown in *Figure 2B*, strongly support this hypothesis, with several instances of measured current intensities matching both the previous and next 5-mer's baseline. These repeated patterns, observed across multiple DRS datasets, provide empirical evidence of the jiggling translocation. Similar patterns are widely observed across the whole data. This suggests that the RNA molecule may move forward and backward while passing through the pore. This observation is also supported by previous reports (*Craig et al., 2017*; *Caldwell and Spies, 2017*), in which the helicase (the motor protein) translocates the DNA strand through the nanopore in a back-and-forth manner. Depending on ATP or ADP binding, the motor protein may translocate the DNA/RNA forward or backward by 0.5~1 nucleotides.

Based on the reported kinetic model (*Craig et al., 2017*), we hypothesize that the RNA is translocated through the pore in a jiggling manner. This "jiggling" hypothesis presents a significant departure from the traditionally accepted view of unidirectional RNA translocation and aligns better with recent evidence from studies on DNA translocation (*Craig et al., 2017*; *Caldwell and Spies, 2017*). Incorporating this hypothesis into segmentation algorithms could lead to more accurate predictions of RNA modifications by accounting for the dynamic nature of translocation. On average, the motor protein sequentially translocates 5-mers on the RNA strand forward, and each 5-mer resides in the pore for a short time. During the short period of a single 5-mer, the motor protein may swiftly drive the RNA molecule forward and backward by 0.5~1 nucleotide in the translocation process of the current 5-mer (*Figure 2A*), which makes the measured current intensity occasionally similar to the previous or the next 5-mer. When the motor protein does not move the RNA molecule, we hypothesize that the RNA molecule undergoes slight thermal fluctuations, causing it to oscillate slightly within the pore and produce a stable current close to its baseline. In contrast, sharp changes in current intensity between consecutive 5-mers define transition blocks, where one 5-mer is replaced by the next.

We also assume that the raw current signal of a read can be segmented into a series of alternating base and transition blocks. In the ideal case, a base block corresponds to the base state where the 5-mer resides in the pore and jiggles between neighboring 5-mers, that is the current 5-mer can transiently jump to the previous or the next 5-mer. A transition block corresponds to the transition state between two consecutive base states where one 5-mer translocates to the next 5-mer in the pore. The current signal should be relatively flat in the base blocks, while a sharp change is expected in the transition blocks. One challenge we encountered was the overestimation of transition blocks. This may be due to a base block actually corresponding to a sub-state of a single 5-mer, rather than each base block corresponding to a full 5-mer, leading to inflated transition counts. To address this issue, SegPore's alignment algorithm was refined to merge multiple base blocks (which may represent substates of the same 5-mer) into a single 5-mer, thereby facilitating further analysis.

## SegPore Workflow

The SegPore workflow (*Figure 1*) consists of five key steps: (1) Preprocess fast5 files to pair raw current signal segments with corresponding RNA sequence fragments; (2) Segment each raw current signal using a hierarchical hidden Markov model (HHMM) into base and transition blocks; (3) Align the derived base blocks with the paired RNA sequence; (4) Fit a two-component GMM to estimate the modification state at the single-molecule level or the modification rate at the site level; (5) Use the results from Step (4) to update relevant parameters. Steps (3) to (5) are iterative and continue until the estimated parameters stabilize.

The final outputs of SegPore are the events and modification state predictions. SegPore's events are similar to the outputs of Nanopolish's 'eventalign' command, in that they pair raw current signal segments with the corresponding RNA reference 5-mers. Each 5-mer is associated with various features, such as start and end positions, mean current, and standard deviation, derived from the paired signal segment. For 5-mers that exhibit one clearly unmodified component and one clearly

**Table 1.** Segmentation benchmark on RNA002 data.

| Test Dataset | Avg. std ($\sigma$) ↓ | | | Avg. log p ($L$) ↑ | | |
|---|---|---|---|---|---|---|
| | Nanopolish | Tombo | SegPore | Nanopolish | Tombo | SegPore |
| HEK293T(WT) | 3.073 | 4.187 | **2.736** | –2.871 | –3.749 | **–2.778** |
| HEK293T(KO) | 2.948 | 4.204 | **2.670** | –2.856 | –3.749 | **–2.745** |
| HCT116 | 3.167 | 4.076 | **2.872** | –2.872 | –3.704 | **–2.746** |

modified component, SegPore reports the modification rate at each site, as well as the modification state of that site on individual reads.

A key component of SegPore is the 5-mer parameter table, which specifies the mean and standard deviation for each 5-mer in both modified and unmodified states (*Figure 1A*). Since the peaks (representing modified and unmodified states) are separable for only a subset of 5-mers, SegPore can provide modification parameters for these specific 5-mers. For other 5-mers, modification state predictions are unavailable.

## Segmentation benchmark

To evaluate SegPore's performance in raw signal segmentation, we compared SegPore with Nanopolish (v0.14.0) and Tombo (v1.5.1) using three Nanopore direct RNA sequencing (DRS) datasets: two HEK293T datasets (wild type and Mettl3 knockout) (*Pratanwanich et al., 2021*) and the HCT116 dataset (*Chen et al., 2025*). Nanopolish and SegPore employed the 'eventalign' method to align 5-mers on each read with their corresponding raw signals, producing the mean and standard deviation (std) of the aligned signal segments. Tombo used the 'resquiggle' method to segment the raw signals, but the resulting signals are not reported on the absolute pA scale. To ensure a fair comparison with SegPore, we standardized the segments using the poly(A) tail in the same way as SegPore (See preprocessing section in Materials and methods).

To benchmark segmentation performance, we used two key metrics (details provided in Appendix 1, Section 8): (1) the log-likelihood of the segment mean, which measures how closely the segment matches ONT's 5-mer parameter table (used as ground truth), and (2) the standard deviation (std) of the segment, where a lower std indicates reduced noise and better segmentation quality. If the raw signal segment aligns correctly with the corresponding 5-mer, its mean should closely match ONT's reference, yielding a high log-likelihood. A lower std of the segment reflects less noise and better performance overall.

As shown in *Table 1*, SegPore consistently achieved the best performance averaged on all 5-mers across all datasets, with the highest log-likelihood and the lowest std values. These results suggest that SegPore provides a more accurate segmentation of the raw signal with significantly reduced noise compared to Nanopolish and Tombo. It is worth noting that the data points corresponding to the transition state between two consecutive 5-mers are not included in the calculation of the standard deviation in SegPore's results in *Table 1*. However, their exclusion does not affect the overall conclusion. Because SegPore contains on average ~6 points per event within the transition state, we similarly removed the first and last three data points of each event for Nanopolish and Tombo prior to recalculating the metrics. The updated results are presented in *Table 2*.

To provide a more intuitive comparison, the segmentation results for two example raw signal clips are illustrated in *Figure 3—figure supplement 2*. These examples demonstrate the clearer, more precise segmentation achieved by SegPore compared to Nanopolish.

**Table 2.** Segmentation benchmark on the RNA002 data excluding boundaries.

| Test Dataset | Avg. std ($\sigma$) ↓ | | | Avg. log p ($L$) ↑ | | |
|---|---|---|---|---|---|---|
| | Nanopolish | Tombo | SegPore | Nanopolish | Tombo | SegPore |
| HEK293T(WT) | 2.862 | 4.090 | **2.736** | –2.838 | –3.772 | **–2.778** |
| HEK293T(KO) | 2.856 | 4.120 | **2.670** | –2.878 | –3.729 | **–2.745** |
| HCT116 | 3.058 | 4.053 | **2.872** | –2.869 | –3.708 | **–2.746** |

**Table 3.** Segmentation benchmark on RNA004 data.

| Test Dataset | Avg. std ($\sigma$) ↓ | | | Avg. log p ($L$) ↑ | | |
|---|---|---|---|---|---|---|
| | f5c | Uncalled4 | SegPore | f5c | Uncalled4 | SegPore |
| *S. cerevisiae* | 3.129 | 3.970 | **3.125** | −2.541 | −3.835 | **−2.489** |
| curlcake(IVT) | 3.424 | 4.729 | **3.359** | −2.716 | −2.904 | **−2.515** |
| curlcake(m6A) | 3.520 | 4.971 | **3.392** | −3.118 | −3.524 | **−2.599** |

To evaluate SegPore's performance on RNA004 data, we compared it with f5c (v1.5) (*Gamaarachchi et al., 2020*) and Uncalled4 (*Kovaka et al., 2025*) (v4.1.0) using three public DRS datasets: the *S. cerevisiae* dataset (*Watson et al., 2025*), the curlcake IVT and m6A datasets (*Cruciani et al., 2025*). The RNA002 data provides reference current levels for 5-mers, whereas RNA004 provides reference values for 9-mers, with Uncalled4 normalizing them to approximately zero mean and unit variance. As there are currently no established poly(A) detection methods available for RNA004, we used f5c to standardize the raw signals of each read prior to segmentation. SegPore was then applied to perform segmentation on the standardized signals. We computed the same benchmarking metrics—average log-likelihood and standard deviation—using the standardized raw signals and the segmentation results from f5c, Uncalled4, and SegPore. The 9-mer parameter table in pA scale for RNA004 data provided by f5c (see Materials and methods) was used in the analysis. As shown in *Table 3*, SegPore achieved the best overall performance across all datasets, indicating its robustness and suitability for RNA004 data. Moreover, we find that the jiggling hypothesis remains valid for RNA004, as illustrated in *Figure 2—figure supplement 1*.

## m6A identification at the site level

We evaluated SegPore's performance in raw signal segmentation and m6A identification using independent public datasets as both training and test data. Since m6A modifications typically occur at DRACH motifs (where D denotes A, G, or U, and H denotes A, C, or U) (*Linder et al., 2015*), this study focuses on estimating m6A modifications on these motifs.

To begin, we estimated the 5-mer parameter table for m6A modifications using Nanopore direct RNA sequencing (DRS) data from three wild-type human HEK293T cell samples (*Pratanwanich et al., 2021*). The fast5 files from all samples were concatenated, and the full SegPore workflow was run to obtain the 5-mer parameter table. The parameter estimation process was iterated five times to ensure stabilization, refining the modification parameters for ten 5-mers where the modification state distribution significantly differs from the unmodified state.

Next, we applied SegPore's segmentation and m6A identification to test data from wild-type mouse embryonic stem cells (mESCs; *Zhong et al., 2023*). Given the comparable methods and input data requirements, we benchmarked SegPore against several baseline tools, including Tombo, MINES (*Lorenz et al., 2020*), Nanom6A (*Gao et al., 2021*), m6Anet, Epinano (*Liu et al., 2019*), and CHEUI (*Acera Mateos et al., 2024*). By default, MINES and Nanom6A use eventalign results generated by Tombo, while m6Anet, Epinano, and CHEUI rely on eventalign results produced by Nanopolish. In *Figure 3C*, 'Nanopolish +m6Anet" refers to the default m6Anet pipeline, whereas 'SegPore +m6Anet' denotes a configuration in which Nanopolish's eventalign results are replaced with those from SegPore. Based on the output of SegPore eventalign, we fine-tune m6Anet using the HCT116 data at all DRACH motifs, aiming to demonstrate that the performance of SegPore benefits downstream models. Additionally, due to the differences in the availability of ground truth labels for specific k-mer motifs between human and mouse (*Figure 3—figure supplement 1*), six shared 5-mers were selected to demonstrate SegPore's performance in modification prediction directly. By utilizing the 5-mer parameter table derived from the training data, SegPore employs a two-component GMM to calculate the modification rates at the selected m6A sites.

SegPore demonstrated improved segmentation compared to Nanopolish. *Figure 3A* shows the eventalign results at an example genomic location with m6A modifications. SegPore's results show a more pronounced bimodal distribution in the raw signal segment mean, indicating clearer separation of modified and unmodified signals. Furthermore, when pooling all reads mapped to m6A-modified

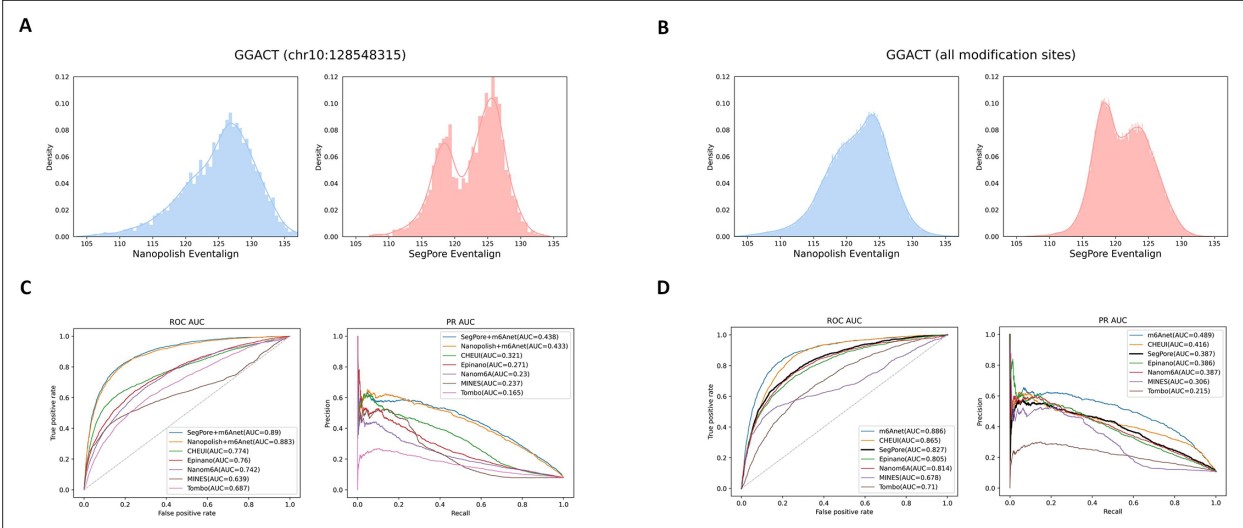

**Figure 3.** m6A identification at the site level. (**A**) Histogram of the estimated mean from current signals mapped to an example m6A-modified genomic location (chr10:128548315, GGACT) across all reads in the training data, comparing Nanopolish (left) and SegPore (right). The x-axis represents current in picoamperes (pA). (**B**) Histogram of the estimated mean from current signals mapped to the GGACT motif at all annotated m6A-modified genomic locations in the training data, again comparing Nanopolish (left) and SegPore (right). The x-axis represents current in picoamperes (pA). (**C**) Site-level benchmark results for m6A identification across all DRACH motifs, showing performance comparisons between SegPore+m6Anet and other methods. (**D**) Benchmark results for m6A identification on six selected motifs at the site level, comparing SegPore and other baseline methods.

The online version of this article includes the following source data and figure supplement(s) for figure 3:

**Source data 1.** site_mod_rate_mES_WT_all_motifs.

**Source data 2.** site_mod_rate_mES_WT_selected_motifs.

**Figure supplement 1.** m6A k-mer statistics differences between the human HEK293T (training data) and the mouse mES_WT (test data).

**Figure supplement 1—source data 1.** xpore kmer statistics.

**Figure supplement 1—source data 2.** mes kmer statistics.

**Figure supplement 2.** Comparison of raw signal segmentation results between SegPore and Nanopolish on RNA002 data.

**Figure supplement 3.** Comparison of eventalign of SegPore and Nanopolish on consecutive 5-mers.

locations at the GGACT motif, SegPore exhibited more distinct peaks (*Figure 3B*), indicating reduced noise and potentially enabling more reliable modification detection.

We evaluated m6A predictions using two approaches: (1) SegPore's segmentation results were fed into m6Anet, referred to as SegPore+m6Anet, which works for all DRACH motifs and (2) direct m6A predictions from SegPore's Gaussian Mixture Model (GMM), which is limited to the six selected 5-mers shown in *Figure 3—figure supplement 1C* that exhibit clearly separable modified and unmodified components in the GMM (see Materials and methods for details).

In terms of m6A identification, SegPore performed strongly on the test data. Using miCLIP2 (*Körtel et al., 2021*) data as the ground truth, we calculated the area under the curve (AUC) for both the receiver operating characteristic (ROC) and precision-recall (PR) curves. SegPore+m6Anet achieved the best performance with an ROC AUC of 89.0% and a PR AUC of 43.8% (*Figure 3C*). For six selected m6A motifs, SegPore achieved an ROC AUC of 82.7% and a PR AUC of 38.7%, earning the third best performance compared with deep learning methods m6Anet and CHEUI (*Figure 3D*). It is noteworthy that SegPore's GMM for m6A estimation is a very simple model, utilizing far fewer parameters than DNN-based methods. Achieving a decent performance with such a simple model is a significant accomplishment. These results highlight SegPore's robust performance in m6A identification. For practical applications, we recommend taking the intersection of m6A sites predicted by SegPore and m6Anet to obtain high-confidence modification sites, while still benefiting from the interpretability provided by SegPore's predictions.

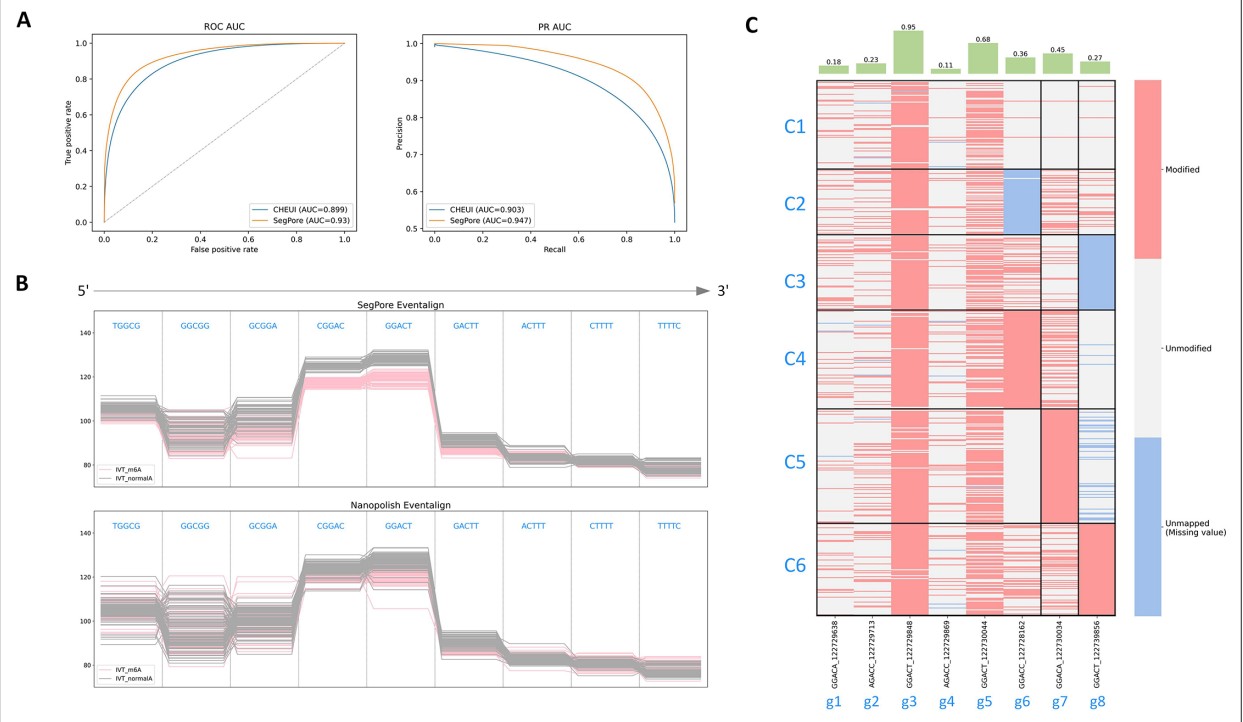

**Figure 4.** m6A identification at the single-molecule level. (**A**) Benchmark results for single-molecule m6A identification on IVT data. SegPore shows better performance compared to CHEUI in both PR-AUC and ROC-AUC. (**B**) Comparison of 'eventalign' results from SegPore and Nanopolish for five consecutive k-mers. Note that DRS is sequenced from 3' to 5', so the k-mers enter the pore from right to left. A total of 100 reads were randomly sampled from transcript locations A1 (positions 711–719) in both the IVT_normalA and IVT_m6A samples (SRA: SRP166020). Each line represents an individual read, and the y-axis shows the raw signal intensity in picoampere (pA). Pink lines represent the IVT_m6A sample, and gray lines represent the IVT_normalA sample. The k-mers 'GCGGA', 'CGGAC', 'GGACT', 'GACTT', and 'ACTTT' all contain N6-Methyladenosine (m6A) in the IVT_m6A sample. SegPore's results show clearer separation between m6A and adenosine, especially for 'CGGAC' and 'GGACT', compared to Nanopolish. (**C**) The upper panel shows the modification rate for selected genomic locations in the example gene ENSMUSG00000003153. The lower panel displays the modification states of all reads mapped to this gene. The black borders in the heatmap highlight the biclustering results, showing distinct modification patterns across different read clusters labeled C1 through C6.

The online version of this article includes the following source data and figure supplement(s) for figure 4:

**Source data 1.** ENSMUSG00000003153.10.info.

**Source data 2.** ENSMUSG00000003153.10.reads.

**Figure supplement 1.** Illustration of Nanopolish raw signal segmentation with eventalign results.

**Figure supplement 2.** Running time of SegPore on datasets of varying sizes.

## m6A identification at the single molecule level

SegPore naturally identifies m6A modifications at the single-molecule level, which is crucial for understanding the heterogeneity of RNA modifications across individual transcripts. We benchmarked SegPore against CHEUI using an in vitro transcription (IVT) dataset containing two samples—one transcribed with m6A and the other with adenine (*Jenjaroenpun et al., 2021*). This dataset provides clear ground truth for m6A modifications at the single-molecule level, with all adenosine positions replaced by m6A in the ivt_m6A sample, and all adenosines unmodified in the ivt_normalA sample. We used 60% of the data for training and 40% for testing, with both SegPore and CHEUI estimating the m6A modification probability at each adenosine site on each read. Based on these probabilities, we calculated the ROC-AUC and PR-AUC by varying the modification probability threshold.

As shown in *Figure 4A*, SegPore outperformed CHEUI on this benchmark dataset, achieving better performance in both PR-AUC (94.7% vs 90.3%) and ROC-AUC (93% vs 89.9%). These results clearly demonstrate SegPore's accuracy and robustness in detecting single-molecule m6A modifications.

Next, we demonstrate SegPore's interpretability through an example comparison of raw signal clips from both ivt_m6A and ivt_normalA samples (*Figure 4B*). The raw signal segments (means) are

aligned to a randomly selected m6A site as well as its neighboring sites in the reference sequence. Each line represents an individual read, with pink lines from the ivt_m6A sample and gray lines from the ivt_normalA sample. SegPore's segmentation clearly distinguishes between m6A and adenosine at the single-molecule level, while Nanopolish's segmentation shows less distinction.

Interestingly, we observed that the position of m6A within a 5-mer can affect the signal intensity. For instance, a clear difference between m6A and adenosine is evident when m6A occupies the fourth position in the 5-mer (e.g. 'CGGAC'), but this difference is less pronounced when m6A is in the second position (e.g. 'GACTT').

To illustrate the benefits of single-molecule m6A estimation, we present an example gene (ENSMUSG00000003153) from the mESC dataset used in site-level m6A identification. *Figure 4C* shows a heatmap of highly modified genomic locations (modification rate >0.1), where rows represent reads and columns represent genomic locations. Biclustering reveals six clusters of reads and three clusters of genomic locations.

The heatmap in *Figure 4C* suggests heterogeneity in m6A modification patterns across different reads of the same gene. Biclustering reveals that modifications at g6 are specific to cluster C4, g7 to cluster C5, and g8 to cluster C6, while the first five genomic locations (g1 to g5) show similar modification patterns across all reads. Additionally, high modification rates are observed at the 3rd and 5th positions across the majority of reads. These results suggest that m6A modification patterns can vary significantly even within a single gene. This observation highlights the complexity of RNA modification regulation and underscores the importance of single-molecule resolution for understanding RNA function at a finer scale.

## Discussion

One of the main computational challenges in direct RNA sequencing (DRS) lies in accurately segmenting the raw current signal. We developed a segmentation algorithm that models the jiggling property in the physical process of DRS, resulting in cleaner current signals for m6A identification at both the site and single-molecule levels. Our results demonstrate that SegPore's segmentation enables clear differentiation between m6A-modified and unmodified adenosines. We believe that the de-noised current signals will be beneficial for other downstream tasks, such as the estimation of m5C, pseudouridine, and other RNA modifications. Nevertheless, several open questions remain for future research. In SegPore, we assume a drastic change between two consecutive 5-mers, which may hold for 5-mers with large difference in their current baselines but may not hold for those with small difference. As with other RNA modification estimation methods, SegPore can be affected by misalignment errors, particularly when the baseline signals of adjacent k-mers are similar. These cases may lead to spurious bimodal signal distributions and require careful interpretation. Another key question concerns the physical interpretation of the derived base blocks. Ideally, one base block would correspond to a single 5-mer, but in practice, multiple base blocks often align with one 5-mer. We hypothesize that the HHMM may segment a 5-mer's current signal into multiple base blocks, where the 5-mer oscillates between different sub-states, each characterized by distinct baselines.

Currently, SegPore models only the modification state of the central nucleotide within the 5-mer. However, modifications at other positions may also affect the signal, as shown in *Figure 4B*. Therefore, introducing multiple states for a 5-mer to account for modifications at multiple positions within the same 5-mer could help to improve the performance of the model. This approach, however, would significantly increase model complexity—introducing two states per location results in $2^5 = 32$ modification states per 5-mer, a challenge for simple GMMs. It remains to be investigated how to model multiple modification states properly to improve the performance in RNA modification estimation. While the current two-state assumption simplifies the model, it has proven effective, as demonstrated by improved performance on m6A benchmarks at both site and single-molecule levels.

A key advantage of SegPore is its interpretability, which sets it apart from DNN-based methods. SegPore provides a clearer understanding of RNA modification predictions. A limitation of DNN-based approaches is that they often struggle to differentiate between m6A and adenosine signals, as their intensity levels are quite similar (*Figure 4B*, *Figure 3—figure supplement 3*). This poses a challenge when applying DNN-based methods to new datasets without short read sequencing-based ground truth. In such cases, it is difficult for researchers to confidently determine whether a predicted m6A modification is genuine. SegPore encodes current intensity levels for different 5-mers in a parameter

table, where unmodified and modified 5-mers are modeled using two Gaussian distributions. One can generally observe a clear difference in the intensity levels between 5-mers with an m6A and those with adenosine, which makes it easier for a researcher to interpret whether a predicted m6A site is genuine (see *Figure 4—figure supplement 1*). A challenge is the relatively small number of 5-mers that show significant changes in their modification states. To improve accuracy, larger training datasets and expanding the scope to 7-mers or 9-mers could help capture more context, potentially revealing more significant baseline changes.

The limited improvement of SegPore combined with m6Anet over Nanopolish+m6Anet in bulk in vivo analysis (*Figure 3*) may be explained by several factors: potential alignment inaccuracies due to pseudogenes or transcript isoforms, the complexity of in vivo datasets containing additional RNA modifications (e.g. m5C, m7G) affecting signal baselines, and the fact that m6Anet is specifically trained on events produced by Nanopolish rather than SegPore. Additionally, the lack of a modification-free control (in vitro transcribed) sample in the benchmark dataset makes it difficult to establish true baselines for each k-mer. Despite these limitations, SegPore demonstrates clear improvement in single-molecule m6A identification in IVT data (*Figure 4*), suggesting it is particularly well suited for in vitro transcription data analysis.

Although SegPore provides clear interpretability, there is potential to explore DNN-based models that can directly leverage SegPore's segmentation results. Currently, m6Anet computes features (e.g., mean and standard deviation) from raw signal segments, which are then fed into a neural network for m6A prediction. However, a more direct approach—where raw signal segments are used as input to a DNN—could allow for the extraction of more intricate features that may exist within the signal but are currently underexplored. These high-order features might capture subtle aspects of the raw signal, leading to improved m6A estimation. Since SegPore currently models only the mean and standard deviation, further work involving advanced DNNs could extend beyond this, uncovering finer patterns in the signal that traditional statistical models might miss.

Computation speed is also a concern when processing fast5 files. We addressed this by implementing a GPU-accelerated inference algorithm in SegPore, resulting in a 10- to 20-fold speedup compared to the CPU-based version. We believe that the GPU implementation will unlock the full potential of SegPore for a wider range of downstream tasks and larger datasets. More details are provided in Appendix 2. SegPore's running times on datasets of varying sizes, using a single NVIDIA DGX-1 V100 GPU and one CPU core, are provided in *Figure 4—figure supplement 2*.

In summary, we developed a novel software SegPore that considered the conformation changes of motor protein to segment raw current signal of Nanopore direct RNA sequencing. SegPore effectively masks out noise in the raw signal, leading to improved m6A identification at both site and single-molecule levels.

## Acknowledgements

We would like to thank Prof. Zhijie Tan from Wuhan University for a useful discussion about the molecule dynamics of Nanopore sequencing, Dr. Dan Zhang from Sichuan University for helpful tutorials about Nanopore analysis workflows. We also thank Prof. Luo Guanzheng for sharing the m6A benchmark baseline results. GC and LC acknowledge the computational resources provided by the Aalto Science-IT project. This work was supported by Research Council of Finland grants [335858, 358086 to GC and LC].

## Additional information

### Funding

| Funder | Grant reference number | Author |
|---|---|---|
| Research Council of Finland | 335858 | Guangzhao Cheng Lu Cheng |
| Research Council of Finland | 358086 | Guangzhao Cheng Lu Cheng |

| Funder | Grant reference number | Author |
|---|---|---|

The funders had no role in study design, data collection and interpretation, or the decision to submit the work for publication.

## Author contributions

Guangzhao Cheng, Data curation, Software, Formal analysis, Validation, Investigation, Visualization, Methodology, Writing – original draft, Writing – review and editing; Aki Vehtari, Resources, Supervision, Writing – original draft; Lu Cheng, Conceptualization, Resources, Software, Formal analysis, Supervision, Funding acquisition, Investigation, Methodology, Writing – original draft, Project administration, Writing – review and editing

## Author ORCIDs

Guangzhao Cheng ⓘ https://orcid.org/0009-0008-6160-3637
Aki Vehtari ⓘ https://orcid.org/0000-0003-2164-9469
Lu Cheng ⓘ https://orcid.org/0000-0002-6391-2360

Reviewer #1 (Public review): https://doi.org/10.7554/eLife.104618.4.sa1
Reviewer #2 (Public review): https://doi.org/10.7554/eLife.104618.4.sa2
Reviewer #3 (Public review): https://doi.org/10.7554/eLife.104618.4.sa3
Author response https://doi.org/10.7554/eLife.104618.4.sa4

# Additional files

## Supplementary files

MDAR checklist

## Data availability

The data utilized in this study are obtained from publicly available repositories. Details regarding the accession number and data processing can be found in Methods. The resulting data and the source code are hosted on GitHub (https://github.com/guangzhaocs/SegPore copy archived at *Cheng, 2025*).

The following previously published datasets were used:

| Author(s) | Year | Dataset title | Dataset URL | Database and Identifier |
|---|---|---|---|---|
| Pratanwanich PN | 2021 | Differential RNA modifications from dRNA-seq | https://www.ncbi.nlm.nih.gov/bioproject/?term=PRJEB40872 | NCBI BioProject, PRJEB40872 |
| Chen Y, Davidson NM, Wan YK | 2021 | SGNEx: A systematic benchmark of Nanopore long read RNA sequencing for transcript level analysis in human cell lines | https://www.ncbi.nlm.nih.gov/bioproject/?term=PRJEB44348 | NCBI BioProject, PRJEB44348 |
| Jenjaroenpun P, Wongsurawat T, Wadley TD, Wassenaar TM, Liu J, Dai Q, Wanchai V, Akel NS, Jamshidi-Parsian A, Franco AT, Boysen G, Jennings ML, Ussery DW, He C, Nookaew I | 2020 | Decoding Epitranscriptional Landscapes form Native RNA Sequences | https://trace.ncbi.nlm.nih.gov/Traces/?view=study&acc=SRP166020 | NCBI Sequence Read Archive, SRP166020 |

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

## Appendix 1

### Notation

| | |
|---|---|
| $\lvert \cdot \rvert$ | Take the length of the inside vector. |
| $Y$ | All reads in the input multi-fast5 file. $Y = (y_1, y_2, \cdots, y_i, \cdots)$. $y_i$ are the raw current measurements of $i$th read. Assuming there are $N$ reads, i.e. $\lvert Y \rvert = N$. |
| $y_i$ | The current signal of ith read. $y_i = (y_{i1}, y_{i2}, \cdots y_{ij}, \cdots y_{il_i})$ where $l_i$ is the length of $y_i$ i.e. $\lvert y_i \rvert = l_i$. |
| $y_i^{(k)}$ | The $k$th mapped current signal fragment of $i$th read $y_i$, given by eventalign using Nanopolish. |
| $R$ | Reference sequences for each read. $R = (r_1, r_2, \cdots r_i, \cdots)$. |
| $r_i$ | The reference sequence corresponding to $i$th read $y_i$, given by basecalling and minimap2. $r_i = (r_{i1}, r_{i2}, \cdots r_{ij}, \cdots, r_{im_i})$, where $r_{ij} \in \{A, C, G, T\}$ and $m_i$ is the length of $r_i$ i.e. $\lvert r_i \rvert = m_i$. |
| $r_i^{(k)}$ | The $k$th reference sequence fragment of $r_i$ corresponding to $y_i^{(k)}$. Note that $\lvert y_i^{(k)} \rvert$ is the larger than $\lvert r_i^{(k)} \rvert$ in general. |
| $s_i$ | The 5mer list corresponds to $r_i$. We have $s_i = (s_{i1}, s_{i2}, \cdots, s_{ij}, \cdots, s_{im_i})$, where $s_{ij} = r_{i,j-2} r_{i,j-1} r_{ij} r_{i,j+1} r_{i,j+2}$ and $s_{ij} \in \{AAAAA, AAAAC, AAAAG, AAAAT, \cdots, TTTTT\}$. Note that $\lvert s_i \rvert = \lvert r_i \rvert = m_i$. |
| $s_i^{(k)}$ | The $k$-mer list fragment corresponds to the $k$th reference sequence fragment $r_i^{(k)}$. $\lvert s_i^{(k)} \rvert = \lvert r_i^{(k)} \rvert$. |
| $Z$ | The estimated states of all reads given by the Hierarchical Hidden Markov Model (HHMM). $Z = (z_1, z_2, \cdots, z_i, \cdots)$. |
| $z_i$ | The hidden states for each current measure of $i$th read. $z_i = (z_{i1}, z_{i2}, \cdots, z_{ij}, \cdots, z_{il_i})$, where $z_{ij} \in \{0, 1, 2, 3, 4\}$ corresponds to $y_{ij}$ and $\lvert y_i \rvert = \lvert z_i \rvert = l_i$, where $z_i^{(k)} = 0$ means the transition state and $z_{ij} = 1, 2, 3, 4$ means base states. |
| $z_i^{(k)}$ | The $k$th fragment in $z_i$, which corresponds to $y_i^{(k)}$. Note that $\lvert z_i^{(k)} \rvert = \lvert y_i^{(k)} \rvert$. |
| $\mu_i$ | The mean values of base segments given by HHMM, which are estimated from $y_i$ and $z_i$. $\mu_i = (\mu_{i1}, \mu_{i2}, \cdots, \mu_{ij}, \cdots, \mu_{in_i})$, where $j = 1 \cdots n_i$ indexes the base segments of $i$th read. In general, we have $l_i > n_i > m_i$. |
| $\mu_i^{(k)}$ | The $k$th fragment of $\mu_i$ which corresponds to $y_i^{(k)}$ and $z_i^{(k)}$. $\mu_i^{(k)} = (\mu_{i1}^{(k)}, \mu_{i2}^{(k)}, \cdots, \mu_{ij}^{(k)}, \cdots)$. In general we have $\lvert y_i^{(k)} \rvert > \lvert \mu_i^{(k)} \rvert > \lvert s_i^{(k)} \rvert$. |
| $u_i^{(k)}$ | The modification state of each 5mer in $s_i^{(k)}$. $u_i^{(k)} = (u_{i1}^{(k)}, u_{i2}^{(k)}, \cdots, u_{ij}^{(k)}, \cdots)$, where $u_{ij}^{(k)} \in \{un, mod\}$ indicates if $s_{ij}^{(k)}$ is in unmodified or modified state. $\lvert u_i^{(k)} \rvert = \lvert s_i^{(k)} \rvert$. |
| $T^{ref}$ | The reference 5mer parameter table provided by Oxford Nanopore Technology (ONT). $T^{ref} = \{(\mu_{s,u}, \sigma_{s,u}), \forall s \in \{AAAAA, AAAAC, \cdots, TTTTT\}, u = \text{“un”}\}$. |
| $T^{kmer}$ | The parameter table for all 5mers. $T^{kmer} = \{(\mu_{s,u}, \sigma_{s,u}), \forall s \in \{AAAAA, AAAAC, \ldots, TTTTT\}, \forall u \in \{\text{“un”}, \text{“mod”}\}\}$. |

### Introduction

Given the parameter table $T^{kmer}$ for all $k$-mers, the overall workflow of SegPore consists of the following major steps: (1) Base-calling and mapping, (2) preprocessing of nanopore raw signal, (3) segmenting nanopore raw signal using hierarchical hidden Markov model (HHMM), and (4) aligning raw signal segments with corresponding $k$-mer list. The input of Segpore is the raw current signal of Nanopore direct RNA sequencing. The output of the Segpore workflow is the pairing of raw signal segments with corresponding $k$-mers with inferred state (unmodified state or modified state). Unless otherwise noted, the following analysis focuses on RNA002 data, using 5-mers rather than $k$-mers.

As illustrated in **Appendix 1—figure 1**, we denote the input FAST5/POD5 file as a set of raw current signal reads $Y = (y_1, y_2, \cdots, y_i, \cdots)$. We use $|Y|$ to denote the number of reads and $|y_i|$ to denote the number of measurements / data points of $y_i$. We then perform basecalling using Guppy and map basecalled sequence to the reference sequences ('Basecalling, mapping and preprocessing'), yielding a pairing between the $i$th read $y_i$ and its corresponding reference sequence $r_i$. We assume there are $k$ matched fragments $[(y_i^{(1)}, r_i^{(1)}), (y_i^{(2)}, r_i^{(2)}), \cdots, (y_i^{(k)}, r_i^{(k)})]$. For the $j$th fragment $y_i^{(j)}$, we use HHMM ('Hierarchical hidden Markov model') to estimate the hidden states for each measurement $z_i^{(j)} = (z_{i1}^{(j)}, z_{i2}^{(j)}, \cdots)$, where $z_{im}^{(j)} \in \{0, 1, 2, 3, 4\}$ and $|y_i^{(j)}| = |z_i^{(j)}|$. Based on $y_i^{(j)}$ and $z_i^{(j)}$, we can convert it into a list of low variation segments, with their means denoted by $\mu_i^{(j)} = (\mu_{i1}^{(j)}, \mu_{i2}^{(j)}, \cdots)$. As a result, the number of measurements in $y_i^{(j)}$ is larger than the number of segments in $\mu_i^{(j)}$, that is $|y_i^{(j)}| = |z_i^{(j)}| > |\mu_i^{(j)}|$. After that, we perform full or partial alignment between the raw signal segments $\mu_i^{(j)}$ and reference sequence $r_i^{(j)}$, which is converted into a list of 5-mers $s_i^{(j)} = (s_{i1}^{(j)}, s_{i2}^{(j)}, \cdots)$ and $|\cdot|$. Based on $\mu_i^{(j)}$ and $s_i^{(j)}$, we perform full or partial alignment ('Alignment of signal segments with reference sequence') to match each 5-mer with one or multiple current signal segments, as well as estimating its modification state $u_i^{(j)} = (u_{i1}^{(j)}, u_{i2}^{(j)}, \cdots)$, where $u_{im}^{(j)} \in \{\text{"un"}, \text{"mod"}\}$.

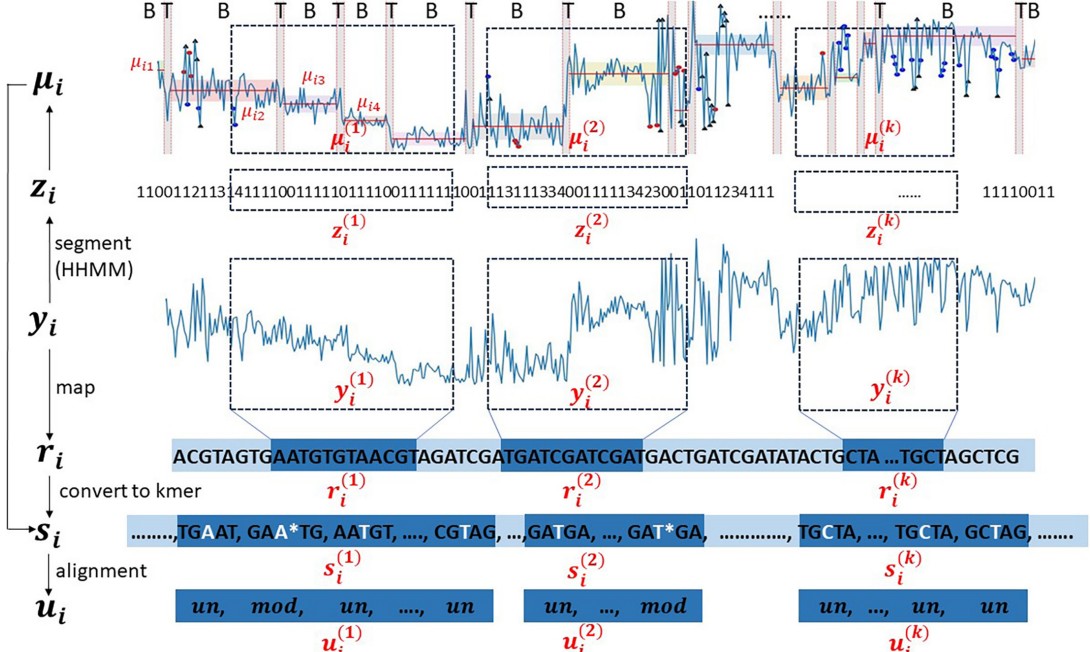

**Appendix 1—figure 1.** SegPore workflow with a fixed 5-mer parameter table $T^{kmer}$. The input is the raw current signal $y_i$ and reference sequence $r_i$ for $i$th read. The output is the paired current signal segments $\mu_i$, 5-mer list $s_i$ and modification states $u_i$.

However, $T^{kmer}$ is not fully known beforehand (**Appendix 1—table 1**), especially the parameters for the modification states of 5-mers. We have to estimate the parameters from the training data through an iterative process ('Estimation of k-mer parameters table'). After obtaining the estimated $T^{kmer}$, we derive the modification states from new test data using SegPore workflow with fixed $T^{kmer}$.

**Appendix 1—table 1.** Example of $T^{kmer}$ (1024×4).
'-' represents 'NA'.

| modle_kmer | un_mu | un_sigma | mod_mu | mod_sigma |
|---|---|---|---|---|
| AAAAA | 108.9 | 2.68 | - | - |
| AAAAC | 107.75 | 2.68 | - | - |
| AAAAG | 101.72 | 2.68 | - | - |
| AAAAT | 112.77 | 2.68 | - | - |

*Appendix 1—table 1 Continued on next page*

*Appendix 1—table 1 Continued*

| modle_kmer | un_mu | un_sigma | mod_mu | mod_sigma |
|---|---|---|---|---|
| AAACA | 99.38 | 2.41 | 95.28 | 2.44 |
| AAACC | 100 | 1.32 | 96.82 | 2.42 |
| AAACG | 101.01 | 3.45 | - | - |
| AAACT | 106.91 | 2.18 | 101.98 | 2.29 |
| AAAGA | 110.54 | 4.06 | - | - |
| AAAGC | 107.69 | 4.06 | - | - |
| AAAGG | 108.29 | 4.06 | - | - |
| AAAGT | 108.73 | 4.06 | - | - |
| AAATA | 114.11 | 3.11 | - | - |
| ... | ... | ... | ... | ... |
| TTTTT | 80.78 | 1.97 | - | - |

## Basecalling, mapping, and preprocessing

This section describes the details about the basecalling and mapping process, as well as raw data preprocessing for standardizing the raw signal of all reads.

First, we obtain the matching between raw current signals and reference sequences. The raw multi-fast5 reads (input file) were basecalled using Guppy (v6.0.1), which results in a FASTQ file. Then we align the yielded FASTQ file to the reference sequences (we use cDNA reference) using minimap2 (v2.24-r1122)(*Li, 2021*), which provides a bam file that matches the basecalled reads with the reference sequence. After that, we feed the raw fast5 file, basecalled FASTQ file, bam file, and reference sequence to Nanopolish (v0.13.2) to get (1) matching fragments between the raw current signal $y_i^{(k)}$ and the reference sequence $r_i^{(k)}$ (as well as $s_i^{(k)}$) and (2) the current signal fragment for poly(A) part of the mRNA molecule.

Then, we standardize the raw current signal based on the poly(A) tail. Due to technical reasons, there exist variations in the current signal of the same sequence fragment across different reads. It is necessary to perform a standardization. Since all reads have a poly(A) tail in Nanopore direct RNA sequencing, we could standardize the reads such that the mean and std of poly(A) part is the same for all reads. The standardization is given by

$$standardized\_signal = \frac{raw\_signal - \mu_{polyA}}{\sigma_{polyA}} * \sigma_{stand\,PolyA} + \mu_{stand\,PolyA} \tag{1}$$

where $\mu_{standPolyA} = 108.9$ and $\sigma_{standPolyA} = 1.67$ (RNA002). Note that $y_i$ refers to the standardized signal throughout this document.

## Hierarchical hidden Markov model

This section describes the hierarchical hidden Markov model (HHMM) to segment the raw current signal of a read. The input of HHMM is $y_i$ and the output is $z_i$. For notation simplicity, we use $y$ and $z$ to represent $y_i$ and $z_i$ in this section, respectively.

### Model assumptions

Here we have several assumptions regarding the raw signal. We first assume there is a sharp change of the baseline levels of the current signal between two consecutive 5-mers. This means there is a large change of current signal within a short period of time. If we fit a line to the corresponding signals, we expect to see a large slope. The second assumption is that the 5-mer resides in the Nanopore, oscillates forward and backward during a 5-mer event, while staying in the pore most of the time. Based on these two assumptions, we can divide the raw current signal into two states (hidden states of the outer HMM): the base states for 5-mer event and the transition state for the transition between two consecutive 5-mer events. We assume that the current signal always starts

with a base state, then switches to a transition state, after that switches back to a base state or transition state alternately, and in the end, the signal should end in a base state. For the base state, we use an inner HMM to model the oscillations, which has four hidden states: 'curr', 'prev', 'next', and 'noise'. For the transition state, we use a linear regression model to model the abrupt change of the current signal.

## Outer HMM

We define the following notations for the HHMM. As illustrated in **Appendix 1—figure 2**, we assume there are $N$ time points of the current signal, that is $\boldsymbol{y} = (y_1, y_2, \cdots, y_i, \cdots, y_N)$. We use $\boldsymbol{g} = (g_1, g_2, \cdots, g_i, \cdots, g_N)$, $g_i \in \{\text{"B", "T"}\}$ to indicate the hidden states of the outer HMM, where $B$ and $T$ stand for 'base' and 'transition', respectively. Similarly, we use $\boldsymbol{h} = (h_1, h_2, \cdots, h_i, \cdots, h_N)$, $h_i \in \{\text{"curr", "prev", "next", "noise"}\}$, respectively. We then integrate $\boldsymbol{g}$ and $\boldsymbol{h}$ into a single vector $\boldsymbol{z} = (z_1, z_2, \cdots, z_i, \cdots, z_N)$, where $z_i = \tilde{g}_i \cdot \tilde{h}_i$. Here $\tilde{g}_i(\text{"B"} \to 1, \text{"T"} \to 0)$ and $\tilde{h}_i(\text{'curr'} \to 1, \text{'prev'} \to 2, \text{'next'} \to 3, \text{'noise'} \to 4)$ are the numerical mappings of $g_i$ and $h_i$. Based on $\boldsymbol{g}$, we can derive that $\boldsymbol{y}$ is divided into $2K + 1$ blocks, which is defined by $\boldsymbol{c} = (c_1, c_2, \cdots, c_i, \cdots, c_{2K+1})$, $c_i \in \{\text{"B", "T"}\}$. Note that $c_i = \text{"B"}\ \forall i = 1, 3, \cdots, 2K + 1$ and $c_i = \text{"T"}\ \forall i = 2, 4, \cdots, 2K$. As a result, we could also represent $\boldsymbol{y} = (\boldsymbol{y}^{(1)}, \boldsymbol{y}^{(2)}, \cdots, \boldsymbol{y}^{(i)}, \cdots, \boldsymbol{y}^{(2K+1)})$, where $\boldsymbol{y}^{(i)}$ represent the $i$th block. The base blocks are given by $(\boldsymbol{y}^{(b_1)}, \boldsymbol{y}^{(b_2)}, \cdots, \boldsymbol{y}^{(b_i)}, \cdots, \boldsymbol{y}^{(b_{K+1})})$, where $b_i = 2i - 1$ and $i \in \{1, 2, \cdots, K + 1\}$. The transition blocks are given by $(\boldsymbol{y}^{(t_1)}, \boldsymbol{y}^{(t_2)}, \cdots, \boldsymbol{y}^{(t_i)}, \cdots, \boldsymbol{y}^{(t_K)})$, where $t_i = 2i$ and $i \in \{1, 2, \cdots, K\}$. Similarly, we have $\boldsymbol{g} = (\boldsymbol{g}^{(1)}, \boldsymbol{g}^{(2)}, \cdots, \boldsymbol{g}^{(i)}, \cdots, \boldsymbol{g}^{(2K+1)})$ and $\boldsymbol{h} = (\boldsymbol{h}^{(1)}, \boldsymbol{h}^{(2)}, \cdots, \boldsymbol{h}^{(i)}, \cdots, \boldsymbol{h}^{(2K+1)})$.

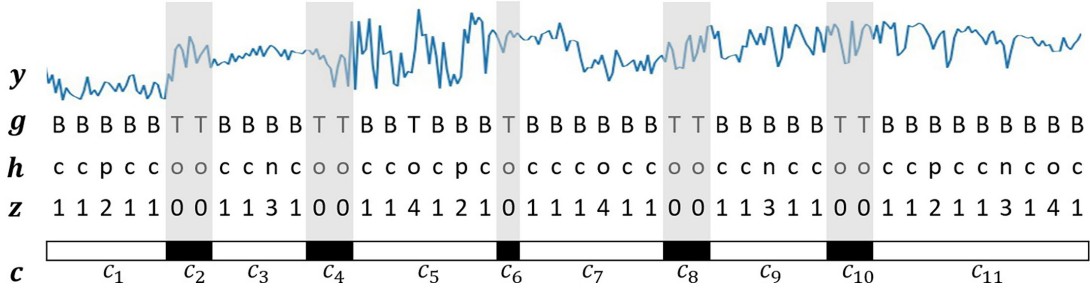

**Appendix 1—figure 2.** Example of HHMM where $g_i \in \{\text{"B", "T"}\}$, $h_i \in \{\text{"curr", "prev", "next", "noise"}\}$ and $c_i \in \{\text{"B", "T"}\}$. In this figure, 'c', 'p', 'n', and 'o' represent 'curr', 'prev', 'next', and 'noise' respectively in $\boldsymbol{h}$. $|\boldsymbol{y}| = |\boldsymbol{g}| = |\boldsymbol{h}|$ and $|\boldsymbol{c}| = 11$.

We use $T^{outer} = \{T_{BB}^{outer}, T_{BT}^{outer}, T_{TB}^{outer}, T_{TT}^{outer}\}$ to denote the transition probabilities between the base state and the transition state in the outer HMM. We use $T^{inner} = \{T_{h_i h_j}^{inner} \forall h_i, h_j \in \{\text{"curr", "prev", "next", "noise"}\}\}$ to denote the transition probabilities between four hidden states of the inner HMM. For base blocks, we have inner HMM parameters $\boldsymbol{\phi} = (\phi^{(b_1)}, \phi^{(b_2)}, \cdots, \phi^{(b_{K+1})})$ and $\boldsymbol{\sigma} = (\sigma^{(b_1)}, \sigma^{(b_2)}, \cdots, \sigma^{(b_{K+1})})$. For transition blocks, we have linear coefficients $\boldsymbol{\beta_0} = (\beta_0^{(t_1)}, \beta_0^{(t_2)}, \cdots, \beta_0^{(t_K)})$, $\boldsymbol{\beta_1} = (\beta_1^{(t_1)}, \beta_1^{(t_2)}, \cdots, \beta_1^{(t_K)})$ and noise std $\boldsymbol{\sigma_\epsilon} = (\sigma_\epsilon^{(t_1)}, \sigma_\epsilon^{(t_2)}, \cdots, \sigma_\epsilon^{(t_K)})$.

The joint likelihood of the HHMM is given by

$$
\begin{aligned}
p(\boldsymbol{y}, \boldsymbol{g}) &= p(\boldsymbol{y} \mid \boldsymbol{g}) p(\boldsymbol{g}) \\
&= p(\boldsymbol{y} \mid \boldsymbol{g}) \pi_{g_1}^{outer} \prod_{i=2}^{N} T_{g_{i-1} g_i}^{outer} \\
&= \prod_{i=1}^{N} p(y_i \mid g_i) \pi_{g_1}^{outer} \prod_{i=2}^{N} T_{g_{i-1} g_i}^{outer}
\end{aligned}
\tag{2}
$$

where $\pi_{g_1}^{outer} = 1$ is the initial probability of hidden state of the $y_1$ (always 'B' state). However, it is not possible to directly derive $p(y_i, h_i | g_i)$, since $y_i$ are not independent given $g_i$. Here we use different models for each block $c_k$ to calculate the joint probability $p(\boldsymbol{y}^{(k)} | \boldsymbol{g}^{(k)})$ to approximate $\prod_{i=1}^{N} p(y_i | g_i)$. Therefore, we have

$$
\begin{aligned}
\prod_{i=1}^{N} p(y_i \mid g_i) &= \prod_{j=0}^{2K+1} p\left(\boldsymbol{y}^{(j)} \mid c_j\right) \\
&= \prod_{j=0}^{K} p\left(\boldsymbol{y}^{(2j+1)} \mid c_{2j+1} = \text{"B"}\right) \prod_{j=1}^{K} p\left(\boldsymbol{y}^{(2j)} \mid c_{2j} = \text{"T"}\right)
\end{aligned}
\tag{3}
$$

The first part is the marginal likelihood for base blocks, while the second part is the marginal likelihood for transition blocks. The specific likelihood formulas are given in the following sections.

For the outer HMM, the transition probabilities $T^{outer}$ are pre-specified and the emission probabilities are obtained from the inner HMM ('Inner HMM for base state') and linear regression model ('Linear model for transition state'). The initial probability for the first hidden state $\pi_{g_1}^{outer}$ is set to 1, since the first state should always be "B".

## Inner HMM for base state

This section provides the derivations for calculating the likelihood of the $i$th base block $b_i$, where $i \in \{1, 2, \cdots, K+1\}$. We assume the data $\mathbf{y}^{(b_i)} = (y_1^{(b_i)}, y_2^{(b_i)}, \cdots, y_{N_{b_i}}^{(b_i)})$ is generated by an HMM with four states: 'curr', 'prev', 'next', and 'noise'. The descriptions of the states are given by *Appendix 1—table 2*, where parameters for the noise state are given and parameters for other states need to be estimated from the data.

**Appendix 1—table 2.** Hidden states of inner HMM for the $b_i$ th block.

| State | Event | Emission distribution | Parameter estimation |
|---|---|---|---|
| 'curr' | current 5-mer resides in pore | $N\big(\phi^{(b_i)}, (\sigma^{(b_i)})^2\big)$ | Yes |
| 'prev' | previous 5-mer resides in pore | $N\big(\phi^{(b_i-1)}, (\sigma^{(b_i-1)})^2\big)$ | Yes |
| 'next' | next 5-mer resides in pore | $N\big(\phi^{(b_i+1)}, (\sigma^{(b_i+1)})^2\big)$ | Yes |
| 'noise' | unknown noise | $Unif(lb, ub)$ | No |

The likelihood for the $k$th block is given by

$$p\big(\mathbf{y}^{(b_i)} \mid c_{b_i} = \text{"B"}\big) = \sum_{\mathbf{h}^{(b_i)}} p\big(\mathbf{y}^{(b_i)}, \mathbf{h}^{(b_i)} \mid c_{b_i} = \text{"B"}\big)$$

$$= \sum_{\mathbf{h}^{(b_i)}} p\big(\mathbf{y}^{(b_i)} \mid \phi, \sigma, \mathbf{h}^{(b_i)}, c_{b_i} = \text{"B"}\big) \, p\big(\mathbf{h}^{(b_i)} \mid c_{b_i} = \text{"B"}\big)$$

$$= \sum_{\mathbf{h}^{(b_i)}} \left[ \prod_{j=1}^{N_{b_i}} p\Big(y_j^{(b_i)} \mid h_j^{(b_i)}, \phi, \sigma\Big) \, \pi_{h_1^{(b_i)}}^{\text{inner}} \prod_{j=2}^{N_{b_i}} T_{h_{j-1}^{(b_i)} h_j^{(b_i)}}^{\text{inner}} \right]$$

$$= \sum_{\mathbf{h}^{(b_i)}} \Bigg[ \prod_{j=1}^{N_{b_i}} \bigg\{ N\Big(y_j^{(b_i)} \mid \phi^{(b_i)}, (\sigma^{(b_i)})^2\Big)^{I(h_j^{(b_i)} = \text{"curr"})} N\Big(y_j^{(b_i)} \mid \phi^{(b_i-1)}, (\sigma^{(b_i-1)})^2\Big)^{I(h_j^{(b_i)} = \text{"prev"})}$$

$$N\Big(y_j^{(b_i)} \mid \phi^{(b_i+1)}, (\sigma^{(b_i+1)})^2\Big)^{I(h_j^{(b_i)} = \text{"next"})} \text{Unif}\Big(y_j^{(b_i)} \mid lb, ub\Big)^{I(h_j^{(b_i)} = \text{"noise"})} \bigg\}$$

$$\pi_{h_1^{(b_i)}}^{\text{inner}} \prod_{j=2}^{N_{b_i}} T_{h_{j-1}^{(b_i)} h_j^{(b_i)}}^{\text{inner}} \Bigg] \tag{4}$$

where $\pi_{h_1^{(b_i)}}^{inner} = 0.25$ is the probability of the first hidden state and $I(\cdot)$ is the indicator function

For the inner HMM, the transition probabilities $T^{inner}$ are pre-specified and the initial probability for the first hidden state $\pi_{h_1^{(b_i)}}^{inner} = 0.25$ is set to 0.25. Other parameters (see *Appendix 1—table 2*) need to be estimated.

By comparing the emission probabilities of the outer HMM (*Equation 3*) and the marginal likelihood of the inner HMM (*Equation 4*), we have

$$p(\mathbf{y}^{(b_i)} = c_{b_i} = \text{"B"}) = \prod_{j=1}^{|\mathbf{y}^{(b_i)}|} p(y_j^{(b_i)} | g_j^{(b_i)}) \tag{5}$$

To approximate $p(y_j^{(b_i)} | g_j^{(b_i)})$, we assume all values of $p(y_j^{(b_i)} | g_j^{(b_i)})$ are the same for any given $j$, we have

$$p(\mathbf{y}^{(b_i)} \mid c_{b_i} = \text{"B"}) = \prod_{j=1}^{|\mathbf{y}^{(b_i)}|} p(y_j^{(b_i)} | g_j^{(b_i)}) = [\hat{p}(y_j^{(b_i)} | g_j^{(b_i)})]^{|\mathbf{y}^{(b_i)}|} \tag{6}$$

The approximated emission probability of each data point in the $i$th base block is given by

$$\hat{p}(y_j^{(b_i)} \mid g_j^{(b_i)}) = \sqrt[|\mathbf{y}^{(b_i)}|]{\hat{p}(\mathbf{y}^{(b_i)} \mid c_{b_i} = \text{"B"})}, \tag{7}$$

where $\hat{p}\left(\boldsymbol{y}^{(b_i)} \mid c_{b_i} = \text{"B"}\right)$ can be obtained by our inference algorithm.

Note that the hidden states of the inner HMM $\boldsymbol{h}$ can be provided as a by-product of our inference algorithm. We will use it to derive the final composite hidden state $z$ of the HHMM ('Parameter inference'), which is utilized by downstream alignment tasks ('Post-processing of alignment path').

## Linear model for transition state

This section provides the derivations for calculating the likelihood of the $i$th transition block $t_i$, where $i \in \{1, 2, \cdots, K\}$. We denote the input data by $\boldsymbol{y}^{(t_i)} = (y_1^{(t_i)}, y_2^{(t_i)}, \cdots, y_{N_{t_i}}^{(t_i)})$ and its corresponding index by $\boldsymbol{x}^{(t_i)} = (1, 2, \cdots, N_{t_i})$. We assume a linear model for data points in the transition state, given by

$$\boldsymbol{y}^{(t_i)} = \beta_1^{(t_i)} \boldsymbol{x}^{(t_i)} + \beta_0^{(t_i)} + \epsilon, \tag{8}$$

where $\beta_1^{(t_i)}$ is the slope, $\beta_0^{(t_i)}$ is the intercept and $\epsilon \sim N(0, (\sigma_\epsilon^{(t_i)})^2)$ is the Gaussian noise.

The likelihood of the linear model is given by

$$
\begin{aligned}
p\left(y^{(t_i)} \mid c_{t_i} = \text{"T"}\right) &= p\left(\boldsymbol{y}^{(t_i)} \mid \boldsymbol{x}^{(t_i)}, \beta_0^{(t_i)}, \beta_1^{(t_i)}, \sigma_\epsilon^{(t_i)}\right) \\
&= \prod_{j=1}^{N_{t_i}} p\left(y_j^{(t_i)} \mid x_j^{(t_i)}, \beta_0^{(t_i)}, \beta_1^{(t_i)}, \sigma_\epsilon^{(t_i)}\right) \\
&= \prod_{j=1}^{N_{t_i}} N\left(y_j^{(t_i)} - \beta_1^{(t_i)} x_j^{(t_i)} - \beta_0^{(t_i)} \mid 0, (\sigma_\epsilon^{(t_i)})^2\right)
\end{aligned}
\tag{9}
$$

Similar to the inner HMM, the approximated emission probability for each data point in $i$th transition block is given by

$$\hat{p}(y_j^{(t_i)} | g_j^{(t_i)}) = |y^{(t_i)}| \sqrt{\hat{p}(y^{(t_i)} | c_{t_i} = \text{"T"})}, \tag{10}$$

$\hat{p}(\boldsymbol{y}^{(b_i)} = c_{b_i} = \text{"T"})$ where can be obtained by our inference algorithm.

## Parameter Inference

Due to the complexity of GPU-accelerated inference algorithm, the details are provided in Appendix 2.

The inference algorithm will infer the following parameters: (1) the hidden states $\boldsymbol{g}$ of the outer HMM; (2) the hidden states of the inner HMM $\boldsymbol{h}$; (3) relevant parameters for emissions (*Appendix 1—table 2*) of each current signal fragment of the $2k+1$ th block (base block) $\boldsymbol{y}^{(2k+1)}$, where $k \in (0, 1, \cdots, K)$ and $c_{2k+1} = \text{"B"}$; and (4) the linear coefficients and noise std of the $2k$th block (transition block) $\boldsymbol{y}^{(2k)}$, where $k \in (1, 2, \cdots, K)$ and $c_{2k} = \text{"T"}$.

We then integrate $\boldsymbol{g}$ and $\boldsymbol{h}$ into a single vector $z = (z_1, z_2, \cdots, z_i, \cdots, z_N)$, where $z_i = \tilde{g}_i \cdot \tilde{h}_i$ and $\tilde{g}_i, \tilde{h}_i$ are the numerical mappings of $g_i, h_i$. $z$ is the final output of HHMM.

## Alignment of signal segments with reference sequence

This section describes the alignment algorithms for aligning an input fragment of signal segments $\boldsymbol{\mu}_i^{(k)}$ and the corresponding 5-mer list $\boldsymbol{s}_i^{(k)}$. After post-processing of the alignment path, we could get the modification states $\boldsymbol{u}_i^{(k)}$. For notation simplicity, we drop the indices and denote $\boldsymbol{\mu}_i^{(k)}$, $\boldsymbol{s}_i^{(k)}$ and $\boldsymbol{u}_i^{(k)}$ by $\boldsymbol{\mu}$, $\boldsymbol{s}$ and $\boldsymbol{u}$, respectively. In general, one 5-mer $s_j$ is aligned with $k$ signal segments $(\mu_i, \mu_{i+1}, \cdots, \mu_{i+k-1})$, where $k \geq 1$.

Here we discuss different scenarios of aligning $s_j$ with $\mu_i$. The first scenario (scenario 1) is that $\mu_i$ is generated by unmodified 5-mer $s_j$, for which we could compute the likelihood using the Normal distribution defined by $(\mu_{s_j}^{un}, \sigma_{s_j}^{un}) = T_{s_j, un}^{ref}$. The second scenario (scenario 2) is the $\mu_i$ is generated by modified 5-mer $s_j$, that is $N(\mu_{s_j}^{mod}, \sigma_{s_j}^{mod})$, where $(\mu_{s_j}^{mod}, \sigma_{s_j}^{mod}) = T_{s_j, mod}^{ref}$. The third scenario (scenario 3) is that $\mu_i$ is generated by an unknown nucleotide insertion event, whose probability is given by a pre-defined uniform distribution $p_{unif}$. The fourth scenario (scenario 4) is that the 5-mer $s_j$ represents a deletion event on the reference sequence, whose probability is given by the same uniform distribution $p_{unif}$. We define the scoring function as

$$f(s_j, \mu_i) = \begin{cases} max\left\{ log\left(N\left(\mu_i \mid \mu_{s_j}^{un}, (\sigma_{s_j}^{un})^2\right)\right), log\left(N\left(\mu_i \mid \mu_{s_j}^{mod}, (\sigma_{s_j}^{mod})^2\right)\right) \right\}, & (scenario\ 1, 2), \\ \\ log(p_{unif}) \quad \#s_j = " - "\ or\ \mu_i = 0.0 & (scenario\ 3, 4) \end{cases} \tag{11}$$

We can modify the classical global alignment algorithm to align $\boldsymbol{\mu}$ with $\boldsymbol{s}$. The log probability of aligning $s_j$ with $\mu_i$ could be used as the matching score (*Equation 11*) in the standard global alignment algorithm. Since one 5-mer $s_j$ could be aligned with multiple signal segments ($\mu_i, \mu_{i+1}, \cdots, \mu_{i+k-1}$), we need to modify the recursion such that $\mu_{i+1}$ could still be aligned with $s_j$ even if $\mu_i$ is already aligned with $s_j$. Therefore, we have the following recursions in our alignment algorithm (*Appendix 1—figure 3a*). Here we denote the scoring matrix by $M$ ($m \times n$) and the traceback matrix by $T$, where $|\boldsymbol{\mu}| = m - 1$ and $|\boldsymbol{s}| = n - 1$.

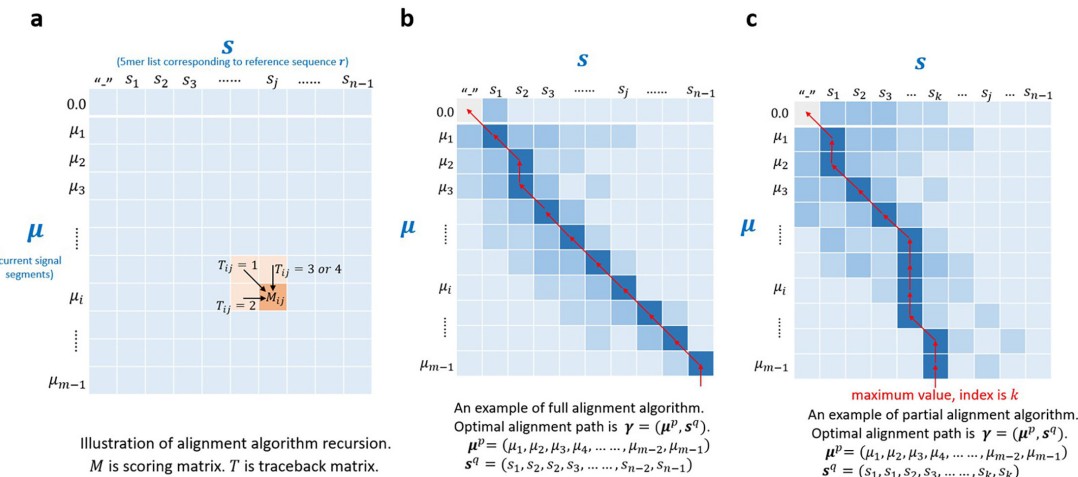

**Appendix 1—figure 3.** Alignment of signal segments with reference sequence.

$$M(i,j) = max \begin{cases} M(i-1, j-1) + f(\mu_i, s_j), & scenario\ 1, 2 & T(i,j) = 1 \\ \\ M(i, j-1) + f(0, s_j), & scenario\ 3 & T(i,j) = 2 \\ \\ M(i-1, j) + f(\mu_i, -), & scenario\ 4 & T(i,j) = 3 \\ \\ M(i-1, j) + f(\mu_i, s_j), & scenario\ 1, 2 & T(i,j) = 4 \end{cases} \tag{12}$$

where both $\mu_{i-1}$ and $\mu_i$ are aligned with $s_j$ when $T(i,j) = 4$. This is the only difference between our recursion and the recursion of the standard global alignment algorithm.

Based on the proposed recursion, we developed two alignment algorithms designed for different problems: the full alignment algorithm ('Full alignment algorithm') allows insertions and deletions in the reference sequence, while the partial alignment algorithm ('Partial alignment algorithm') does not allow any insertions or deletions in the partial reference sequence. The full alignment algorithm tries to align all the current signal segments to the whole reference sequence, while the partial alignment algorithm tries to align all the current signal segments to a consecutive sub-sequence of the 5-mer list from the beginning.

The output of the alignment algorithms is a matching path between the current signal segments $\boldsymbol{\mu}$ and the reference 5-mer list $\boldsymbol{s}$. We denote it by $\gamma = (\boldsymbol{\mu}^p, \boldsymbol{s}^q)$, where $\boldsymbol{\mu}^p = (\mu_{p_1}, \mu_{p_2}, \cdots, \mu_{p_N})$ and $\boldsymbol{s}^q = (s_{q_1}, s_{q_2}, \cdots, s_{q_N})$. The length of the alignment path is given by $|\gamma| = N$. $\boldsymbol{p} = (p_1, p_2, \cdots, p_N)$ are the indexes of the current signal segment $\boldsymbol{\mu}$ in the alignment path $\gamma$, while $\boldsymbol{q} = (q_1, q_2, \cdots, q_N)$ are the indexes of the 5-mer list $\boldsymbol{s}$. Note that there are duplicated values in $\boldsymbol{p}$ and $\boldsymbol{q}$ to allow one 5-mer to match with multiple current signal segments. *Appendix 1—figure 3b* provides an illustration of the alignment result.

To obtain the modification states $u$, we will first process the optimal alignment path $\gamma$ into $\hat{\gamma}$ ('Post-processing of alignment path'), from which we can derive the modification states $u$ ('Modification state estimation on single molecule level').

## Full alignment algorithm

The full alignment algorithm is used when the read may possess differences (insertions or deletions) compared with the reference sequence, which is the general case. The algorithm is described in Appendix 1–algorithm 1.

---

**Appendix 1–algorithm 1. Full alignment algorithm.**

---

Input: $\mu$, $s$, $T^{kmer}$
Output: Alignment path $\gamma$
```
// Add special symbol before μ and s
```
$\mu = \{0.0, \mu\}$, $s = \{"-", s\}$
```
// Get the length of and μ and s
```
$m = |\mu|$, $n = |s|$
```
 // Initialize the score matrix M and traceback matrix T
```
Initialize a $m \times n$ matrix $M$, with all elements set to 0.0
Initialize a $m \times n$ matrix $T$, with all elements set to 0
```
 // Initialize the first row of M and T
```
for $i \in \{1, 2, \ldots, n-1\}$ do
 $M(0, i) \leftarrow M(0, i - 1) + f(0.0, s_i)$
 $T(0, i) \leftarrow 2$
end
```
// Intialize the first column of M and T
```
for $i \in \{1, 2, \cdots, m - 1\}$ do
 $M(i, 0) \leftarrow M(i - 1, 0) + f(\mu_i, " - ")$
 $T(i, 0) \leftarrow 3$
end
```
// Loop for all matrix
```
for $i \in \{1, 2, \cdots, m - 1\}$ do
 for $j \in \{1, 2, \cdots, n - 1\}$ do
 $v_1 \leftarrow M(i - 1, j - 1) + f(\mu_i, s_j)$
 $v_2 \leftarrow M(i, j - 1) + f(0.0, s_j)$
 $v_3 \leftarrow M(i - 1, j) + f(\mu_i, " - ")$
 $v_4 \leftarrow M(i - 1, j) + f(\mu_i, s_j)$
 $M(i, j) \leftarrow max\{v_1, v_2, v_3, v_4\}$
 $T(i, j) \leftarrow argmax\{v_1, v_2, v_3, v_4\}$ # index of maximum
 end
end
```
// Traceback
```
Traceback from position $(m - 1, n - 1)$ to position $(0, 0)$ using $T$ to generate the optimal alignment path $\gamma$.

---

## Partial alignment algorithm

The partial alignment algorithm is used when the read must agree with the reference sequence (no insertions or deletions). If the sequencing data is generated by in vitro transcription from a single template, all reads should agree with the reference sequence. However, the read may only match the reference sequence from the start to a middle point due to the break of the RNA molecule in the reverse transcription.

The partial alignment algorithm is a modification of the full alignment algorithm by considering this special case. Since the alignment may not be full, we traceback from the maximum value in the last row of the scoring matrix $\mu$. This means all current signal segments must be aligned, while only the first part of the 5-mer list needs to be aligned. *Appendix 1—figure 3c* provides an illustration of the alignment result.

The partial alignment algorithm is described in Algorithm 1–algorithm 2. We can see that scenarios 3 and 4 have been removed from the algorithm compared with the full alignment algorithm, that is $T(i, j) = 2 \, or \, 3$. In other words, there should be no insertions or deletions in the 5-mer list $s$.

**Appendix 1—algorithm 2. Partial alignment algorithm.**

Input: $\mu$, $s$, $T^{kmer}$
Output: Alignment path $\gamma$
```
// Add special symbol before μ and s
```
$\mu = \{0.0, \mu\}$, $s = \{" - ", s\}$
```
// Get the length of μ and s
```
$m = |\mu|$, $n = |s|$
```
// Initialize the score matrix M and traceback matrix T
```
Initialize a $m \times n$ matrix $M$, with all elements set to 0.0
Initialize a $m \times n$ matrix $T$, with all elements set to 0
```
// Initialize the first row of M and T
```
for $i \in \{1, 2, \cdots, n - 1\}$ do
 $M(0, i) \leftarrow M(0, i - 1) + f(0.0, s_i)$
 $T(0, i) \leftarrow 2$
end
```
// Initialize the first column of M and T
```
for $i \in \{1, 2, \cdots, m - 1\}$ do
 $M(i, 0) \leftarrow M(i - 1, 0) + f(\mu_i, " - ")$
 $T(i, 0) \leftarrow 3$
end
```
// Loop for all matrix
```
for $i \in \{1, 2, \cdots, m - 1\}$ do
 for $j \in \{1, 2, \cdots, n - 1\}$ do
 $v_1 \leftarrow M(i - 1, j - 1) + f(\mu_i, s_j)$
 $v_4 \leftarrow M(i - 1, j) + f(\mu_i, s_j)$
 $M(i, j) \leftarrow max\{v_1, v_4\}$
 if $v_1 \geq v_4$ then
 $T(i, j) \leftarrow 1;$
 else
 $T(i, j) \leftarrow 4;$
 end
 end
end
```
// Traceback
```
Find the maximum value $M(m - 1, k)$ in the last row of the scoring matrix M.
Traceback from position $(m - 1, k)$ to position $(0, 0)$ using $T$ to generate the optimal
 alignment path $\gamma$.

## Post-processing of alignment path

In the alignment path $\gamma$, one $k$-mer might be aligned to noise (scenario 4) or one current signal segment may be aligned with '-' (scenario 3). Firstly, we remove such pairs from the alignment path. Then, we re-compute a weighted mean for each remaining $k$-mer, since one $k$-mer may be aligned to multiple current signal segments. Here, we combine these segments and calculate the weighted mean over corresponding data points with hidden state 1, that is $z_i = 1$.

Let us assume there are $m$ segments aligned to the same $k$-mer $s_i$, with $n_j$ denotes the number of data points assigned to 'curr' state ($z_i = 1$) in the $j$-th segment and $\mu_j$ denotes the mean of the $j$-th segment. The weighted mean, denoted by $\hat{\mu}_i$, corresponding to the $k$-mer $s_i$, is defined as follows:

$$\hat{\mu}_i = \sum_{j=1}^{m} \frac{n_j}{n_1 + n_2 + \cdots + n_m} \mu_j. \tag{13}$$

In the end, we obtain a one-to-one correspondence between each $k$-mer and its newly derived weighted mean, denoted by $\hat{\mu} = (\hat{\mu}_1, \hat{\mu}_2, \cdots)$. Note that the length of $\hat{\mu}$ and $s$ is the same, that is $|\hat{\mu}| = |s|$. We denote the post-processed alignment path by $\hat{\gamma} = (\hat{\mu}, s)$.

### Modification estimation
### Modification state estimation on single molecule level

We are interested in estimating the modification state for 5-mer after alignment. $T^{kmer}$ has provided the mean and std for each 5-mer, as well as its modifications. Note that $T^{kmer}$ may only contain modification parameters for only a subset of 5-mers. For those 5-mers without modification parameters, their mean and std are set to 'NA'. We could evaluate the probability density of a given 5-mer under the unmodified distribution or the modified distribution. By comparing the likelihoods, we could infer the modification state. Given the post-processed alignment path $\hat{\gamma} = (\hat{\mu}, s)$, we estimate the modification state of $j$th 5-mer $s_j$ by

$$u_j = \begin{cases} \text{"un"} & if\ \mu_{s_j}^{un} = \text{"NA"}\ or\ \sigma_{s_j}^{un} = \text{"NA"}, \\[2ex] \text{"un"} & if\ N\left(\hat{\mu}_j \mid \mu_{s_j}^{un}, (\sigma_{s_j}^{un})^2\right) \geq N\left(\hat{\mu}_j \mid \mu_{s_j}^{mod}, (\sigma_{s_j}^{mod})^2\right) \\[2ex] \text{"mod"} & if\ N\left(\hat{\mu}_j \mid \mu_{s_j}^{un}, (\sigma_{s_j}^{un})^2\right) < N\left(\hat{\mu}_j \mid \mu_{s_j}^{mod}, (\sigma_{s_j}^{mod})^2\right) \end{cases} \tag{14}$$

where $(\mu_{s_j}^{un}, \sigma_{s_j}^{un}) = T_{s_j,un}^{kmer}$ and $(\mu_{s_j}^{mod}, \sigma_{s_j}^{mod}) = T_{s_j,mod}^{kmer}$.

In the end, we get the modification state vector $\boldsymbol{u} = (u_1, u_2, \cdots, u_i, \cdots, u_{|\hat{\gamma}|})$ for each 5-mer in the post-processed alignment path $\hat{\gamma}$.

## Modification probability estimation on single molecule level

In addition to estimating the modification state, it is also crucial to determine the modification probability of each 5-mer on each read, which is essential for accurately benchmarking the performance at the single-molecule level.

Given the post-processed alignment path $\hat{\gamma} = (\hat{\boldsymbol{\mu}}, s)$, we estimate the modification probability of $j$th 5-mer $s_j$ by

$$p_j = \begin{cases} \text{"NA"} & if\ \mu_{s_j}^{un} = \text{"NA"}\ or\ \sigma_{s_j}^{un} = \text{"NA"} \\[2ex] \dfrac{p\left(\hat{\mu}_j \mid T_{s_j,mod}^{kmer}\right)}{p\left(\hat{\mu}_j \mid T_{s_j,mod}^{kmer}\right) + p\left(\hat{\mu}_j \mid T_{s_j,un}^{kmer}\right)}, & if\ \mu_{s_j}^{un} \neq \text{"NA"}\ and\ \sigma_{s_j}^{un} \neq \text{"NA"} \end{cases} \tag{15}$$

where $(\mu_{s_j}^{un}, \sigma_{s_j}^{un}) = T_{s_j,un}^{kmer}$ and $(\mu_{s_j}^{mod}, \sigma_{s_j}^{mod}) = T_{s_j,mod}^{kmer}$. Indeed, the $p_j$ is the posterior probability given the distribution of modified state:

$$p_j = p\left(T_{sj,mod}^{kmer} \mid \hat{\mu}_j\right) = \frac{p\left(\hat{\mu}_j \mid T_{sj,mod}^{kmer}\right) \cdot p\left(T_{sj,mod}^{kmer}\right)}{p\left(\hat{\mu}_j \mid T_{sj,mod}^{kmer}\right) \cdot p\left(T_{sj,mod}^{kmer}\right) + p\left(\hat{\mu}_j \mid T_{sj,un}^{kmer}\right) \cdot p\left(T_{sj,un}^{kmer}\right)} \tag{16}$$

where we assume that the prior probabilities are equal, $p(T_{sj,mod}^{kmer}) = p(T_{sj,un}^{kmer})$.

In the end, we get the modification probability vector $\boldsymbol{p} = (p_1, p_2, \cdots, p_i, \cdots, p_{|\hat{\gamma}|})$ for each 5-mer in the post-processed alignment path $\hat{\gamma}$.

## Modification rate estimation on site level

In real data analysis, we need to decide the modification rate of a genomic location, which quantifies the probability of a read mapped to this location to be in the modified state. For a genomic location, we denote the 5-mer from reads mapped to this location as $(s_1, s_2, \cdots, s_N)$. Note that all $s_i, i = 1, 2, \cdots, N$ are identical 5-mers on different reads. The modification state of these 5-mers on different reads is denoted by $(u_1, u_2, \cdots, u_N)$. The modification rate is given by

$$\eta = \frac{\sum_{i=1}^{N} u_i = \text{"mod"}}{N} \tag{17}$$

### Estimation of *k*-mer parameters table

This section provides the inference algorithms for 5-mer parameter table $\boldsymbol{T}^{kmer}$. This is necessary for training data, as we do not have accurate *prior* parameters for $\boldsymbol{T}^{kmer}$.

Given input data $\boldsymbol{Y} = (\boldsymbol{y_1}, \boldsymbol{y_2}, \cdots, \boldsymbol{y_i}, \cdots, \boldsymbol{y_N})$, we first perform base calling and mapping to get $\boldsymbol{S} = (\boldsymbol{s_1}, \boldsymbol{s_2}, \cdots, \boldsymbol{s_i}, \cdots, \boldsymbol{s_N})$ run HHMM for all reads such we get $\boldsymbol{M} = (\boldsymbol{\mu_1}, \boldsymbol{\mu_2}, \cdots, \boldsymbol{\mu_i}, \cdots, \boldsymbol{\mu_N})$. We assume there are $n_i$ fragments for the $i$th read $\boldsymbol{y_i} = (\boldsymbol{y_i}^{(1)}, \boldsymbol{y_i}^{(2)}, \cdots, \boldsymbol{y_i}^{(n_i)})$. Similarly, we have $\boldsymbol{s_i} = (\boldsymbol{s_i}^{(1)}, \boldsymbol{s_i}^{(2)}, \cdots, \boldsymbol{s_i}^{(n_i)})$ and $\boldsymbol{\mu_i} = (\boldsymbol{\mu_i}^{(1)}, \boldsymbol{\mu_i}^{(2)}, \cdots, \boldsymbol{\mu_i}^{(n_i)})$.

The inference algorithm (Appendix 1–algorithm 3) then estimate $\boldsymbol{T}^{kmer}$ given $\boldsymbol{S}$, $\boldsymbol{M}$ and $\boldsymbol{T}^{ref}$ (https://github.com/nanoporetech/kmer_models; *Brennen, 2023*). As illustrated in *Appendix 1—figure 4*, the inference algorithm infers the parameters iteratively. With a given $\boldsymbol{T}^{kmer}$, we can use

it to obtain the pairing between raw current signal segments and 5-mers, as well as modification states, through the alignment algorithm ('Alignment of signal segments with reference sequence'). Given this pairing, we extract all current signal segments for a specific 5-mer, which may correspond to different genomic locations and different reads. As there are two states for this 5-mer (unmodified and modified), we expect to observe two peaks in the density of the current signal segments. Therefore, we could fit a two-component Gaussian mixture model to the current signal segments to estimate relevant parameters of the given 5-mer in $T^{kmer}$.

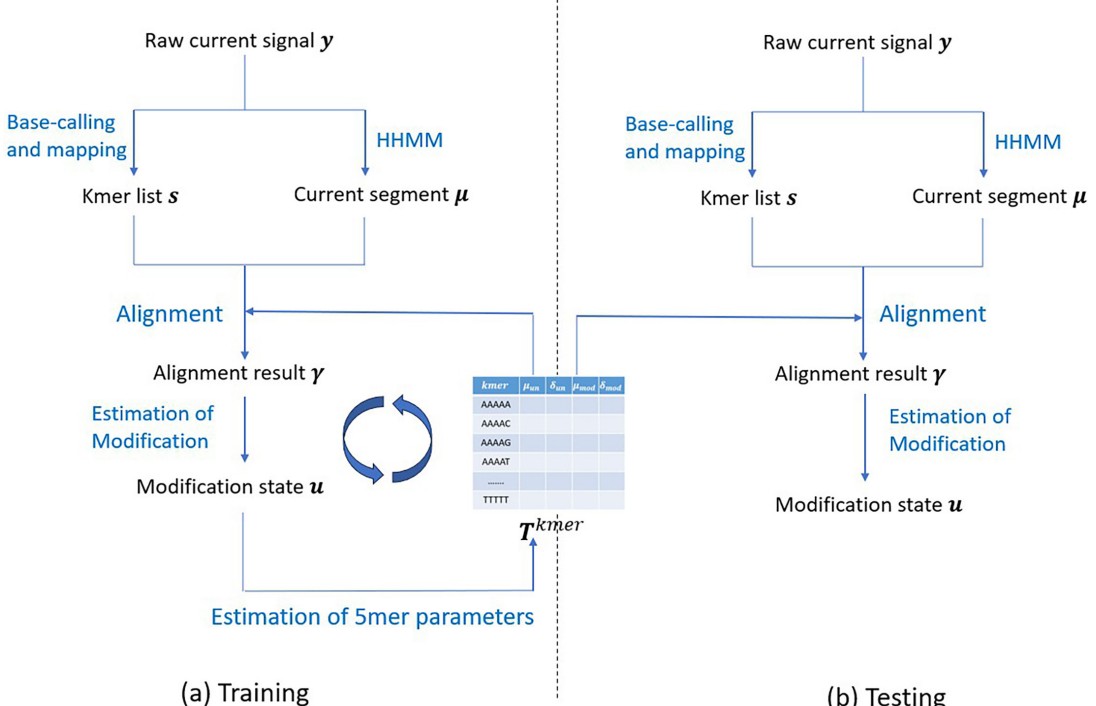

**Appendix 1—figure 4.** Estimation of 5-mer parameters. $T^{kmer}$ is initialized using $T^{ref}$. In training, our algorithm infers the parameters iteratively. After training, $T^{ref}$ is fixed in testing.

**Appendix 1–algorithm 3.** Inference algorithm of k-mer parameter table.

Input: $S$, $M$, $T^{ref}$
Output: $T^{kmer}$
 `// Initialize` $T^{kmer}$, $\forall s \in \{AAAAA, AAAAC, \cdots, TTTTT\}$, $\forall u \in \{"un", "mod"\}$
for $(\mu_{s,u}, \sigma_{s,u}) \in T^{kmer}$ do
 $(\mu_{s,u}, \sigma_{s,u}) = T^{ref}_{s,"un"}$
end
 `// iterate` $T$ `times to update` $T^{kmer}$
for $round = 1, 2 \cdots T$ do
 `// align all fragments of all reads`
 Perform full alignment algorithm (Appendix 1–algorithm 1) for $\mu_i^{(k)}$ and $s_i(k)$ to get the alignment path $\gamma_i^{(k)}$,
 where $i = 1 \cdots N$ and $k = 1 \cdots n_i$.

 Post-process (Post-processing of alignment path) the alignment path $\gamma_i^{(k)}$ into $\hat{\gamma}_i^{(k)} = [(\hat{\mu}_1^{(k)}, s_1^{(k)}), (\hat{\mu}_2^{(k)}, s_2^{(k)}), \cdots]$.

 Concatenate all $\hat{\gamma}_i^{(k)}$ and re-index into one vector $\hat{\mathbf{\Gamma}} = [(\hat{\mu}, s_1), (\hat{\mu}, s_2), \cdots]$.
 `// parameter estimation`
 for s0 $\in \{AAAAA, AAAAC, \cdots, TTTTT\}$ do

 Collect all $\hat{\mu}_j$ in $\hat{\mathbf{\Gamma}}$ where sj = s0 into one vector $x = \{\hat{\mu}_j, \forall s_j = s_0, \forall(\hat{\mu}_j, s_j)\} \in \hat{\mathbf{\Gamma}}$.
 Fit $x$ using a two-component Gaussian Mixture Model with the first component mean

 fixed to $\mu_{2_0}$, where $(\mu_{2_0}, \sigma_{2_0}) = T^{ref}_{s_0, "un"}$, and obtain the std of the first component $\sigma_1$,
 the mean and std of the second component $\mu_2$, $\sigma_2$.
 Update the parameters of 5-mer s0 in $T^{kmer}$ : $T^{kmer}_{s_0, "un"} \leftarrow (\mu_{s_0}, \sigma_1)$ and

 $T^{kmer}_{s_0, "mod"} \leftarrow (\mu_2, \sigma_2)$
 end
 Update $T^{kmer}$ manually based on several heuristic criteria.
end

In practice, the following heuristic criteria are used: (1) for a given 5-mer, perform GMM for each genomic location with the same 5-mer and only pool those sites with high modification rate (2) there must be two peaks in the density plot of a given 5-mer; (3) the mean of the two peaks are far apart; (4) the weight of second component (modified state) must be large, e.g. >0.1; (5) the weight of the first component (unmodified state) should be larger than the second component (modified state), as majority nucleotides are unmodified; (6) limit the size of std for both components in the GMM ($\sigma_1 < 5$, $\sigma_2 < 5$) to comply with the physical settings.

In the end, we will get an updated $T^{kmer}$ from the training data, which contains the modification parameters for the strongest 5-mers. As illustrated in **Appendix 1—table 1**, we do have parameters for the modification state for certain $k$-mers such as 'AAACA', 'AAACT' etc.

## Evaluation of segmentation and event alignment

The output of SegPore eventalign is the alignment path $\gamma = (\mu, s)$, which represents a one-to-one correspondence between the mean list $\mu = (\mu_1, \mu_2, ...)$ and the $k$-mer lists = $(s_1, s_2, ...)$. In addition, the corresponding standard deviation list $\sigma = (\sigma_1, \sigma_2, ...)$ is also obtained along the alignment path. **Appendix 1—table 3** is an example of the SegPore eventalign output, which illustrates one alignment path. The column ref_kmer corresponds to $\mathbf{s}$, the column mean corresponds to $\boldsymbol{\mu}$, and the column stdv corresponds to $\sigma$.

**Appendix 1—table 3.** SegPore eventalign output example.

| read_idx | read_name | contig | pos | strand | ref_kmer | model_kmer | mean | stdv | start_idx | end_idx | event_len |
|---|---|---|---|---|---|---|---|---|---|---|---|
| 80741 | 003655f4-ff77-409d-a29e-20dc778d1c47 | A1 | 57 | + | GGTGT | GGTGT | 79.339 | 2.414 | 47296 | 47321 | 18 |
| 80741 | 003655f4-ff77-409d-a29e-20dc778d1c47 | A1 | 58 | + | GTGTC | GTGTC | 89.809 | 3.579 | 47276 | 47292 | 14 |
| 80741 | 003655f4-ff77-409d-a29e-20dc778d1c47 | A1 | 59 | + | TGTCT | TGTCT | 117.017 | 6.287 | 47251 | 47272 | 18 |
| 80741 | 003655f4-ff77-409d-a29e-20dc778d1c47 | A1 | 60 | + | GTCTT | GTCTT | 76.448 | 1.038 | 47148 | 47247 | 87 |
| 80741 | 003655f4-ff77-409d-a29e-20dc778d1c47 | A1 | 61 | + | TCTTA | TCTTA | 76.036 | 1.044 | 47117 | 47144 | 25 |

*Appendix 1—table 3 Continued on next page*

*Appendix 1—table 3 Continued*

| read_idx | read_name | contig | pos | strand | ref_kmer | model_kmer | mean | stdv | start_idx | end_idx | event_len |
|---|---|---|---|---|---|---|---|---|---|---|---|
| 80741 | 003655f4-ff77-409d-a29e-20dc778d1c47 | A1 | 62 | + | CTTAG | CTTAG | 78.886 | 1.127 | 47092 | 47113 | 21 |
| 80741 | 003655f4-ff77-409d-a29e-20dc778d1c47 | A1 | 63 | + | TTAGT | TTAGT | 92.971 | 2.307 | 47007 | 47088 | 73 |
| 80741 | 003655f4-ff77-409d-a29e-20dc778d1c47 | A1 | 64 | + | TAGTG | TAGTG | 121.129 | 4.2 | 46882 | 46983 | 90 |
| 80741 | 003655f4-ff77-409d-a29e-20dc778d1c47 | A1 | 65 | + | AGTGT | AGTGT | 90.22 | 4.554 | 46822 | 46878 | 33 |
| 80741 | 003655f4-ff77-409d-a29e-20dc778d1c47 | A1 | 67 | + | TGTGC | TGTGC | 104.566 | 2.563 | 46798 | 46818 | 8 |
| 80741 | 003655f4-ff77-409d-a29e-20dc778d1c47 | A1 | 68 | + | GTGCT | GTGCT | 91.513 | 2.51 | 46778 | 46794 | 15 |
| 80741 | 003655f4-ff77-409d-a29e-20dc778d1c47 | A1 | 69 | + | TGCTT | TGCTT | 111.485 | 4.001 | 46754 | 46774 | 10 |

Given the eventalign results, we use two metrics to evaluate the performance of the segmentation and event alignment: (1) the average std $\hat{\sigma}$, and (2) the average log-likelihood $\hat{L}$. Assuming there are $N$ reads in total (each read has one path), and the $n$th read has $K_n$ events ($|\gamma_n| = K_n$). The average std $\hat{\sigma}$ is defined as

$$\hat{\sigma} = \frac{1}{N} \sum_{n=1}^{N} \left\{ \frac{1}{K_n} \sum_{k=1}^{K_n} \sigma_k \right\}, \tag{18}$$

and the average log-likelihood $\hat{L}$ is defined as

$$\hat{L} = \frac{1}{N} \sum_{n=1}^{N} \left\{ \frac{1}{K_n} \sum_{k=1}^{K_n} \log \mathcal{N}\left(\mu_k | \mu_{s_k}^{ref}, \sigma_{s_k}^{ref}\right) \right\} \tag{19}$$

where $(\mu_{s_k}^{ref}, \sigma_{s_k}^{ref}) = T_{s_k}^{ref}$.

As shown in **Appendix 1—figure 5**, the red line represents the event mean $\mu_k$ and the shaded area represents the std $\sigma_{s_k}^{est}$ for event $k$. A poorly segmented raw signal corresponding to an event will exhibit a large standard deviation. Therefore, a smaller $\hat{\sigma}$ indicates lower variations within the raw signal segment, signifying better performance.

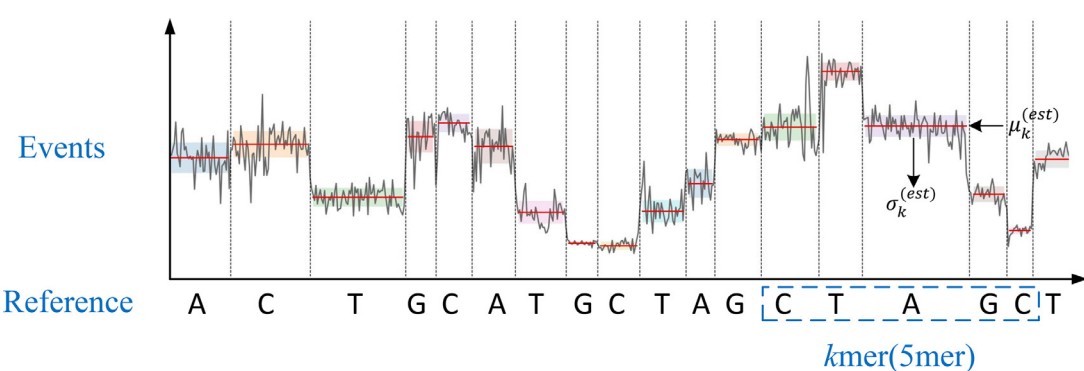

**Appendix 1—figure 5.** Illustration of segmentation and event alignment.

If an event is aligned to the correct reference $k$-mer, the mean $\mu_k$ will be close to the reference $\mu_{s_k}^{ref}$ and the log-likelihood $\hat{L}$ will be large. So higher $\hat{L}$ means more similar results to ONT's estimates and better performances.

## Appendix 2

### Introduction

We have introduced the hierarchical hidden Markov model (HHMM) in Appendix 1. The HHMM is a two-layer HMM, consisting of the outer HMM and the inner HMM. The outer HMM specifies the base blocks and transition blocks. The inner HMM specifies the base blocks. Linear models are used for the transition blocks. However, we have not provided the details for the parameter inference. This document aims to provide a comprehensive introduction to the inference algorithm.

As we know, the raw fast5 files are very large, which contain millions of reads. Each read has 20,000 measurements in the raw current signal. This poses a significant computational burden to the HHMM inference algorithm. Therefore, we used GPU to accelerate the inference algorithm.

We will first introduce the general inference algorithm, then describe the implementation details.

For simplicity, we assume the input is a raw current signal of a single read with $N$ measurements $\boldsymbol{y} = (y_1, y_2, \cdots, y_N)$. The output is the hidden states of the outer HMM $\boldsymbol{g} = (g_1, g_2, \cdots, g_N)$ and the hidden states of the inner HMM $\boldsymbol{h} = (h_1, h_2, \cdots, h_N)$, where $g_i \in \{\text{``}B\text{''}, \text{``}T\text{''}\}$ and $h_i \in \{\text{``}curr\text{''}, \text{``}prev\text{''}, \text{``}next\text{''}, \text{``}noise\text{''}\}$.

Given a hidden state configuration of the outer HMM $\boldsymbol{g}$, it partitions the raw current signal $\boldsymbol{y}$ into alternating base blocks and transition blocks $\boldsymbol{c} = (c_1, c_2, \cdots, c_i, \cdots, c_{2K+1})$, where $c_i \in \{\text{``}B\text{''}, \text{``}T\text{''}\}$ denotes the label of $i$th block. Note that $c_i = \text{``}B\text{''} \ \forall i = 1, 3, \cdots, 2K + 1$ and $c_i = \text{``}T\text{''} \ \forall i = 2, 4, \cdots, 2K$. Now we can denote $\boldsymbol{y} = (\boldsymbol{y^{(1)}}, \boldsymbol{y^{(2)}}, \cdots, \boldsymbol{y^{(2K+1)}})$.

The general idea of the inference algorithm is that we will calculate the likelihood for all possible configurations of the hidden states of the outer HMM $\boldsymbol{g}$ and choose the one with the largest likelihood as our final estimation. Due to the special constraints of $\boldsymbol{g}$, we can enumerate all possible configurations ('Enumeration of hidden states of outer HMM'). Here we denote the full configuration set of $\boldsymbol{g}$ by $\mathcal{G}$. The general inference algorithm is discussed in detail in 'General inference algorithm'.

The general inference algorithm provides the analytic forms for calculating the exact likelihood. However, a direct implementation of the algorithm (Python and C++) requires huge amounts of computation resources and time to handle a typical fast5 file. Therefore, we implement a GPU version of the inference algorithm ('Implementation details'), in which base blocks from different configurations are simultaneously inferred on different GPU cores.

### Enumeration of hidden states of outer HMM

This section describes how to get the full configuration set $\mathcal{G} = (\boldsymbol{g_1}, \boldsymbol{g_2}, \cdots, \boldsymbol{g_i}, \cdots, \boldsymbol{g_M})$ given the raw signal $\boldsymbol{y}$. Obviously, it is not possible to enumerate all hidden state $\boldsymbol{g_i} = (g_{i1}, g_{i2}, \cdots, g_{ij}, \cdots, g_{iN})$ (where $g_{ij} \in \{\text{``}B\text{''}, \text{``}T\text{''}\}$) because there are $2^N$ configurations and $N$ can be larger than 20,000. However, there exist special properties of the raw signal $\boldsymbol{y}$: (1) the signal should be in alternating base and transition blocks (2) the hidden state $\boldsymbol{g_i}$ does not change within a block (3) the average length of base blocks is larger than that of the transition blocks (4) the variation of base blocks is smaller than that of transition blocks. Based on these data properties, we could partition the raw signal $\boldsymbol{y}$ into atomic level base and transition blocks ('Atomic block initialization'). Multiple consecutive atomic level base and transition blocks can be merged into a single larger base block. As shown in *Appendix 2—figure 1a*, we partition the raw signal $\boldsymbol{y}$ into 15 atomic-level blocks $(B1, T1, B2, T2, \cdots, B7, T7, B8)$. We can merge $(B1, T1, B2)$ into a new base block or $(B5, T5, B6, T6, B7)$ into a new base block (*Appendix 2—figure 1b*).

Given a pre-defined partition $\boldsymbol{g_0}$ of raw signal $y$, we can derive the $2K + 1$ atomic blocks $(B_1, T_1, B_2, T2, \cdots, B_K, T_K, B_{K+1}))$. To facilitate the description of the enumeration algorithm, we define $seg(i, j) = (B_{i+1}, T_{i+1}, B_{i+2}, \cdots, T_{j-1}, B_j)$ to be the merged block of all blocks in $seg(i, j)$, where $0 \leq i < j \leq K + 1$. A path is then a list of non-overlapping and consecutive blocks. For example, the path corresponding to $\boldsymbol{g_i}$ in *Appendix 2—figure 1b* is given by $[seg(0, 2), T_2, B_3, T_3, B_4, T_4, seg(4, 7), T_7, B_8]$. To reduce the number of paths, we set the maximum number of base blocks in any merged block to $N_{max}$. The enumeration algorithm is given by Appendix 2–algorithm 1.

**Appendix 2–algorithm 1. Enumeration of hidden states of outer HMM.**

Input: $g_0$, $N_{max}$, $K$
Output: $\mathcal{G}$
Derive $(B_1, T_1, B_2, T_2, \cdots, B_K, T_K, B_K+1))$ from $g_0$
$\mathcal{G} \leftarrow \{\}$
**for** *path in recur_enum(0)* **do**
 Convert path to $g$
 Append $g$ to $\mathcal{G}$
**end**
 // Recursive function to enumerate all paths starting from position *st*
**Function** recur_enum(st):
 res = []
 **if** *st = K* **then**
 **return** $[B_{K+1}]$
 **end**
 // "[" is inclusive and ")" is exclusive
 **for** *i ∈ [st, K + 1)* **do**
 **if** $i − st > N_{max}$ **then**
 **return** res
 **end**
 tmp_seg = seg(st, i)
 all_paths_from_i = recur_enum(i)
 **for** *path in all_ paths_ from_i* **do**
 tmp_path = concatenate tmp_seg with path
 Append tmp_path to res
 **end**
 **end**
 **return** res;

## General inference algorithm

Given all possible hidden state configurations of the outer HMM $\mathcal{G}$, we want to calculate the likelihood for all $g \in \mathcal{G}$. In order to calculate the likelihood, we need to specify the parameters for the emission distributions.

Given $g$, we can derive that $y$ is divided into $2K + 1$ alternating base and transition blocks. We denote all blocks by $y = (y^{(1)}, y^{(2)}, \cdots, y^{(i)}, \cdots, y^{(2K+1)})$, where $y^{(i)}$ represent the $i$th block. The base blocks are given by $(y^{(b_1)}, y^{(b_2)}, \cdots, y^{(b_i)}, \cdots, y^{(b_{K+1})})$, where $b_i = 2i − 1$ and $i \in \{1, 2, \cdots, K + 1\}$. The transition blocks are given by $(y^{(t_1)}, y^{(t_2)}, \cdots, y^{(t_i)}, \cdots, y^{(t_K)})$, where $t_i = 2i$ and $i \in \{1, 2, \cdots, K\}$.

We pre-specify the transition probabilities $T^{outer} = \{T_{BB}^{outer}, T_{BT}^{outer}, T_{TB}^{outer}, T_{TT}^{outer}\}$ between the base state and the transition state in the outer HMM, as shown in *Appendix 2—table 1*. The pre-specification has two considerations: (1) the probability of a transition from B to T after 10 consecutive B states is 0.1; (2) the probability of a transition from T to B after 10 consecutive T states is larger than 0.9; (3) to reduce computation load in the parameter inference. We pre-specify the initial probability of the first hidden state to 1, since $g$ always starts with a 'B' state.

**Appendix 2—table 1.** Transition parameters of the outer HMM ($T^{outer}$).

|  | 'B' | 'T' |
| --- | --- | --- |
| 'B' | 0.99 | 0.01 |
| 'T' | 0.10 | 0.90 |

For inner HMM, we pre-specify the transition probabilities $T^{inner}$ shown in *Appendix 2—table 2*, which is based on the following consideration (1) large proportion of the data should come from the 'curr' state, that is it is easy to transit from other states to 'curr' state; (2) it is unlikely to transit from 'prev' to 'next', and vice versa; (3) it is unlikely to transit from other states to the 'noise' state. We pre-specify the initial probability of the first hidden state to 0.25, which gives equal probabilities for all four states. We pre-specify the parameters of the 'noise' state ($Unif(lb, ub)$) to $lb = 50$ and $ub = 130$, which are estimated from the current signals across different datasets to represent a reasonable range.

*Appendix 2—table 2 Continued on next page*

**Appendix 2—table 2.** Transition parameters of the inner HMM ($T^{inner}$).

|  | 'curr' | 'prev' | 'next' | 'noise' |
|---|---|---|---|---|
| 'curr' | 0.925 | 0.025 | 0.025 | 0.025 |
| 'prev' | 0.300 | 0.500 | 0.100 | 0.100 |
| 'next' | 0.300 | 0.100 | 0.500 | 0.100 |
| 'noise' | 0.300 | 0.100 | 0.100 | 0.500 |

The remaining parameters, mainly associated with emission distributions, need to be estimated. For base blocks, we have inner HMM parameters $\phi = (\phi^{(b_1)}, \phi^{(b_2)}, \cdots, \phi^{(b_{K+1})})$ and $\sigma = (\sigma^{(b_1)}, \sigma^{(b_2)}, \cdots, \sigma^{(b_{K+1})})$. For transition blocks, we have linear coefficients $\beta_0 = (\beta_0^{(t_1)}, \beta_0^{(t_2)}, \cdots, \beta_0^{(t_K)})$, $\beta_1 = (\beta_1^{(t_1)}, \beta_1^{(t_2)}, \cdots, \beta_1^{(t_K)})$ and noise std $\sigma_\epsilon = (\sigma_\epsilon^{(t_1)}, \sigma_\epsilon^{(t_2)}, \cdots, \sigma_\epsilon^{(t_K)})$.

The general idea of the inference algorithm is to calculate the joint likelihood of the outer HMM given for any $g$. However, it is not possible to calculate the emission probability for each measurement as there exist dependencies with a base or transition block. Here we first calculate the joint likelihood for a base block (inner HMM) or a transition block (linear model), then average the joint likelihood to each measurement to use it as substitute for the emission probability of each measurement in the outer HMM. Here we use $\omega = (\omega_1, \omega_2, \cdots, \omega_N)$ to denote the approximated emission probability of each measurement. Note that $\omega_i$ is derived either based on the inner HMM or the linear model, and $|\omega| = |y| = |g| = |h|$.

The general inference algorithm is given by Appendix 2–algorithm 2.

---

**Appendix 2–algorithm 2. General inference algorithm for HHMM.**

---

**Input:** $y$, $\mathcal{G}$, $T^{outer}$, $\pi^{outer}$, $T^{inner}$, $\pi^{inner}$,
**Output:** best path $\tilde{g}$ of outer HMM, best path $\tilde{h}$ of inner HMM, $\phi$, $\sigma$, $\beta_0$, $\beta_1$, $\sigma_\epsilon$
`// Initialization`
set all elements of $\tilde{\omega}, \tilde{g}$ and $\tilde{h}$ to 0, and $|\tilde{g}| = |\tilde{h}| = |y|$
$\mathcal{H} \leftarrow \{\}$
$\Gamma \leftarrow \{\}$
`// Calculate likelihood for each g`
for $g \in \mathcal{G}$ do

$\quad h \rightarrow \tilde{h}$
Divide $y$ into base and transition blocks ($y^{(1)}, y^{(2)}, \cdots, y^{(2K+1)}$) based on $g$
`// transition state parameter estimation`

Estimate $\hat{\omega}^{(t_i)}$, $\hat{\beta}_0^{(t_i)}$, $\hat{\beta}_1^{(t_i)}, \sigma_\epsilon^{(t_i)}$ using linear model (Appendix 2–algorithm 3), where $t_i = 2i$ and $i \in \{1, 2, \cdots, K\}$
Update relevant part of $\omega$ using $\hat{\omega}^{(t_i)}$.
`// base state parameter estimation`
Initialize $\phi^{(b_i)}$ and $\sigma^{(b_i)}$ use the empirical mean and std of $y^{(b_i)}$, where $b_i = 2i - 1$ and
$\quad i \in \{1, 2, \cdots, K+1\}$
$\phi^{new} \leftarrow \phi$
$\sigma^{new} \leftarrow \sigma$
for $round = 1, 2, \cdots, R$ do
$\quad$ for $i \in (1, 2, \cdots, K+1)$ do
$\quad\quad b_i \leftarrow 2i - 1$
$\quad\quad$ `// inner HMM parameter estimation`
$\quad\quad$ Estimate $\hat{\omega}^{(b_i)}$, $\hat{h}^{(b_i)}$, $\hat{\phi}^{(b_i)}$, $\hat{\sigma}^{(b_i)}$ using Forward-backward algorithm (Appendix 2–algorithm 4) given
$\quad\quad y^{(b_i)}$, $\phi^{(b_i)}$, $\sigma^{(b_i)}$, $\phi^{(b_{i-1})}$, $\sigma^{(b_{i-1})}$, $\phi^{(b_{i+1})}$, $\sigma^{(b_{i+1})}$, $T^{inner}$, $\pi^{inner}$
$\quad\quad$ Update corresponding elements of $\phi^{new}$ and $\sigma^{new}$ using $\hat{\phi}^{(b_i)}, \hat{\sigma}^{(b_i)}$
$\quad\quad$ Update relevant part of $h$ and $\omega$ using $\hat{h}^{(b_i)}$ and $\hat{\omega}^{(b_i)}$
$\quad$ end
$\quad \phi \leftarrow \phi^{new}$
$\quad \sigma \leftarrow \sigma^{new}$
end
Calculate the joint likelihood of the outer HMM using $g$, $\omega$, $T^{outer}$, $\pi^{outer}$

$$\gamma = \pi_{g_1}^{outer} \prod_{i=2}^{N} T_{g_{i-1}g_i}^{outer} \prod_{j=1}^{N} \omega_j$$

Append $h$ and $\gamma$ to $\mathcal{H}$ and $\Gamma$, respectively.
end
Choose the largest likelihood from $\Gamma$ and retrieve the corresponding $g$ and $h$ from $\mathcal{G}$ and $\mathcal{H}$, respectively.
$\tilde{g} \leftarrow g$
$\tilde{h} \leftarrow h$

---

## Implementation details

### Atomic block initialization

As mentioned in 'Enumeration of hidden states of outer HMM', we need to partition the raw signal $y$ into $2K + 1$ atomic blocks

$(B_1, T_1, B_2, T2, \cdots, B_K, T_K, B_{K+1})$. If we can identify all atomic transition blocks $(T_1, T_2, \cdots, T_K)$, then the atomic base blocks are automatically determined.

We assume atomic transition blocks $(T_1, T_2, \cdots, T_K)$ have higher variations than atomic base blocks. If we fit a line to a transition block, the absolute value of the slope of the line should be large. Based on this idea, we can obtain the atomic transition blocks as follows:

1. For each measurement $i$, fit another line to $(y_{i-1}, y_i, y_{i+1})$ and obtain the slope $\beta_1$, fit a line to $(y_{i-2}, y_{i-1}, y_i, y_{i+1}, y_{i+2})$ and obtain the slope $\beta_2$, use $\beta_i = |0.5 * (\beta_1 + \beta_2)|$ as the absolute slope for $y_i$.
2. Smooth the obtain slopes $\boldsymbol{\beta} = (\beta_1, \beta_2, \cdots, \beta_N)$ by taking the mean of a sliding window of size. Now we get the smoothed $\bar{\boldsymbol{\beta}}$
3. Find peaks from $\bar{\boldsymbol{\beta}}$ using the find_peaks() function in the Python Scipy.signal package. Here, set the minimum distance of detected peaks to 10. Now we get a list of peak positions $\boldsymbol{\theta} = (\theta_1, \theta_2, \cdots, \theta_K)$.
4. Expand $\theta_k$ to the atomic transition block $T_k$. We fit a 3-segment spline to

   $(y_{\theta_k-8}, \cdots, y_{\theta_k-l}, \cdots, y_{\theta_k}, \cdots, y_{\theta_k+r}, \cdots, y_{\theta_k+8})$, where $0 < l, r < 8$. The slopes of the first segment $(y_{\theta_k-8}, \cdots, y_{\theta_k-l-1})$ and the third segment $(y_{\theta_k+r+1}, \cdots, y_{\theta_k+8})$ are set to 0, while a standard linear model is fit to the second segment $(y_{\theta_k-l}, \cdots, y_{\theta_k}, \cdots, y_{\theta_k+r})$. In the fitting, the slope of the second segment should be larger than 5.0. After the fitting of the spline, we take the residuals and calculate the joint likelihood assuming each residual follows a Gaussian distribution $N(0, 4)$. By enumerating all $l, r$, we pick the $\hat{l}, \hat{r}$ with the largest joint likelihood. Therefore, the final $k$th transition block $T_k$ corresponding to the interval $[\theta_k - l, \theta_k + r]$.

### Emission probabilities of linear model

For the $i$th transition block, the measurements are given by $\boldsymbol{y}^{(t_i)} = (y_1^{(t_i)}, y_2^{(t_i)}, \cdots, y_{N_{t_i}}^{(t_i)})$, where $t_i = 2i$ and $i \in 1, 2, \cdots, K$. We denote the corresponding index of $\boldsymbol{y}^{(t_i)}$ by $\boldsymbol{x}^{(t_i)} = (1, 2, \cdots, N_{t_i})$. Here we want to get the emission probabilities for each measurement $\hat{\boldsymbol{\omega}}^{(t_i)}$, which can be used by the outer HMM. Detailed steps are provided in the following Appendix 2–algorithm 3.

---

**Appendix 2–algorithm 3. Emission probabilities of linear model.**

---

Input: $\boldsymbol{y}^{(t_i)}$, $\boldsymbol{x}^{(t_i)}$

Output: $\hat{\boldsymbol{\omega}}^{(t_i)}$, $\beta_0^{(t_i)}$, $\beta_1^{(t_i)}$, $\sigma_\epsilon^{(t_i)}$

initialize $\hat{\boldsymbol{\omega}}^{(t_i)}$ to a $1 \times N_{t_i}$ all zero vector.

Fit a linear model of $\boldsymbol{y}^{(ti)}$ versus $\boldsymbol{x}^{(ti)}$, obtain the maximum likelihood estimates $\beta_0^{(t_i)}, \beta_1^{(t_i)}$ and $\sigma_\epsilon^{(t_i)}$

Calculate the maximum log joint likelihood $\log p(\boldsymbol{y}^{(t_i)}|\boldsymbol{x}^{(t_i)}, \beta_0^{(t_i)}, \beta_1^{(t_i)}, \sigma_\epsilon^{(t_i)}) =$

$\log N(\boldsymbol{y}^{(t_i)} - \beta_1^{(t_i)} \boldsymbol{x}^{(t_i)} - \beta_0^{(t_i)} \mid 0, (\sigma_\epsilon^{(t_i)})^2)$

Set each element of $\hat{\boldsymbol{\omega}}^{t_i}$ to $\exp\left(\frac{1}{N_{t_i}} \log p\left(\boldsymbol{y}^{(t_i)} \mid \boldsymbol{x}^{(t_i)}, \beta_0^{(t_i)}, \beta_1^{(t_i)}, \sigma_\epsilon^{(t_i)}\right)\right)$

---

### Emission probabilities of inner HMM

This section describes how to obtain the approximated emission probabilities of the outer HMM for a the $i$th base block $\boldsymbol{y}^{(b_i)} = (y_1^{(b_i)}, y_2^{(b_i)}, \cdots, y_{N_{b_i}}^{(b_i)})$, which is modeled using an inner HMM. Here $b_i = 2i - 1$ and $i \in \{1, 2, \cdots, K\}$.

For the inner HMM, we list all the parameters in *Appendix 2—table 3*.

**Appendix 2—table 3.** Parameters of the inner HMM for the $b_i$ th block.

| Parameter | Fixed parameter | Description |
|---|---|---|
| $T^{inner}$ | Yes | Transition probabilities between hidden states ('curr', 'prev', 'next', 'noise'), specified in **Appendix 2—table 2** |
| $\pi_k^{inner}$ | Yes | probability of the first hidden state, $\pi_k^{inner} = 0.25$, where $k \in \{1, 2, 3, 4\}$ |
| $lb$ | Yes | lower bound of uniform distribution, $lb = 50$ |
| $ub$ | Yes | upper bound of uniform distribution, $ub = 130$ |
| $\phi^{(b_i)}$ | No | Gaussian mean of $i$ th base block |
| $\sigma^{(b_i)}$ | No | Gaussian std of $i$ th base block |
| $\phi^{(b_{i-1})}$ | No | Gaussian mean of $i - 1$ th base block |
| $\sigma^{(b_{i-1})}$ | No | Gaussian std of $i - 1$ th base block |
| $\phi^{(b_{i+1})}$ | No | Gaussian mean of $i + 1$ th base block |
| $\sigma^{(b_{i+1})}$ | No | Gaussian std of $i + 1$ th base block |

We want to obtain the approximated emission probabilities $\hat{\omega}^{(b_i)}$, hidden states $\hat{h}^{(b_i)}$, estimated parameters $\hat{\phi}^{(b_i)}$ and $\hat{\sigma}^{(b_i)}$ for the $i$th base block $y^{(b_i)}$. To achieve this, we follow the Forward-backward algorithm in **Bishop and Nasrabadi, 2006** (Chapter 13.2). The general idea of the Forward-backward algorithm is to use a $Q$-function to approximate the marginal likelihood, then use an expectation-maximization (EM) algorithm for the parameter estimation.

We introduce the following notations for the Forward-backward algorithm. We use $Z = (z_1, z_2, \cdots, z_{N_{b_i}})$ to denote the hidden states, where $Z$ is a $N_{b_i} \times 4$ matrix, $z_j = (z_{j1}, z_{j2}, z_{j3}, z_{j4})$ and $z_{jk} \in \{0, 1\}$. If $z_{jk}$ is 1, then it means $h_j$ corresponds to the $k$th hidden state. We use $\alpha$ $(N_{b_i} \times 4)$ to denote the forward probabilities and $\beta$ $(N_{b_i} \times 4)$ to denote the backward probabilities. $\gamma$ $(N_{b_i} \times 4)$ denotes the posterior distribution calculated from $\alpha$ and $\beta$. $\xi$ $((N_{b_i} - 1) \times 4 \times 4)$ denotes the joint distribution of two successive hidden states. We use $\theta = (\phi^{(b_i)}, \sigma^{(b_i)})$ to denote all parameters to be estimated.

$$
\begin{aligned}
Q\left(\theta, \theta^{\text{old}}\right) \quad &= \sum_Z p\left(Z \mid y^{(b_i)}, \theta^{\text{old}}\right) \log p\left(y^{(b_i)}, Z \mid \theta\right) \\
&= \sum_{k=1}^4 \gamma_{1k} \log \pi_k^{inner} + \sum_{n=2}^{N_{b_i}} \sum_{j=1}^4 \sum_{k=1}^4 \xi_{n-1,j,k} \log T_{jk}^{inner} \\
&+ \sum_{n=1}^{N_{b_i}} \left\{ \gamma_{n1} \log p\left(y_n^{(b_i)} \mid \phi^{(b_i)}, \sigma^{(b_i)}\right) + \gamma_{n2} \log p\left(y_n^{(b_i)} \mid \phi^{(b_{i-1})}, \sigma^{(b_{i-1})}\right) \right. \\
&\left. + \gamma_{n3} \log p\left(y_n^{(b_i)} \mid \phi^{(b_{i+1})}, \sigma^{(b_{i+1})}\right) + \gamma_{n4} \log p\left(y_n^{(b_i)} \mid lb, ub\right) \right\}
\end{aligned} \tag{1}
$$

Here we only need to update $\alpha$, $\beta$, $\gamma$, $\xi$, $\phi^{(b_i)}$ and $\sigma^{(b_i)}$ in the inference process of inner HMM for $i$th base block. Our inference algorithm is a modification of the standard Forward-backward algorithm based on the defined $Q$-function, as described in Appendix 2–algorithm 4.

---

**Appendix 2–algorithm 4. Inference algorithm of inner HMM.**

---

Input: $y^{(bi)}, \phi^{(bi)}, \sigma^{(bi)}, \phi^{(bi-1)}, \sigma^{(bi-1)}, \phi^{(bi+1)}, \sigma^{(bi+1)}, T^{inner}, \pi^{inner}, lb, ub$

Output: $\hat{\omega}^{(bi)}, \hat{h}^{(bi)}, \hat{\phi}^{(bi)}, \hat{\sigma}^{(bi)}$

Initialize $\hat{h}^{(bi)}$ and $\hat{\omega}^{(bi)}$ to the $1 \times N_{b_i}$ all zero vector.

Initialize α, β, γ, ξ

 // EM algorithm

for *round* = 1, 2, · · · , *R* do

 E-step: update α, β, γ, and ξ given $y^{(bi)}, \phi^{(bi)}, \sigma^{(bi)}, \phi^{(bi-1)}, \sigma^{(bi-1)}, \phi^{(bi+1)}, \sigma^{(bi+1)}, T^{inner}, \pi^{inner}, lb, ub$

 M-step: calculate $\hat{\phi}^{(bi)}$ and $\hat{\sigma}^{(bi)}$ by *Equation 2* and *Equation 3*

 $\phi^{(bi)} \leftarrow \hat{\phi}^{(bi)}$

 $\sigma^{(bi)} \leftarrow \hat{\sigma}^{(bi)}$

 Calculate the Q-function (log marginal likelihood) $\hat{q}$ given all estimated parameters

end

Derive $\hat{h}^{(b_i)}$ from γ by taking the state with highest probability for each data point, i.e.

$h_n^{(b_i)} = argmax([\gamma_{n1}, \gamma_{n2}, \gamma_{n3}, \gamma_{n4}])$

Set each element of $\hat{\omega}^{(b_i)}$ to $\exp(\hat{q}/N_{b_i})$.

---

$$\phi^{(b_i)} = \frac{\sum_{n=1}^{N_{b_i}} \gamma_{n1} y_n^{(b_i)}}{\sum_{n=1}^{N_{b_i}} \gamma_{n1}} \tag{2}$$

$$\sigma^{(b_i)} = \frac{\sum_{n=1}^{N_{b_i}} \gamma_{n1} (y_n^{(b_i)} - \phi^{(b_i)})^2}{\sum_{n=1}^{N_{b_i}} \gamma_{n1}} \tag{3}$$

## GPU-accelerated parameter inference

Given the general inference algorithm ('General inference algorithm') and algorithms for calculating the emission probabilities for the inner HMM (Appendix 2–algorithm 4) and linear model (Appendix 2–algorithm 3), we have the full algorithm for the parameter inference. The actual inference, however, can be really slow since the input fast5 data are huge. Here we use heuristics and GPU parallelization to accelerate the parameter inference.

We divide the raw current signal of each read $y$ into overlapping regions, which start at an atomic base block and also end with an atomic base block. As shown in *Appendix 2—figure 2*, we process the *r*th region $y^{(r)}$ independently to get the segmentation results $z^{(r)}$. This process is based on the following heuristics: the segmentation result of one region will not affect another region when they are far away from each other. The inference is performed in two rounds. In the first round, we obtain the segmentation results for non-overlapping regions. The center part of the segmentation is not affected by the flank parts, which may be affected by neighboring regions. In the second round, we merge relevant parts of neighboring regions near the touch point as a new region based on segmentation results of the first round. After that, we perform the inference for the new regions to get the final segmentations for the whole read.

We use GPU to parallelize the inference algorithm of inner HMM (Alg. 4). For each region, we first enumerate all possible paths and calculate the likelihood for each candidate path ('Enumeration of hidden states of outer HMM'). As shown in *Appendix 2—figure 2*, we need to perform inner HMM parameter inference for different paths of different regions of one read. We consider parallelizing the computation of different merged blocks $seg(i, j)$ in different paths of different regions. We first collect all merged blocks and sort them by size and only keep unique blocks, which can be identified by the combinatorial key of region index, start atomic block index, end atomic block index. Then we run the inner HMM inference (Alg. 4) of each merged block on a GPU core. Note that all GPU cores perform the same computation step of Alg. 4 simultaneously, but for different merged blocks.

**a**

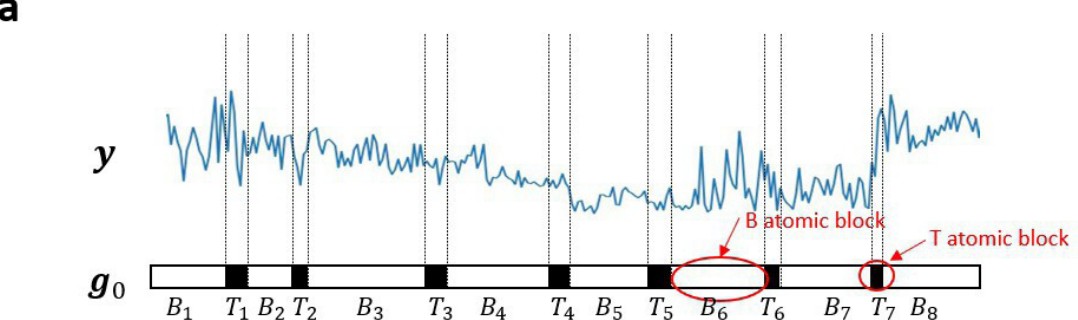

**b**

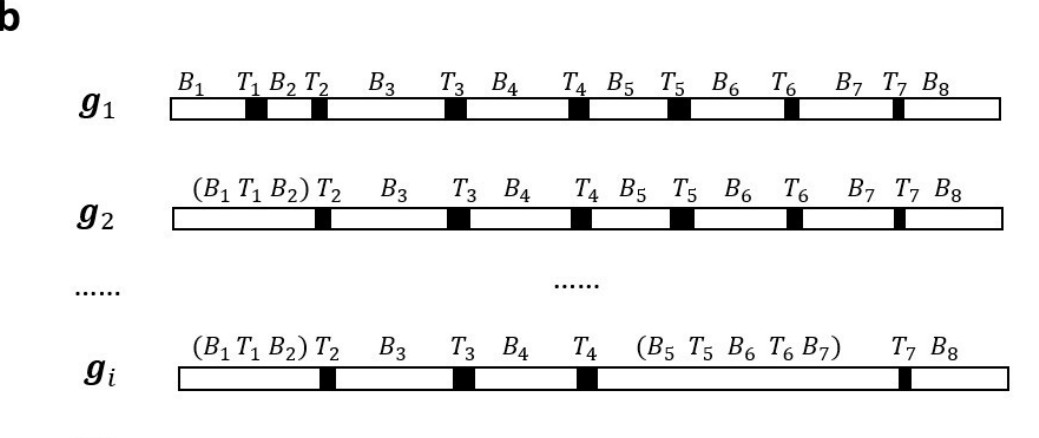

**Appendix 2—figure 1.** Illustration of enumeration of hidden states of outer HMM. (**a**) Raw signal and initialized to atomic segmentations. (**b**) Examples of the full configuration set $\mathcal{G} = (g_1, g_2, \cdots, g_i, \cdots, g_M)$.

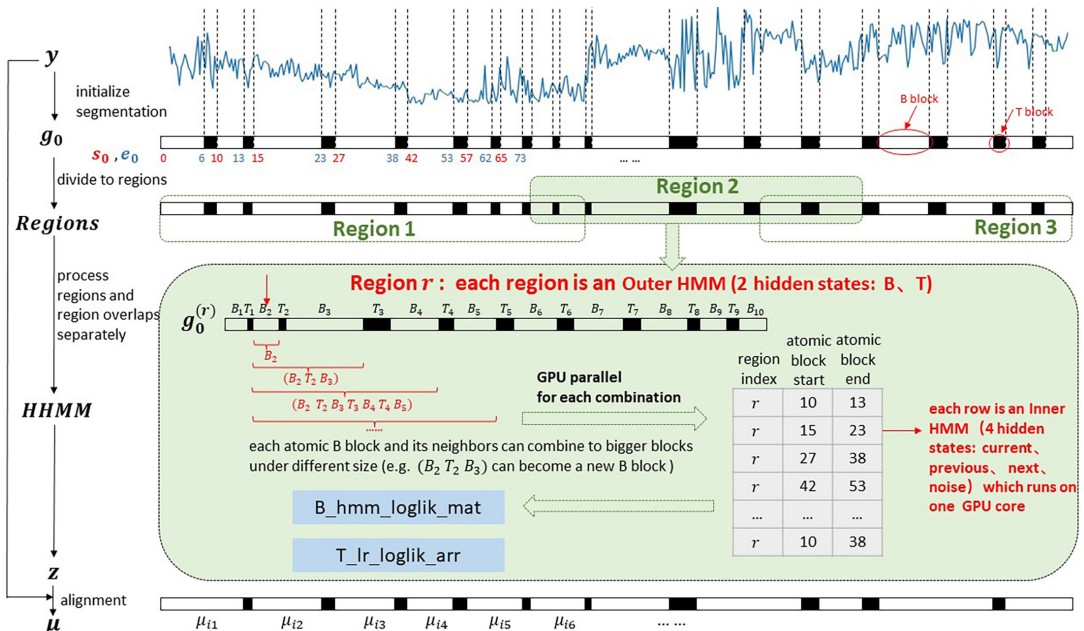

**Appendix 2—figure 2.** Illustration of GPU-accelerated parameter inference.

