## [Editor Report · eLife Assessment]

This study presents SegPore, a **valuable** new method for processing direct RNA nanopore sequencing data, which improves the segmentation of raw signals into individual bases and boosts the accuracy of modified base detection. The evidence presented to benchmark SegPore is **solid**, and the authors provide a fully documented implementation of the method. SegPore will be of particular interest to researchers studying RNA modifications.

---

## [Referee Report · Reviewer #1 (Public review)]

Summary:

In this manuscript, the authors describe a new computational method (SegPore), which segments the raw signal from nanopore direct RNA-Seq data to improve the identification of RNA modifications. In addition to signal segmentation, SegPore includes a Gaussian Mixture Model approach to differentiate modified and unmodified bases. SegPore uses Nanopolish to define a first segmentation, which is then refined into base and transition blocks. SegPore also includes a modification prediction model that is included in the output. The authors evaluate the segmentation in comparison to Nanopolish and Tombo (RNA002) as well as f5c and Uncalled 4 (RNA004), and they evaluate the impact on m6A RNA modification detection using data with known m6A sites. In comparison to existing methods, SegPore appears to improve the ability to detect m6A, suggesting that this approach could be used to improve the analysis of direct RNA-Seq data.

Strengths:

SegPore address an important problem (signal data segmentation). By refining the signal into transition and base blocks, noise appears to be reduced, leading to improved m6A identification at the site level as well as for single read predictions. The authors provide a fully documented implementation, including a GPU version that reduces run time. The authors provide a detailed methods description, and the approach to refine segments appears to be new.

---

## [Referee Report · Reviewer #2 (Public review)]

Summary:

The work seeks to improve detection of RNA m6A modifications using Nanopore sequencing through improvements in raw data analysis. These improvements are said to be in the segmentation of the raw data, although the work appears to position the alignment of raw data to the reference sequence and some further processing as part of the segmentation, and result statistics are mostly shown on the 'data-assigned-to-kmer' level.

As such, the title, abstract and introduction stating the improvement of just the 'segmentation' does not seem to match the work the manuscript actually presents, as the wording seems a bit too limited for the work involved.

The work itself shows minor improvements in m6Anet when replacing Nanopolish' eventalign with this new approach, but clear improvements in the distributions of data assigned per kmer. However, these assignments were improved well enough to enable m6A calling from them directly, both at site-level and at read-level.

A large part of the improvements shown appear to stem from the addition of extra, non-base/kmer specific, states in the segmentation/assignment of the raw data, removing a significant portion of what can be considered technical noise for further analysis. Previous methods enforced assignment of (almost) all raw data, forcing a technically optimal alignment that may lead to suboptimal results in downstream processing as datapoints could be assigned to neighbouring kmers instead, while random noise that is assigned to the correct kmer may also lead to errors in modification detection.

For an optimal alignment between the raw signal and the reference sequence, this approach may yield improvements for downstream processing using other tools.

Additionally, the GMM used for calling the m6A modifications provides a useful, simple and understandable logic to explain the reason a modification was called, as opposed to the black models that are nowadays often employed for these types of tasks.

Appraisal:

The authors have shown their methods ability to identify noise in the raw signal and remove their values from the segmentation and alignment, reducing its influences for further analyses. Figures directly comparing the values per kmer do show a visibly improved assignment of raw data per kmer. As a replacement for Nanopolish' eventalign it seems to have a rather limited, but improved effect, on m6Anet results. At the single read level modification modification calling this work does appear to improve upon CHEUI.

---

## [Referee Report · Reviewer #3 (Public review)]

Summary:

Nucleotide modifications are important regulators of biological function, however, until recently, their study has been limited by the availability of appropriate analytical methods. Oxford Nanopore direct RNA sequencing preserves nucleotide modifications, permitting their study, however many different nucleotide modifications lack an available base-caller to accurately identify them. Furthermore, existing tools are computationally intensive, and their results can be difficult to interpret.

Cheng et al. present SegPore, a method designed to improve the segmentation of direct RNA sequencing data and boost the accuracy of modified base detection.

Strengths:

This method is well described and has been benchmarked against a range of publicly available base callers that have been designed to detect modified nucleotides.

Comment from the Reviewing Editor:

The authors have provided responses to the weaknesses highlighted previously and the reviewers were not asked to comment. The authors have now requested a Version of Record.

---

## [Author Response]

The following is the authors’ response to the previous reviews

**Public Reviews:**

**Reviewer #1 (Public review):**
Summary:In this manuscript, the authors describe a new computational method (SegPore), which segments the raw signal from nanopore direct RNA-Seq data to improve the identification of RNA modifications. In addition to signal segmentation, SegPore includes a Gaussian Mixture Model approach to differentiate modified and unmodified bases. SegPore uses Nanopolish to define a first segmentation, which is then refined into base and transition blocks. SegPore also includes a modification prediction model that is included in the output. The authors evaluate the segmentation in comparison to Nanopolish and Tombo (RNA002) as well as f5c and Uncalled 4 (RNA004), and they evaluate the impact on m6A RNA modification detection using data with known m6A sites. In comparison to existing methods, SegPore appears to improve the ability to detect m6A, suggesting that this approach could be used to improve the analysis of direct RNA-Seq data.Strengths:SegPore address an important problem (signal data segmentation). By refining the signal into transition and base blocks, noise appears to be reduced, leading to improved m6A identification at the site level as well as for single read predictions. The authors provide a fully documented implementation, including a GPU version that reduces run time. The authors provide a detailed methods description, and the approach to refine segments appears to be new.Weaknesses:The authors show that SegPore reduces noise compared to other methods, however the improvement in accuracy appears to be relatively small for the task of identifying m6A. To run SegPore, the GPU version is essential, which could limit the application of this method in practice.

As discussed in Paragraph 4 of the Discussion, we acknowledge that the improvement of SegPore combined with m6Anet over Nanopolish+m6Anet in bulk in vivo analysis is modest. This outcome is likely influenced by several factors, including alignment inaccuracies caused by pseudogenes or transcript isoforms, the presence of additional RNA modifications that can affect signal baselines, and the fact that m6Anet is specifically trained on Nanopolish-derived events. Additionally, the absence of a modification-free (in vitro transcribed) control sample in the benchmark dataset makes it challenging to establish true k-mer baselines.

Importantly, these challenges do not exist for in vitro data, where the signal is cleaner and better defined. As a result, SegPore achieves a clear and substantial improvement at the single-molecule level, demonstrating the strength of its segmentation approach and its potential to significantly enhance downstream analyses. These results indicate that SegPore is particularly well suited for benchmarking and mechanistic studies of RNA modifications under controlled experimental conditions, and they provide a strong foundation for future developments.

We also recognize that the current requirement for GPU acceleration may limit accessibility in some computational environments. To address this, we plan to further optimize SegPore in future versions to support efficient CPU-only execution, thereby broadening its applicability and impact.

**Reviewer #2 (Public review):**
Summary:The work seeks to improve detection of RNA m6A modifications using Nanopore sequencing through improvements in raw data analysis. These improvements are said to be in the segmentation of the raw data, although the work appears to position the alignment of raw data to the reference sequence and some further processing as part of the segmentation, and result statistics are mostly shown on the 'data-assigned-to-kmer' level.As such, the title, abstract and introduction stating the improvement of just the 'segmentation' does not seem to match the work the manuscript actually presents, as the wording seems a bit too limited for the work involved.The work itself shows minor improvements in m6Anet when replacing Nanopolish' eventalign with this new approach, but clear improvements in the distributions of data assigned per kmer. However, these assignments were improved well enough to enable m6A calling from them directly, both at site-level and at read-level.A large part of the improvements shown appear to stem from the addition of extra, non-base/kmer specific, states in the segmentation/assignment of the raw data, removing a significant portion of what can be considered technical noise for further analysis. Previous methods enforced assignment of (almost) all raw data, forcing a technically optimal alignment that may lead to suboptimal results in downstream processing as datapoints could be assigned to neighbouring kmers instead, while random noise that is assigned to the correct kmer may also lead to errors in modification detection.For an optimal alignment between the raw signal and the reference sequence, this approach may yield improvements for downstream processing using other tools.Additionally, the GMM used for calling the m6A modifications provides a useful, simple and understandable logic to explain the reason a modification was called, as opposed to the black models that are nowadays often employed for these types of tasks.Weaknesses:The manuscript suggests the eventalign results are improved compared to Nanopolish. While this is believably shown to be true (Table 1), the effect on the use case presented, downstream differentiation between modified and unmodified status on a base/kmer, is likely limited for during downstream modification calling the noisy distributions are often 'good enough'. E.g. Nanopolish uses the main segmentation+alignment for a first alignment and follows up with a form of targeted local realignment/HMM test for modification calling (and for training too), decreasing the need for the near-perfect segmentation+alignment this work attempts to provide. Any tool applying a similar strategy probably largely negates the problems this manuscript aims to improve upon. Should a use-case come up where this downstream optimisation is not an option, SegPore might provide the necessary improvements in raw data alignment.

Thank you for this thoughtful comment. We agree that many current state-of-the-art (SOTA) methods perform well on benchmark datasets, but we believe there is still substantial room for improvement. Most existing benchmarks are based on limited datasets, primarily focusing on DRACH motifs in human and mouse transcriptomes. However, m6A modifications can also occur in non-DRACH motifs, where current models tend to underperform. Furthermore, other RNA modifications, such as pseudouridine, inosine, and m5C, remain less studied, and their detection is likely to benefit from more accurate and informative signal modeling.

It is also important to emphasize that raw signal segmentation and RNA modification detection are fundamentally distinct tasks. SegPore focuses on improving the segmentation step by producing a cleaner and more interpretable signal, which provides a stronger foundation for downstream analyses. Even if RNA modification detection algorithms such as m6Anet can partially compensate for noisy segmentation in specific cases, starting from a more accurate signal alignment can still lead to improved accuracy, robustness, and interpretability—particularly in challenging scenarios such as non-canonical motifs or less characterized modifications.

Scientific progress in this field is often incremental, and foundational improvements can have a significant long-term impact. By enhancing raw signal segmentation, SegPore contributes an essential building block that we expect will enable the development of more accurate and generalizable RNA modification detection algorithms as the community integrates it into more advanced workflows.

Appraisal:The authors have shown their methods ability to identify noise in the raw signal and remove their values from the segmentation and alignment, reducing its influences for further analyses. Figures directly comparing the values per kmer do show a visibly improved assignment of raw data per kmer. As a replacement for Nanopolish' eventalign it seems to have a rather limited, but improved effect, on m6Anet results. At the single read level modification modification calling this work does appear to improve upon CHEUI.Impact:With the current developments for Nanopore based modification calling largely focusing on Artificial Intelligence, Neural Networks and the likes, improvements made in interpretable approaches provide an important alternative that enables deeper understanding of the data rather than providing a tool that plainly answers the question of wether a base is modified or not, without further explanation. The work presented is best viewed in context of a workflow where one aims to get an optimal alignment between raw signal data and the reference base sequence for further processing. For example, as presented, as a possible replacement for Nanopolish' eventalign. Here it might enable data exploration and downstream modification calling without the need for local realignments or other approaches that re-consider the distribution of raw data around the target motif, such as a 'local' Hidden Markov Model or Neural Networks. These possibilities are useful for a deeper understanding of the data and further tool development for modification detection works beyond m6A calling.
**Reviewer #3 (Public review):**
Summary:Nucleotide modifications are important regulators of biological function, however, until recently, their study has been limited by the availability of appropriate analytical methods. Oxford Nanopore direct RNA sequencing preserves nucleotide modifications, permitting their study, however many different nucleotide modifications lack an available base-caller to accurately identify them. Furthermore, existing tools are computationally intensive, and their results can be difficult to interpret.Cheng et al. present SegPore, a method designed to improve the segmentation of direct RNA sequencing data and boost the accuracy of modified base detection.Strengths:This method is well described and has been benchmarked against a range of publicly available base callers that have been designed to detect modified nucleotides.Weaknesses:However, the manuscript has a significant drawback in its current version. The most recent nanopore RNA base callers can distinguish between different ribonucleotide modifications, however, SegPore has not been benchmarked against these models.The manuscript would be strengthened by benchmarking against the rna004_130bps_hac@v5.1.0 and rna004_130bps_sup@v5.1.0 dorado models, which are reported to detect m5C, m6A_DRACH, inosine_m6A and PseU.A clear demonstration that SegPore also outperforms the newer RNA base caller models will confirm the utility of this method.

Thank you for highlighting this important limitation. While Dorado, the new ONT basecaller, is publicly available and supports modification-aware basecalling, suitable public datasets for benchmarking m5C, inosine, m6A, and PseU detection on RNA004 are currently lacking. Dorado’s modification-aware models are trained on ONT’s internal data, which is not publicly released. Therefore, it is currently not feasible to directly evaluate or compare SegPore’s performance against Dorado for these RNA modifications.

We would also like to emphasize that SegPore’s primary contribution lies in raw signal segmentation, which is an upstream and foundational step in the RNA modification detection pipeline. As more publicly available datasets for RNA004 modification detection become accessible, we plan to extend our work to benchmark and integrate SegPore with modification detection tasks on RNA004 data in future studies.

**Recommendations for the authors:**

**Reviewer #2 (Recommendations for the authors):**
Comments based on Author Response“However, it is valid to compare them on the segmentation task, where SegPore exhibits better performance (Table 1).”This dodges the point of the actual use case of this approach, as Nanopolish indeed does not support calling modifications for this kind of data, but the general approach it uses might, if adapted for this data, nullify the gains made in the examples presented.

We respectfully disagree with the comment that the advantages demonstrated by SegPore could be “nullified”. Although SegPore’s performance is indeed more modest in in vivo datasets, it shows substantially better performance than CHEUI in in vitro data, clearly demonstrating that improved segmentation directly contributes to more accurate RNA modification estimation.

It is worth noting that CHEUI relies on Nanopolish’s segmentation results for m6A detection. Despite this, SegPore outperforms CHEUI, further supporting the conclusion that segmentation quality has a meaningful impact on downstream modification calling.

In conclusion, based on our current experimental results, SegPore is particularly well suited for RNA modification analysis from in vitro transcribed data, where its improved segmentation provides a clear advantage over existing methods.

Further comments(2) “(2) Page 3 employ models like Hidden Markov Models (HMM) to segment the signal, but they are prone to noise and inaccuracies”“That's the alignment/calling part, not the segmentation?”“Current methods, such as Nanopolish, employ models like Hidden Markov Models (HMM) to segment the signal”I get the impression the word 'segment' has a different meaning in this work than what I'm used to based on my knowledge around Nanopolish and Tombo, see the deeper code examples further down below.Additionally, in Nanopolish there is a clear segmentation step (or event detection) without any HMM, then a sort of dynamic timewarping step that aligns the segments and re-combines some segments into a single segment where necessary afterwards. I believe the HMM in Nanopolish is not used at all unless modification calling, but if you can point out otherwise I'm open for proof.Now I believe it is the meaning of 'segmenting the signal' that confuses me, and now the clarification makes it a bit odd as well:“Nanopolish and Tombo align the raw signal to the reference sequence to determine which portion of the signal corresponds to each k-mer. We define this process as the segmentation task, referred to as "eventalign" in Nanopolish.”So now it's clearly stated the raw signal is being 'aligned' and then the process is suddenly defined as the 'segmentation task', and again referred to as "eventalign". Why is it not referred to as the 'alignment task' instead?I understand the segmentation and alignment parts are closely connected but to me, it seems this work picks the wrong word for the problem being solved.“Unlike Nanopolish and Tombo, which directly align the raw signal to the reference sequence,…”Looking at their code, I believe both Nanopolish and Tombo actually do segment the data first (or "event detection"), then they align the segments/events they found, and finally multiple events aligned to the same section are merged. See for yourself:Nanopolish:
here
Line 233:cpptrim_and_segment_raw(fast5_data.rt, trim_start, trim_end, varseg_chunk, varseg_thresh);event_table et = detect_events(fast5_data.rt, *ed_params);Line 270:cpp// align events to the basecalled readstd::vector event_alignment = adaptive_banded_simple_event_align(*this, *this->base_model[strand_idx], read_sequence);Where event detection is further defined at line 268 here:
here
Tombo:
here
line 1162 and onwards shows a ‘segment_signal’ call and the results are used in a ‘find_adaptive_base_assignment’ call, where ‘segment_signal’ starting at line 1057 tries to find where the signal jumps from a series of similar values to another (start of a base change in the pore), stored in ‘valid_cpts’, and the ‘find_adaptive_base_assignment’ tries to align the resulting segment values to the expected series of values:pythonvalid_cpts, norm_signal, new_scale_values = segment_signal(map_res, num_events, rsqgl_params, outlier_thresh, const_scale)event_means = ts.compute_base_means(norm_signal, valid_cpts)dp_res = find_adaptive_base_assignment (valid_cpts, event_means, rsqgl_params, std_ref, map_res.genome_seq,start_clip_bases=map_res.start_clip_bases,seq_samp_type=seq_samp_type, reg_id=map_res.align_info.ID)These implementations are also why I find the choice of words for what is segmentation and what is alignment a bit confusing in this work, as both Tombo and Nanopolish do a similar, clear segmentation step (or an "event detection" step), followed by the alignment of the segments they determined. The terminology in this work appears to deviate from these.

We thank the reviewer for the detailed comments!

First of all, we sincerely apologize for our earlier misunderstanding regarding how Nanopolish and Tombo operate. Based on a closer examination of their source codes, we now recognize that both tools indeed include a segmentation step based on change-point detection methods, after which the resulting segments are aligned to the reference sequence. We have revised the relevant text in the manuscript accordingly:

- “Current methods, such as Nanopolish, employ change-point detection methods to segment the signal and use dynamic programming methods and HMM to align the derived segments to the reference sequence,”

- “We define this process as the segmentation and alignment task (abbreviated as the segmentation task), which is referred to as “eventalign” in Nanopolish.”

- “In SegPore, we segment the raw signal into small fragments using a Hierarchical Hidden Markov Model (HHMM) and align the mean values of these fragments to the reference, where each fragment corresponds to a sub-state of a k-mer. By contrast, Nanopolish and Tombo use change-point–based methods to segment the signal and employ dynamic programming approaches together with profile HMMs to align the resulting segments to the reference sequence.”

Regarding terminology, we originally borrowed the term “segmentation” from speech processing, where it refers to dividing continuous audio signals into meaningful units. In the context of nanopore signal analysis, segmentation and alignment are often tightly coupled steps. Because of this and because our initial focus was on methodological development rather than terminology, we used the term “segmentation task” to describe the combined process of signal segmentation and alignment.

However, we now recognize that this terminology may cause confusion. Changing every instance of “segmentation” to “segmentation and alignment” or “alignment” would require substantial rewriting of the manuscript. Therefore, in this revision, we have clearly defined “segmentation task” as referring to the combined process of segmentation and alignment. We apologize for any earlier confusion and will adopt the term “alignment” in future work for greater clarity.

(3) I think I do understand the meaning, but I do not understand the relevance of the Aj bit in the last sentence. What is it used for?Based on the response and another close look at Fig1, it turns out the j refers to extremely small numbers 1 and 2 in step 3. You may want in improve readability for these.

Thank you for the suggestion. We have added subscripts to all nucleotides in the reference sequence in Figure 1A and revised the legend to clarify the notation and improve readability. Specifically, we now include the following explanation:

“For example, A_j_ denotes the base ‘A’ at the j-th position on the reference sequence. In this example, A_1_ and A_2_ refer to the first and second occurrences of ‘A’ in the reference sequence, respectively. Accordingly, μ_1_ and μ_2_ are aligned to A_1_, while μ_3_ is aligned to A_2_”.

(6) “We chose to use the poly(A) tail for normalization because it is sequence-invariant- i.e., all poly(A) tails consist of identical k-mers, unlike transcript sequences which vary in composition. In contrast, using the transcript region for normalization can introduce biases: for instance, reads with more diverse k-mers (having inherently broader signal distributions) would be forced to match the variance of reads with more uniform k-mers, potentially distorting the baseline across k-mers.”While the next part states there was a benchmark showing SegPore still works without this normalization, I think this answer does not touch upon the underlying issue I'm trying to point out here.- The biases mentioned here due to a more diverse (or different) subsets of k-mers in a read indeed affects the variance of the signal overall.- As I pointed out in my earlier remark here, this can be resolved using an approach of 'general normalization', 'mapping to expected signal', 'theil-sen fitting of scale and offset', 're-mapping to expected signal', as Tombo and Nanopolish have implemented.- Alternatively, one could use the reference sequence (using the read mapping information) and base the expected signal mean and standard deviation on that instead.- The polyA tail stability as an indicator for the variation in the rest of the signal seems a questionable assumption to me. A 'noisy' pore could introduce a large standard deviation using the polyA tail without increasing the deviations on the signal induced by the variety of k-mers, rather it would be representative for the deviations measured within a single k-mer segment. I thought this possible discrepancy is to be expected from a worn out pore, hence I'd imagine reads sequenced later in a run to provide worse results using this method.In the current version it is not the statement that is unclear, it is the underlying assumption of how this works that I question.

We thank the reviewer for raising this important point and for the insightful discussion. Our choice of using the poly(A) tail for normalization is based on the working hypothesis that the poly(A) signal reflects overall pore-level variability and provides a stable reference for signal scaling. We find this to be a practical and effective approach in most experimental settings.

We agree that more sophisticated strategies, such as “general normalization” or iterative fitting to the expected signal (as implemented in Tombo and Nanopolish), could in principle generate a "better" normalization. However, these approaches are significantly more challenging to implement in practice. This is because signal normalization and alignment are mutually dependent processes: baseline estimates for k-mers influence alignment accuracy, while alignment accuracy, in turn, affects baseline calculation. This interdependence becomes even more complex in the presence of RNA modifications, which alter signal distributions and further confound model fitting.

It is worth noting that this limitation is already evident in our results. As shown in Figure 4B (first and second k-mers), Nanopolish produces more dispersed baselines than SegPore, even for these unmodified k-mers, suggesting inherent limitations in its normalization strategy. Ideally, baselines for the same k-mer should remain highly consistent across different reads.

In contrast, poly(A)-based normalization offers a simpler and more robust solution that avoids this circular dependency. Because poly(A) sequences are compositionally homogeneous, they enable reliable estimation of scaling parameters without assumptions about k-mer composition or modification state. Regarding the reviewer’s concern about pore instability, we mitigate this issue by including only high-quality, confidently mapped reads in our analysis, which reduces the likelihood of incorporating signals from degraded or “noisy” pores.

We fully agree that exploring more advanced normalization strategies is an important direction for future work, and we plan to investigate such approaches as the field progresses.

(8) “In the remainder of this paper, we refer to these resulting events as the output of eventalign analysis or the segmentation task.”Picking only one descriptor rather than two alternatives would be easier to follow (and I'd prefer the first).

Thank you for the suggestion. We have revised the sentence to:

“In the remainder of this paper, we refer to these resulting events as the output of eventalign analysis, which also represents the final output of the segmentation and alignment task.”

(9) “Additionally, a complete explanation of how the weighted mean is computed is provided in Section 5.3 of Supplementary Note 1. It is derived from signal points that are assigned to a given 5mer.”I believe there's no more mention of a weighted mean, and I don't get any hits when searching for 'weight'. Is that intentional?

We apologize for the misplacement of the formulas. We have updated Section 5.3 of Supplementary Note 1 to clarify the definition of the weighted mean. Because multiple current signal segments may be aligned to a single *k*-mer, we computed the weighted mean for each *k*-mer across these segments, where the weight corresponds to the number of data points assigned to “curr” state in each event.

(17) Response: We revised the sentence to clarify the selection criteria: "For selected 5mers “that exhibit both a clearly unmodified and a clearly” “modified signal component”, “SegPore reports the modification rate at each site,” “as well as the modification state of that site on individual reads.””So is this the same set described on page 13 ln 343 or not?“Due to the differences between human (Supplementary Fig. S2A) and mouse (Supplementary Fig. S2B), only six 5mers were found to have m6A annotations in the test data's ground truth (Supplementary Fig. S2C). For a genomic location to be identified as a true m6A modification site, it had to correspond to one of these six common 5mers and have a read coverage of greater than 20.”

I struggle to interpret the 'For selected 5mers' part, as I'm not sure if this is a selection I'm supposed to already know at this point in the text or if it's a set just introduced here. If the latter, removing the word 'selected' would clear it up for me.

We apologize for the confusion. What we mean is that when pooling signals aligned to the same k-mer across different genomic locations and reads, only a subset of k-mers exhibit a bimodal distribution — one peak corresponding to the unmodified state and another to the modified state. Other k-mers show a unimodal distribution, making it impossible to reliably estimate modification levels. We refer to the subset of k-mers that display a bimodal distribution as the “selected” k-mers.

The “selected k-mers” described on page 13, line 343, must additionally have ground truth labels available in both the training and test datasets. There are 10 k-mers with ground truth annotations in the training data and 11 in the test data, and only 6 of these k-mers are shared between the two datasets, therefore only those 6 overlapping k-mers are retained for evaluation. These 6 k-mers satisfy both criteria: (1) exhibiting a bimodal distribution and (2) having ground truth annotations in both training and test sets.

To improve clarity, we have removed the term “selected” from the sentence.

(21) "Tombo used the "resquiggle" method to segment the raw signals, and we standardized the segments using the “poly(A)” tail to ensure a fair comparison “(See” “preprocessing section in Materials and Methods)."”In the Materials and Methods:“The raw signal segment corresponding to the poly(A) tail is used to standardize the raw signal for each read.”I cannot find more detailed information here on what the standardization does, do you mean to refer to Supplementary Note 1, Section 3 perhaps?

Thank you for pointing this out. Yes, the standardization procedure is described in detail in Supplementary Note 1, Section 3. Tombo itself does not segment and align the raw signal on the absolute pA scale, which can result in very large variance in the derived events if the raw signal is used directly. To ensure a fair comparison, we therefore applied the same preprocessing steps to Tombo’s raw signals as we did for SegPore, using only the event boundary information from Tombo while standardizing the signal in the same way.

We have revised the sentence for clarity as follows:

“Tombo used the "resquiggle" method to segment the raw signals, but the resulting signals are not reported on the absolute pA scale. To ensure a fair comparison with SegPore, we standardized the segments using the poly(A) tail in the same way as SegPore (See preprocessing section in Materials and Methods).”

(22A) The table shown does help showing the benchmark is unlikely to be 'cheated'. However I am suprised to see the Avg std for Nanopolish and Tombo going up instead of down, as I'd expect the transition values to increase the std, and hence, removing them should decrease these values. So why does this table show the opposite?I believe this table is not in the main text or the supplement, would it not be a good idea to cover this point somewhere in the work?

Thank you for this insightful comment. In response, we carefully re-examined our analysis and identified a bug in the code related to boundary removal for Nanopolish. We have now corrected this issue and included the updated results in Supplementary Table S1 of the revised manuscript. As shown in the updated table, the average standard deviations decrease after removing the boundary regions for both Nanopolish and Tombo.

We have now included this table in Supplementary Table S1 in the revised manuscript and added the following clarification:

“It is worth noting that the data points corresponding to the transition state between two consecutive 5-mers are not included in the calculation of the standard deviation in SegPore’s results in Table 1. However, their exclusion does not affect the overall conclusion, as there are on average only ~6 points per 5-mer in the transition state (see Supplementary Table S1 for more details).”

(22B) As mentioned in 2, I'm happy there's a clear definition of what is meant but I found the chosen word a bit odd.

We apologize for the earlier unclear terminology. We now refer to it as the segmentation and alignment task, abbreviated as the segmentation task.

(23) Reading back I can gather that from the text earlier, but the summation of what is being tested is this:“including Tombo, MINES (31), Nanom6A (32), m6Anet, Epinano (33), and CHEUI (20). “next, the identifier "Nanopolish+m6Anet" is, aside from the figure itself, only mentioned in the discussion. Adding a line that explains that "Nanopolish+m6Anet" is the default method of running m6Anet and "SegPore+m6Anet" replaces the Nanopolish part for m6Anet with Segpore, rather than jumping straight to "SegPore+m6Anet", would clarify where this identifier came from.

Thank you for the helpful suggestion. We have added the identifier to the revised manuscript as follows:

“Given their comparable methodologies and input data requirements, we benchmarked SegPore against several baseline tools, including Tombo, MINES (31), Nanom6A (32), m6Anet, Epinano (33), and CHEUI (20). By default, MINES and Nanom6A use eventalign results generated by Tombo, while m6Anet, Epinano, and CHEUI rely on eventalign results produced by Nanopolish. In Fig. 3C, ‘Nanopolish+m6Anet’ refers to the default m6Anet pipeline, whereas ‘SegPore+m6Anet’ denotes a configuration in which Nanopolish’s eventalign results are replaced with those from SegPore.”

(24) For completeness I'd expect tickmarks and values on the y-axis as well.

Thank you for the suggestion. We have updated Figures 3A and 3B in the revised manuscript to include tick marks and values on the y-axis as requested.

(25) Considering this statement and looking back at figure 3a and 3b, wouldn't this be easier to observe if the histograms/KDE's were plotted with overlap in a single figure?

We appreciate the suggestion. However, we believe that overlaying Figures 3A and 3B into a single panel would make the visualization cluttered and more difficult to interpret.

(29) Please change the sentence in the text to make that clear. As it is written now (while it's the same number of motifs, so one might guess it) it does not seem to refer to that particular set of motifs and could be a new selection of 6 motifs.

We appreciate the suggestion and have revised the sentence for clarity as follows:

“We evaluated m6A predictions using two approaches: (1) SegPore’s segmentation results were fed into m6Anet, referred to as SegPore+m6Anet, which works for all DRACH motifs and (2) direct m6A predictions from SegPore’s Gaussian Mixture Model (GMM), which is limited to the six selected 5-mers shown in Supplementary Fig. S2C that exhibit clearly separable modified and unmodified components in the GMM (see Materials and Methods for details). ”

(31) I think we have a different interpretation of the word 'leverage', or perhaps what it applies to. I'd say it leverages the jiggling if there's new information drawn from the jiggling behaviour. It's taking it into account if it filters for it. The HHMM as far as I understand tries to identify the jiggles, and ignore their values for the segmentation etc. So while one might see this as an approach that "leverages the hypothesis", I don't see how this HHMM "leverages the jiggling property" itself.

Thank you for the helpful suggestion. We have replaced the word “leverages” with “models” in the revised manuscript.

New pointspg6ln166: “…we extract the aligned raw signal segment and reference sequence segment from Nanopolish's events [...] we extract the raw signal segment corresponding to the transcript region for each input read based on Nanopolish's poly(A) detection results.”It is not clear as to why this different approach is applied for these two cases in this part of the text.

Thank you for pointing this out. The two approaches refer to different preprocessing strategies for in vivo and in vitro data.

For in vivo data, a large proportion of reads do not span the full-length transcript and often map only to a portion of the reference sequence. Moreover, because a single gene can generate multiple transcript isoforms, a read may align equally well to several possible transcripts. Therefore, we extract only the raw signal segment that corresponds to the mapped portion of the transcript for each read.

In contrast, for in vitro data, the transcript sequence is known precisely. As a result, we can directly extract all raw signals following the poly(A) tail and align them to the complete reference sequence.

pg10ln259: An important distinction from classical global alignment algorithms is that one or multiple base blocks may align with a single 5mer.”If there was usually a 1:1 mapping the alignment algorithm would be more or less a direct match, so I think the multiple blocks aligning to a 5mer thing is actually quite common.

Thank you for the comment. The “classical global alignment algorithm” here refers to the Needleman–Wunsch algorithm used for sequence alignment. Our intention was to highlight the conceptual difference between traditional sequence alignment and nanopore signal alignment. In classical sequence alignment, each base typically aligns to a single position in the reference. In contrast, in nanopore signal alignment, one or multiple signal segments — corresponding to varying dwell times of the motor protein — can align to a single 5-mer.

We have revised the sentence as follows:

“An important distinction from classical global alignment algorithms (Needleman–Wunsch algorithm)……”

pg13ln356: "dwell time" is not defined or used before, I guess it's effectively the number of raw samples per segment but this should be clarified.

Thank you for pointing this out. We have now added a clear definition of dwell time in the text as follows:

"such as the normalized mean μ_i, standard deviation σ_i, dwell time l_i (number of data points in the event)."

pg13ln358: “Feature vectors from 80% of the genomic locations were used for training, while the remaining 20% were set aside for validation.”I assume these are selected randomly but this is not explicitly stated here and should be.

Yes, they are randomly selected. We have revised the sentence as follows:

“Feature vectors from a randomly selected 80% of the genomic locations were used for training, while the remaining 20% were set aside for validation.”

pg18ln488: The manuscript now evaluates RNA004 and compares against f5c and Uncalled4. It mentions the differences between RNA004 and RNA002, namely kmer size and current levels, but does not explain where the starting reference model values for the RNA004 model come from: In pg18ln492 they state "RNA004 provides reference values for 9mers", then later they seem to use a 5mer parameter table (pg19ln508), are they re-using the same table from RNA002 or did they create a 5mer table from the 9mer reference table?

We apologize for the confusion. The reference model table for RNA004 9-mers is obtained from f5c (the array named ‘rna004_130bps_u_to_t_rna_9mer_template_model_builtin_data’in here).

**Author response image 1. sa4fig1:** 

We have revised the subsection header “5-mer parameter table” in the Method to “5-mer & 9-mer parameter table” to highlight this and added a paragraph about how to obtain the 9-mer parameter table:

“In the RNA004 data analysis (Table 2), we obtained the 9-mer parameter table from the source code of f5c (version 1.5). Specifically, we used the array named ‘rna004_130bps_u_to_t_rna_9mer_template_model_builtin_data’ from the following file: here (accessed on 17 October 2025).”

Also, in page 18 line 195, we added the following sentence:

“The 9-mer parameter table in pA scale for RNA004 data provided by f5c (see Materials and Methods) was used in the analysis.”

pg19ln520: “Additionally, due to the differences of the k-mer motifs between human and mouse (Supplementary Fig. S2), six shared 5mers were selected to demonstrate SegPore's performance in modification prediction directly.”"the differences" - in occurrence rates, as I gather from the supplementary figure, but it would be good to explicitly state it in this sentence itself too.

Thank you for the helpful suggestion. We agree that the original sentence was vague. The main reason for selecting only six 5-mers is the difference in the availability of ground truth labels for specific k-mer motifs between human and mouse datasets. We have revised the sentence accordingly:

“Additionally, due to the differences in the availability of ground truth labels for specific k-mer motifs between human and mouse (Supplementary Fig. S2), six shared 5-mers were selected to directly demonstrate SegPore’s performance in modification prediction.”

pg24ln654: “SegPore codes current intensity levels”"codes" is meant to be "stores" I guess? Perhaps "encodes"?

Thank you for the suggestion. We have now replaced it with “encodes” in the revised manuscript.

Lastly, looking at the feedback from the other reviewers comment:The 'HMM' mentioned in line 184 looks fine to me, the HHMM is 2 HMM's in a hierarchical setup and the text now refers to one of these HMM layers. If this is to be changed it would need to state the layer (e.g. "the outer HHMM layer") throughout the text instead.

We agree with this assessment and believe that the term “inner HMM” is accurate in this context, as it correctly refers to one of the two HMM layers within the HHMM structure. Therefore, we have decided to retain the current terminology.

**Reviewer #3 (Recommendations for the authors):**
I recommend the publication of this manuscript, provided that the following comments are addressed.Page 5, Preprocessing: You comment that the poly(A) tail provides a stable reference that is crucial for the normalisation of all reads. How would this step handle reads that have interrupted poly(A) tails (e.g. in the case of mRNA vaccines that employ a linker sequence)? Or cell types that express TENT4A/B, which can include transcripts with non-A residues in the poly(A) tail: here.

It depends on Nanopolish’s ability to reliably detect the poly(A) tail. In general, the poly(A) region produces a long stretch of signals fluctuating around a current level of ~108.9 pA (RNA002) with relatively stable variation, which allows it to be identified and used for normalization.

For in vivo data, if the poly(A) tail is interrupted (e.g., due to non-A residues or linker sequences), two scenarios are possible:

(1) The poly(A) tail may not be reliably detected, in which case the corresponding read will be excluded from our analysis.

(2) Alternatively, Nanopolish may still recognize the initial uninterrupted portion of the poly(A) signal, which is typically sufficient in length and stability to be used for signal normalization.

For in vitro data, the poly(A) tails are uninterrupted, so this issue does not arise.

All analyses presented in this study are based exclusively on reads with reliably detected poly(A) tails.

Page 7, 5mer parameter table: r9.4_180mv_70bps_5mer_RNA is an older kmer model (>2 years). How does your method perform with the newer RNA kmer models that do permit the detection of multiple ribonucleotide modifications? Addressing this comment would be beneficial, however I understand that it would require the generation of new data, as limited RNA004 datasets are available in the public domain.

“r9.4_180mv_70bps_5mer_RNA” is the most widely used k-mer model for RNA002 data. Regarding the newer k-mer models, we believe the reviewer is referring to the “modification basecalling” models available in Dorado, which are specifically designed for RNA004 data. At present, SegPore can perform RNA modification estimation only on RNA002 data, as this is the platform for which suitable training data and ground truth annotations are available. Evaluating SegPore’s performance with the newer RNA004 modification models would require new datasets containing known modification sites generated with RNA004 chemistry. Since such data are currently unavailable, we have not yet been able to assess SegPore under these conditions. This represents an important future direction for extending and validating our method.

The Methods and Results sections contain redundant information -please streamline the information in these sections and reduce the redundancy.

We thank the reviewer for this suggestion and acknowledge that there is some overlap between the Methods and Results sections. However, we feel that removing these parts could compromise the clarity and readability of the manuscript, especially given that Reviewer 2 emphasized the need for clearer explanations. We therefore decided to retain certain methodological descriptions in the Results section to ensure that key steps are understandable without requiring the reader to constantly cross-reference the Methods.

Minor commentsPlease be consistent when referring to k-mers and 5-mers (sometimes denoted as 5mers - please change to 5-mers throughout).

We have revised the manuscript to ensure consistency and now use “5-mers” throughout the text.

IntroductionLines 80 - 112: Please condense this section to roughly half the length (1-2 paragraphs). In general, the results described in the introduction should be very brief, as they are described in full in the results section.

Thank you for the suggestion. We have condensed the original three paragraphs into a single, more concise paragraph as follows:

"SegPore is a novel tool for direct RNA sequencing (DRS) signal segmentation and alignment, designed to overcome key limitations of existing approaches. By explicitly modeling motor protein dynamics during RNA translocation with a Hierarchical Hidden Markov Model (HHMM), SegPore segments the raw signal into small, biologically meaningful fragments, each corresponding to a k-mer sub-state, which substantially reduces noise and improves segmentation accuracy. After segmentation, these fragments are aligned to the reference sequence and concatenated into larger events, analogous to Nanopolish’s “eventalign” output, which serve as the foundation for downstream analyses. Moreover, the “eventalign” results produced by SegPore enhance interpretability in RNA modification estimation. While deep learning–based tools such as m6Anet classify RNA modifications using complex, non-transparent features (see Supplementary Fig. S5), SegPore employs a simple Gaussian Mixture Model (GMM) to distinguish modified from unmodified nucleotides based on baseline current levels. This transparent modeling approach improves confidence in the predictions and makes SegPore particularly well-suited for biological applications where interpretability is essential."

Line 104: Please change "normal adenosine" to "adenosine".

We have revised the manuscript as requested and replaced all instances of “normal adenosine” with “adenosine” throughout the text.

Materials and MethodsLine 176: Please reword "...we standardize the raw current signals across reads, ensuring that the mean and standard deviation of the poly(A) tail are consistent across all reads." To "...we standardize the raw current signals for each read, ensuring that the mean and standard deviation are consistent across the poly(A) tail region."

We have changed sentence as requested.

“Since the poly(A) tail provides a stable reference, we standardize the raw current signals for each read, ensuring that the mean and standard deviation are consistent across the poly(A) tail region.”

Line 182: Please describe the RNA translocation hypothesis, as this is the first mention of it in the text. Also, why is the Hierachical Hidden Markov model perfect for addressing the RNA translocation hypothesis? Explain more about how the HHMM works and why it is a suitable choice.

We have revised the sentence as requested:

“The RNA translocation hypothesis (see details in the first section of Results) naturally leads to the use of a hierarchical Hidden Markov Model (HHMM) to segment the raw current signal.”

The motivation of the HHMM is explained in detail in the the first section “RNA translocation hypothesis” of Results. As illustrated in Figure 2, the sequencing data suggest that RNA molecules may translocate back and forth (often referred to as jiggling) while passing through the nanopore. This behavior results in complex current fluctuations that are challenging to model with a simple HMM. The HHMM provides a natural framework to address this because it can model signal dynamics at two levels. The outer HMM distinguishes between two major states — base states (where the signal corresponds to a stable sub-state of a k-mer) and transition states (representing transitions from one base state to the next). Within each base state, an inner HMM models finer signal variation using three states — “curr”, “prev”, and “next” — corresponding to the current k-mer sub-states and its neighboring k-mer sub-states. This hierarchical structure captures both the stable signal patterns and the stochastic translocation behavior, enabling more accurate and biologically meaningful segmentation of the raw current signal.

Line 184: do you mean HHMM? Please be consistent throughout the text.

As explained in the previous response, the HHMM consists of two layers: an outer HMM and an inner HMM. The term “HMM” in line 184 is meant to be read together with “inner” at the end of line 183, forming the phrase “inner HMM.” It seems the reviewer may have overlooked this when reading the text.

Line 203: please delete: "It is obviously seen that".

We have removed the phrase “It is obviously seen that” from the sentence as requested. The revised sentence now reads:

“The first part of Eq. 2 represents the emission probabilities, and the second part represents the transition probabilities.”

Line 314, GMM for 5mer parameter table re-estimation: "Typically, the process is repeated three to five times until the5mer parameter table stabilizes." How is the stabilisation of the 5mer parameter table quantified? What is a reasonable cut-off that would demonstrate adequate stabilisation of the 5mer parameter table? Please add details of this to the text.

We have revised the sentence to clarify the stabilization criterion as follows:

“Typically, the process is repeated three to five times until the 5-mer parameter table stabilizes (when the average change of mean values of all 5-mers is less than 5e-3).”

ResultsLine 377: Please edit to read "Traditional base calling algorithms such as Guppy and Albacore assume that the RNA molecule is translocated unidirectionally through the pore by the motor protein."

We have revised the sentence as:

“In traditional basecalling algorithms such as Guppy and Albacore, we implicitly assume that the RNA molecule is translocated through the pore by the motor protein in a monotonic fashion, i.e., the RNA is pulled through the pore unidirectionally.”

Line 555, m6A identification at the site level: "For six selected m6A motifs, SegPore achieved an ROC AUC of 82.7% and a PR AUC of 38.7%, earning the third best performance compared with deep leaning methods m6Anet and CHEUI (Fig. 3D)." So SegPore performs third best of all deep learning methods. Do you recommend its use in conjunction with m6Anet for m6A detection? Please clarify in the text. This will help to guide users to possible best practice uses of your software.

Thank you for the suggestion. We have added a clarification in the revised manuscript to guide users.

“For practical applications, we recommend taking the intersection of m6A sites predicted by SegPore and m6Anet to obtain high-confidence modification sites, while still benefiting from the interpretability provided by SegPore’s predictions.”

Figures.Figure 1A please refer to poly(A) tail, rather than polyA tail.

We have updated it to poly(A) tail in the revised manuscript.